# HypeBoy: Generative Self-Supervised Representation Learning on Hypergraphs

**Sunwoo Kim**[†] **Shinhwan Kang**[†] **Fanchen Bu**[‡] **Soo Yong Lee**[†] **Jaemin Yoo**[‡] **Kijung Shin**[†‡]
[†] Kim Jaechul Graduate School of AI, [‡] School of Electrical Engineering
Korea Advanced Institute of Science and Technology (KAIST)
{kswoo97, shinhwan.kang, boqvezen97, syleetolow, jaemin, kijungs}@kaist.ac.kr

## Abstract

Hypergraphs are marked by complex topology, expressing higher-order interactions among multiple nodes with hyperedges, and better capturing the topology is essential for effective representation learning. Recent advances in generative self-supervised learning (SSL) suggest that hypergraph neural networks learned from generative self-supervision have the potential to effectively encode the complex hypergraph topology. Designing a generative SSL strategy for hypergraphs, however, is not straightforward. Questions remain with regard to its generative SSL task, connection to downstream tasks, and empirical properties of learned representations. In light of the promises and challenges, we propose a novel generative SSL strategy for hypergraphs. We first formulate a generative SSL task on hypergraphs, *hyperedge filling*, and highlight its theoretical connection to node classification. Based on the generative SSL task, we propose a hypergraph SSL method, HypeBoy. HypeBoy learns effective general-purpose hypergraph representations, outperforming 16 baseline methods across 11 benchmark datasets. Code and datasets are available at https://github.com/kswoo97/hypeboy.

## 1 Introduction

Many real-world interactions occur among multiple entities, such as online group discussions on social media, academic collaboration of researchers, and joint item purchases (Benson et al., 2018; Kim et al., 2023a; Lee et al., 2024). Representing such higher-order interactions with an ordinary graph can cause topological information loss (Dong et al., 2020; Yoon et al., 2020). Thus, hypergraphs have emerged as an effective tool for representing high-order interactions in various domains, including recommender systems (Xia et al., 2022; Yu et al., 2021), financial analysis (Sawhney et al., 2021; 2020), and drug analysis (Ruan et al., 2021; Saifuddin et al., 2023).

For representation learning on such a complex topology, hypergraph neural networks (HNNs) have been developed. Training HNNs has primarily relied on task-related label supervision, such as external node labels. However, simply learning from external supervision may limit HNNs in capturing more complex patterns in hypergraph topology. Incorporating self-supervision related to topology, hence, can substantially improve HNNs' representation learning.

Particularly, *generative self-supervised learning* (SSL) holds promise for effective hypergraph representation learning. Generative SSL has recently shown remarkable success in encoding complex patterns in multiple domains. GPT (OpenAI, 2023) in natural language processing and Masked Autoencoder (He et al., 2022) in computer vision are notable examples. With generative self-supervision, HNNs may encode complex topology more effectively, leading to improved representation learning.

However, designing a generative SSL strategy for hypergraphs is not straightforward. First, questions remain unanswered for generative *SSL tasks*: **(1.a)** What should be the target generative SSL task for HNNs? **(1.b)** How does the generative SSL task relate to downstream tasks (e.g., node classification with external labels)? Second, even after determining the task, details of the *SSL method* (based on the task) remain unspecified. **(2.a)** What empirical properties should the method aim to satisfy? **(2.b)** How can the method achieve effective general-purpose hypergraph representations? Moreover, if not carefully designed, the generative SSL strategy can suffer from severe computational burden, as the number of potential hyperedges grows exponentially with the number of nodes.

In light of these promises and challenges, we systematically investigate and propose a hypergraph generative SSL strategy. Our contributions and the rest of the paper are organized as follows:

- **SSL Task**: We formulate a generative SSL task, *hyperedge filling*, for hypergraph representation learning. Notably, we establish its theoretical connections to node classification (Section 3).
- **SSL Method**: Based on the hyperedge filling task, we propose HYPEBOY, a novel hypergraph SSL method. HYPEBOY is designed to satisfy desirable properties of hypergraph SSL, mitigating (1) over-emphasized proximity, (2) dimensional collapse, and (3) non-uniformity/-alignment of learned representations (Section 4).
- **Experiments**: We demonstrate that HYPEBOY learns effective general-purpose hypergraph representations. It significantly outperforms SSL-based HNNs in both node classification and hyperedge prediction across 11 benchmark hypergraph datasets (Section 5; code and datasets are available at https://github.com/kswoo97/hypeboy).

## 2 RELATED WORK

In this section, we review the literature on hypergraph neural networks and self-supervised learning.

**Hypergraph neural networks (HNNs).** HNNs learn hypergraph representations. Converting hyperedges into cliques (fully connected subgraphs) allows graph neural networks to be applied to hypergraphs (Feng et al., 2019; Yadati et al., 2019). Such conversion, however, may result in topological information loss, since high-order interactions (hyperedges) are reduced to pair-wise interactions (edges). As such, most HNNs pass messages through hyperedges to encode hypergraphs. Some notable examples include HNHN (Dong et al., 2020) with hyperedge encoders, UniGNN (Huang & Yang, 2021) with generalized message passing functions for graphs and hypergraphs, AllSet (Chien et al., 2022) with set encoders, ED-HNN (Wang et al., 2023a) with permutation-equivariant diffusion operators, and PhenomNN (Wang et al., 2023b) with hypergraph-regularized energy functions.

**Self-supervised learning (SSL).** SSL strategies aim to learn representation from the input data itself, without relying on external labels. They can largely be categorized into contrastive or generative types. Contrastive SSL aims to maximize the agreement between data obtained from diverse views (Chen et al., 2020; Grill et al., 2020; You et al., 2020). Generative SSL, on the other hand, predicts or reconstructs parts of the input data. The success of generative SSL demonstrates its strengths in learning complex input data, in domains including natural language processing (Devlin et al., 2019; OpenAI, 2023) and computer vision (He et al., 2022; Tong et al., 2022). Recently, generative SSL for graphs has gained significant attention, with their main focuses on reconstructing edges (Tan et al., 2023; Li et al., 2023) or node features (Hou et al., 2022; 2023).

**Self-supervised learning on hypergraphs.** The interest in SSL for hypergraphs is on the rise. Early hypergraph SSL strategies mainly targeted specific downstream tasks, such as group (Zhang et al., 2021) and session-based recommendation (Xia et al., 2022). Recent ones aim to obtain general-purpose representation. TriCL (Lee & Shin, 2023a) utilizes a tri-directional contrastive loss, which consists of node-, hyperedge-, and membership-level contrast. Kim et al. (2023c) enhances the scalability of TriCL with a partitioning technique. HyperGCL (Wei et al., 2022) utilizes neural networks to generate views for contrast and empirically demonstrates its superiority in node classification, outperforming rule-based view generation methods. HyperGRL (Du et al., 2022) is a generative SSL method sharing similar spirits with our approach; however, the underlying motivations and methodological designs are markedly distinct. We provide a detailed comparison in Appendix C.1.

## 3 PROPOSED TASK AND THEORETICAL ANALYSIS

In this section, after providing some preliminaries, we formulate *hyperedge filling*, our generative SSL task on hypergraphs. Then, we establish a theoretical connection between hyperedge filling and node classification, which is a widely-considered important downstream task.

**Preliminaries.** A hypergraph $\mathcal{G} = (\mathcal{V}, \mathcal{E})$ is defined by a node set $\mathcal{V}$ and a hyperedge set $\mathcal{E}$. Each hyperedge $e_j \in \mathcal{E}$ is a non-empty set of nodes (i.e., $\emptyset \neq e_j \subseteq \mathcal{V}, \forall e_j \in \mathcal{E}$). Each node $v_i \in \mathcal{V}$ is equipped with a feature vector $\boldsymbol{x}_i \in \mathbb{R}^d$, and $\mathbf{X} \in \mathbb{R}^{|\mathcal{V}| \times d}$ denotes the node feature matrix where the $i$-th row $\mathbf{X}_i$ corresponds to $\boldsymbol{x}_i$.

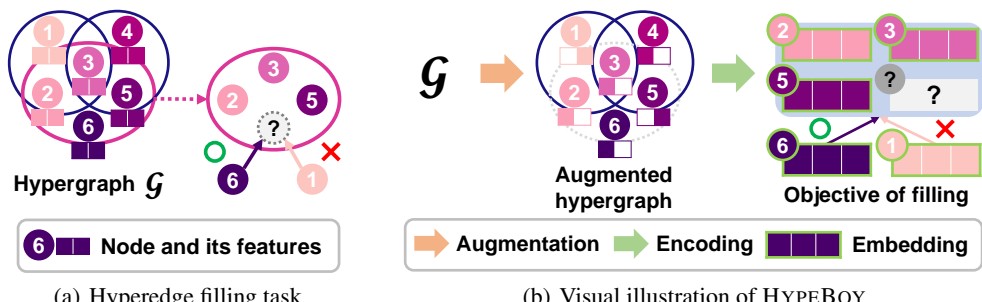

(a) Hyperedge filling task        (b) Visual illustration of HYPEBOY

Figure 1: Overview of (a) the hyperedge filling task and (b) HYPEBOY, our proposed SSL method based on the task. The goal of the task is to find the missing node for a given query subset (i.e., the other nodes in a hyperedge). HYPEBOY trains HNNs aiming to correctly predict the missing node.

A hypergraph neural network (HNN) $f_\theta$ is a function that receives a node feature matrix $\mathbf{X}$ and a set of hyperedges $\mathcal{E}$ as inputs to return node embeddings $\mathbf{Z} \in \mathbb{R}^{|\mathcal{V}| \times k}$ (i.e., $\mathbf{Z} = f_\theta(\mathbf{X}, \mathcal{E})$, where $\theta$ is a set of learnable parameters) [1].

## 3.1 PROPOSED TASK: HYPEREDGE FILLING

We formulate *hyperedge filling*, a generative SSL task for hypergraph representation learning. We first define the task and discuss the superiority of the hyperedge filling task over alternatives. An illustration of the hyperedge filling task is provided in Figure 1(a).

**Task definition.** Given a set of nodes, hyperedge filling aims to predict a node that is likely to form a hyperedge together. Specifically, for each hyperedge $e_j \in \mathcal{E}$, we divide it into a (missing) node $v_i \in e_j$ and a (query) subset $q_{ij} = e_j \setminus \{v_i\}$. The target of the task is to correctly fill the missing node $v_i$ for each given subset $q_{ij}$. This can be formalized by maximizing the probability of $v_i$ correctly completing $q_{ij}$, which is denoted as $p_{(\mathbf{X}, \mathcal{E}, \Theta)}(v_i \mid q_{ij})$, where $\Theta$ is a set of parameters we aim to optimize in this task. We will further elaborate on our design of $p_{(\mathbf{X}, \mathcal{E}, \Theta)}(\cdot)$ in Section 4.3.

**Advantage over alternatives.** Potential alternatives include naive extensions of generative SSL tasks for ordinary graphs: (a) generating hyperedges from scratch and (b) classifying given sets of nodes into hyperedges and non-hyperedges. Compared to (a), by shifting the focus of prediction from the set level (hyperedge itself) to the node level, the hyperedge filling task reduces the prediction space from computationally prohibitive $O(2^{|\mathcal{V}|})$ to affordable $O(|\mathcal{V}|)$. Compared to (b), the hyperedge filling task provides richer and more diversified generative SSL signals. Specifically, for each hyperedge $e_j$, our task offers $|e_j|$ distinct node-subset combinations that can serve as SSL signals. In contrast, classifying the mere existence of $e_j$ yields a singular and, thus, limited signal.

## 3.2 THEORETICAL RESULTS ON HYPEREDGE FILLING

To demonstrate the effectiveness of hyperedge filling as a general SSL task, we present its theoretical connection to node classification. In essence, we demonstrate that node representations optimized for the hyperedge filling task can improve node classification accuracy.

### 3.2.1 BASIC SETTING

First, we assume a data model of a hypergraph $\mathcal{G} = (\mathcal{V}, \mathcal{E})$ where (1) each node belongs to a single class, (2) the features of each node are generated from a Gaussian distribution, and (3) each hyperedge is generated according to a given hyperedge affinity parameter $\mathcal{P} \in [0, 1]$.

**Assumption 1** (Node classes and features). *Assume that there are $2N$ nodes and node classes $C_1$ and $C_0$ such that $C_1 \cup C_0 = \mathcal{V}, C_1 \cap C_0 = \emptyset$, and $|C_1| = |C_0| = N$. Each node feature vector $\boldsymbol{x}_i$ is independently generated from $\mathcal{N}(\boldsymbol{x}; \boldsymbol{\mu}_1, \boldsymbol{\Sigma})$ if $v_i \in C_1$, and $\mathcal{N}(\boldsymbol{x}; \boldsymbol{\mu}_0, \boldsymbol{\Sigma})$ if $v_i \in C_0$. For simplicity, we assume $\boldsymbol{\mu}_1 = (0.5)_{i=1}^d$, $\boldsymbol{\mu}_0 = (-0.5)_{i=1}^d$, and $\boldsymbol{\Sigma} = \mathbf{I}$, where $\mathbf{I}$ is the $d$-by-$d$ identity matrix.*

---

[1]In this paper, we assume an HNN returns only vector representations of nodes unless otherwise stated, while we acknowledge that some HNNs return embeddings of hyperedges as well.

**Assumption 2** (Hypergraph topology). *Assume that the number of hyperedges and the size of each hyperedge are given, and there is no singleton hyperedge (i.e., $|e_j| \geq 2, \forall e_j \in \mathcal{E}$). Let B denote the binomial distribution. Each hyperedge $e_j \in \mathcal{E}$ has a class membership $c_j \in \{0,1\}$, where $c_j \sim B(1, 0.5)$. Given the number $|e_j|$ of nodes and the class $c_j \in \{0,1\}$ of a hyperedge, the number of its members belonging to $C_1$ (i.e., $|e_j \cap C_1|$) $\sim B(|e_j|, \mathcal{P}^{c_j}(1-\mathcal{P})^{1-c_j})$.*

Note that the tendency of each hyperedge containing nodes of the same class is symmetric about $\mathcal{P} = 0.5$ under the binary class setting. In addition, when $\mathcal{P}$ approaches 1, each hyperedge $e_j$ is more likely to contain nodes of the same class (spec., node class $c_j$); as $\mathcal{P}$ approaches 0, each hyperedge $e_j$ is again more likely to contain nodes of the same class (spec., node class $1 - c_j$).

Second, we describe how node representations are updated via the hyperedge filling task. In this theoretical analysis, we define the updating process of node representations as follows:

**(F1)** Filling probability $p_{(\mathbf{X}, \mathcal{E}, \Theta)}(\cdot)$ is defined on each node-subset pair as follows:

$$p_{(\mathbf{X}, \mathcal{E}, \Theta)}(v_i \mid q_{ij}) := \frac{\exp(\boldsymbol{x}_i^T(\sum_{v_k \in q_{ij}} \boldsymbol{x}_k))}{\sum_{v_t \in \mathcal{V}} \exp(\boldsymbol{x}_t^T(\sum_{v_k \in q_{ij}} \boldsymbol{x}_k))}. \tag{1}$$

**(F2)** Node representation $\boldsymbol{z}_i$ is obtained via gradient descent with respect to $\boldsymbol{x}_i$ from $\mathcal{L}$, which is the negative log-likelihood of Eq. (1), (i.e., $\mathcal{L} = -\log p_{(\mathbf{X}, \mathcal{E}, \Theta)}(v_i \mid q_{ij})$). For ease of analysis, we assume $\boldsymbol{z}_i = \boldsymbol{x}_i - \gamma \nabla_{\boldsymbol{x}_i} \mathcal{L}$, where $\gamma \in \mathbb{R}^+$ is a fixed constant.

At last, we assume a Gaussian naive Bayes classifier $\mathcal{F}$ (Bishop, 2006), which is defined as:

$$\mathcal{F}(\boldsymbol{x}_i) = \underset{k \in \{0,1\}}{\arg\max} \, f(\boldsymbol{x}_i; \boldsymbol{\mu}_k, \mathbf{I}), \text{ where } f \text{ is the probability density function of } \mathcal{N}(\boldsymbol{x}; \boldsymbol{\mu}_k, \mathbf{I}). \tag{2}$$

### 3.2.2 HYPEREDGE FILLING HELPS NODE CLASSIFICATION

Our goal is to show that for accurate classification of $v_i$, the representation $\boldsymbol{z}_i$, which is obtained for hyperedge filling as described in (**F1**) and (**F2**), is more effective than the original feature $\boldsymbol{x}_i$. First, we assume a node $v_i$ belonging to the class $C_1$ (i.e., $v_i \in C_1$), and we later generalize the result to $C_0$. The effectiveness of an original feature is defined as the expected accuracy of a classifier $\mathcal{F}$ with $\boldsymbol{x}_i$ (i.e., $\mathbb{E}_{\boldsymbol{x}}[\mathbf{1}_{\mathcal{F}(\boldsymbol{x}_i)=1}] := P_{\boldsymbol{x}}(f(\boldsymbol{x}_i; \boldsymbol{\mu}_1, \mathbf{I}) > f(\boldsymbol{x}_i; \boldsymbol{\mu}_0, \mathbf{I}))$). Similarly, that with a derived representation is defined as $\mathbb{E}_{\boldsymbol{z}}[\mathbf{1}_{\mathcal{F}(\boldsymbol{z}_i)=1}] = P_{\boldsymbol{z}}(f(\boldsymbol{z}_i; \boldsymbol{\mu}_1, \mathbf{I}) > f(\boldsymbol{z}_i; \boldsymbol{\mu}_0, \mathbf{I}))$. Below, we show that the effectiveness of a derived representation $\boldsymbol{z}_i$ is greater than that of an original feature $\boldsymbol{x}_i$ under a certain condition (Theorem 1).

**Theorem 1** (Improvement in effectiveness). *Assume a hyperedge $e_j$ s.t. $e_j \cap C_1 \neq \emptyset$ and node features $\mathbf{X}$ that are generated under Assumption 1. For any node $v_i \in e_j \cap C_1$, the following holds:*

$$\left(\vec{\mathbf{1}}^T \sum_{v_k \in q_{ij}} \boldsymbol{x}_k > 0\right) \Rightarrow \mathbb{E}_{\boldsymbol{z}}[\mathbf{1}_{\mathcal{F}(\boldsymbol{z}_i)=1}] > \mathbb{E}_{\boldsymbol{x}}[\mathbf{1}_{\mathcal{F}(\boldsymbol{x}_i)=1}], \text{ where } \vec{\mathbf{1}} \text{ denotes } (1)_{k=1}^d. \tag{3}$$

*Proof.* A full proof is provided in Appendix A.1. □

Theorem 1 states that when a certain condition (specified in the parentheses in Eq. (3)) is met, the effectiveness of $\boldsymbol{z}_i$ is greater than that of $\boldsymbol{x}_i$. This result implies that node representations, when refined using the objective function associated with the hyperedge filling task, are more proficient in performing accurate node classification compared to the original node features.

While the finding in Theorem 1 demonstrates the usefulness of the hyperedge filling task in node classification, its validity relies on the condition (in the parentheses in Eq. (3)). We further analyze the probability that the condition is met by stochastic $G$ under Assumptions 1 and 2 for a given $\mathcal{P}$.

**Theorem 2** (Realization of condition). *Assume node features $\mathbf{X}$ and a hyperedge $e_j$ s.t. (i) generated under Assumption 1 and 2 respectively, and (ii) $e_j \cap C_1 \neq \emptyset$. For any $q_{ij}$ where $v_i \in e_j \cap C_1$, the following holds:*

*1.* $P_{\boldsymbol{x},e}\left(\mathbf{1}^T \sum_{v_k \in q_{ij}} \boldsymbol{x}_k > 0 \,\middle|\, \mathcal{P}\right) \geq 0.5, \forall \mathcal{P} \in [0,1].$

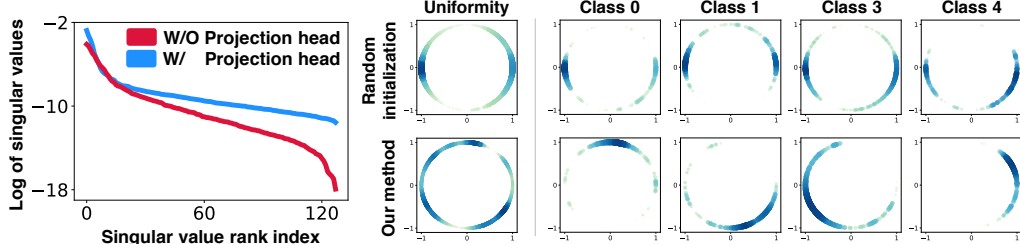

(a) Representation spectrum analysis. A sudden singular-value drop implies **dimensional collapse**.

(b) Representations on the unit hypersphere. Uniformly distributed representations achieve **uniformity**, and representations of nodes of the same class located close to each other achieve **alignment**.

Figure 2: Analysis regarding Property 2 (prevention of dimensional collapse) and Property 3 (representation uniformity and alignment) of HYPEBOY. As shown in (a), while HYPEBOY without projection heads (red) suffers from dimensional collapse, HYPEBOY (blue) does not, demonstrating the necessity of the projection head. Furthermore, as shown in (b), representations from an HNN trained by HYPEBOY meet both uniformity and alignment, justifying our design choice of the loss function. Experiments are conducted on the Cora dataset.

2. $P_{\boldsymbol{x},e}\left(\mathbf{1}^T \sum_{v_k \in q_{ij}} \boldsymbol{x}_k > 0 \,\Big|\, \mathcal{P}\right)$ *is a strictly decreasing function w.r.t.* $\mathcal{P} \in [0, 0.5]$ *and strictly increasing function w.r.t.* $\mathcal{P} \in [0.5, 1]$.

*Proof.* A full proof is provided in Appendix A.2. □

Theorem 2 states that the probability of the condition being satisfied is at least 0.5 for any affinity parameter $\mathcal{P} \in [0, 1]$. Moreover, the probability strictly increases w.r.t. $\mathcal{P} \in [0.5, 1]$ and that strictly decreases w.r.t. $\mathcal{P} \in [0, 0.5]$. Thus, it can be inferred that as the probability of each hyperedge including nodes of the same class increases, so does the likelihood of the condition being met. Notably, many real-world group interactions exhibit homophilic traits (Laakasuo et al., 2020; Khanam et al., 2023). Therefore, the hyperedge filling task can improve node classification in many real-world scenarios, as evidenced by our theoretical findings and real-world characteristics.

**Generalization to the class** $C_0$**.** The above results can be easily generalized to the class $C_0$ due to the symmetry. Specifically, for each node $v_i \in C_0$, the effectiveness (spec., the expected accuracy of the classifier $\mathcal{F}$) of a derived representation $\boldsymbol{z}_i$ is greater than that of an original feature $\boldsymbol{x}_i$ under a certain condition. The probability for such a condition to hold strictly increases w.r.t. $\mathcal{P} \in [0.5, 1]$ and strictly decreases w.r.t. $\mathcal{P} \in [0, 0.5]$. We theoretically show this in Appendix A.1 and A.2.

## 4 PROPOSED METHOD FOR HYPEREDGE FILLING

In this section, we present **HYPEBOY** (**Hype**rgraphs, **b**uild **o**wn h**y**peredges), a hypergraph generative SSL method based on the hyperedge filling task (Section 3). HYPEBOY exhibits the following desirable properties of hypergraph generative SSL, as empirically demonstrated later:

**Property 1.** Does not over-emphasize proximity information (Veličković et al., 2019).

**Property 2.** Avoids dimensional collapse (Jing et al., 2022) in node representations.

**Property 3.** Learns node representation to be aligned and uniform (Wang & Isola, 2020).

Our proposed method, HYPEBOY, is illustrated in Figure 1(b). After presenting each of its steps, we discuss its role in satisfying the above properties. Lastly, we introduce a two-stage training scheme for further enhancing HYPEBOY.

### 4.1 STEP 1: HYPERGRAPH AUGMENTATION

HYPEBOY first obtains augmented feature matrix $\mathbf{X}'$ and hyperedge set $\mathcal{E}'$ by using augmentation functions $\boldsymbol{\tau}_{\boldsymbol{x}}$ and $\boldsymbol{\tau}_{\mathcal{E}}$, respectively. The feature augmentation function $\boldsymbol{\tau}_{\boldsymbol{x}} : (\mathbf{X}, p_v) \mapsto \mathbf{X}'$ masks certain entries of $\mathbf{X}$ based on Bernoulli sampling (spec., $\mathbf{X}' = \mathbf{X} \odot \mathbf{M}$, where $\odot$ is an element-wise product and $\mathbf{M}_{ij} \sim B(1, 1 - p_v), \forall i \in [|\mathcal{V}|], j \in [d]$). The topology augmentation function

$\tau_{\mathcal{E}} : (\mathcal{E}, p_e) \mapsto \mathcal{E}'$ samples $\lceil |\mathcal{E}|(1 - p_e) \rceil$ hyperedges uniformly at random from $\mathcal{E}$. Note that the magnitudes of feature and topology augmentations are proportional to $p_v, p_e \in [0, 1]$, respectively.

**Role.** For hypergraph SSL, augmentations are crucial for mitigating overly-emphasized proximity information. That is, using all hyperedges and/or features for both message passing and objective-function construction may risk HNNs to heavily rely on direct neighborhood information (Tan et al., 2023). Many graph SSL strategies mask certain input edges and/or node features, preventing their encoder models from overfitting to the neighbor distribution and features (Hou et al., 2022; Li et al., 2023; Tan et al., 2023). Motivated by their findings, we use the augmentation step. In Appendix F.1, we demonstrate that augmentation enhances node classification performance of HYPEBOY.

## 4.2 STEP 2: HYPERGRAPH ENCODING

After augmentation, HYPEBOY obtains node and query subset representations. First, HYPE-BOY obtains node embeddings $\mathbf{Z} \in \mathbb{R}^{|\mathcal{V}| \times d'}$ by feeding the augmented hypergraph into an encoder HNN: $\mathbf{Z} = f_\theta(\mathbf{X}', \mathcal{E}')$. Then, HYPEBOY obtains projected representations of query subsets (i.e., $q_{ij}, v_i \in e_j, e_j \in \mathcal{E}$) and nodes. To this end, we utilize a node projection head $f'_\phi$ and a set projection head $f''_\psi$. Specifically, projected embeddings of node $v_i$ and query subset $q_{ij}$ are $\mathbf{h}_i = f'_\phi(\mathbf{z}_i) \in \mathbb{R}^k$ and $\mathbf{q}_{ij} = f''_\rho(\sum_{v_t \in q_{ij}} \mathbf{z}_t) \in \mathbb{R}^k$, respectively. Here, the design choice of the set projection head is motivated by Deep Sets (Zaheer et al., 2017).

**Role.** We investigate the role of the projection heads, which are non-trivial components, in the context of the dimensional collapse of embeddings. Dimensional collapse is a phenomenon in which embedding vectors occupy only the lower dimensional sub-space of their full dimension (Jing et al., 2022). This is identified by observing whether or not certain singular values of the embedding covariance matrix drop to zero. To prevent dimensional collapse, we employ projection heads in HY-PEBOY, and this is in line with the prior findings of Jing et al. (2022) and Song et al. (2023). Figure 2(a) illustrates that an HNN trained using HYPEBOY avoids dimensional collapse, whereas an HNN trained with a HYPEBOY variant (without projection heads) does not. Results on more datasets are in Appendix F.2. Furthermore, we provide a theoretical analysis of why HYPEBOY without projection heads may suffer from dimensional collapse in Appendix B.2. Note that this distinction leads to a performance discrepancy in node classification (Section 5.3).

## 4.3 STEP 3: HYPEREDGE FILLING LOSS

The last step is to compute the SSL loss based on the hyperedge filling probability. We design $p_{(\mathbf{X}, \mathcal{E}, \Theta)}(v_k \mid q_{ij})$ to be normalized over $\mathcal{V}$ (i.e., $\sum_{v_k \in \mathcal{V}} p_{(\mathbf{X}, \mathcal{E}, \Theta)}(v_k \mid q_{ij}) = 1$). To this end, we utilize a Softmax function to model probabilities. In sum, with projected embeddings $\mathbf{h}_i$ and $\mathbf{q}_{ij}$ (Section 4.2), the probability of a node $v_i$ completing a query subset $q_{ij}$ is defined as follows:

$$p_{(\mathbf{X}, \mathcal{E}, \Theta)}(v_i \mid q_{ij}) := \frac{\exp(\texttt{sim}(\mathbf{h}_i, \mathbf{q}_{ij}))}{\sum_{v_k \in \mathcal{V}} \exp(\texttt{sim}(\mathbf{h}_k, \mathbf{q}_{ij}))}, \tag{4}$$

where $\texttt{sim}$ is the cosine similarity function (other similarity functions are also applicable). Lastly, HYPEBOY is optimized for all possible hyperedge filling cases (i.e., $\Pi_{e_j \in \mathcal{E}} \Pi_{v_i \in e_j} p_{(\mathbf{X}, \mathcal{E}, \Theta)}(v_i \mid q_{ij})$). To sum up, HYPEBOY minimizes the negative log-likelihood of all possible cases as follows:

$$\mathcal{L} := -\sum_{e_j \in \mathcal{E}} \sum_{v_i \in e_j} \log \frac{\exp(\texttt{sim}(\mathbf{h}_i, \mathbf{q}_{ij}))}{\sum_{v_k \in \mathcal{V}} \exp(\texttt{sim}(\mathbf{h}_k, \mathbf{q}_{ij}))}. \tag{5}$$

Note that the set $\Theta$ of all parameters consists of the parameters of the encoder HNN $f_\theta$, the node projection head $f'_\phi$, and the set projection head $f''_\rho$ (i.e., $\Theta = (\theta, \phi, \rho)$). They are updated by gradient descent, aiming to minimize the loss $\mathcal{L}$ defined in Eq. (5).

**Role.** Our design choice of $p_{(\mathbf{X}, \mathcal{E}, \Theta)}(\cdot)$ ensures that representations learned by HYPEBOY achieve both alignment and uniformity (Wang & Isola, 2020). In our case, *alignment* indicates that node embeddings belonging to the same class are closely located to each other, and *uniformity* indicates that embeddings are uniformly distributed over the embedding space. Through the numerator of Eq. (5), HYPEBOY pulls representations of a missing node and a query subset, promoting the alignment as discussed in our theoretical analysis (Section 3.2). At the same time, the denominator of

Eq. (5) pushes away every node representation from each query subset representation, encouraging that representations are uniformly distributed. This intuition is supported by the findings of Wang & Isola (2020): the denominators of their contrastive loss, which push away representations from each other, encourage uniformity. As shown in Figure 2(b), we verify that node representations from HYPEBOY achieve both alignment and uniformity. Results on more datasets are in Appendix F.3.

## 4.4 TWO-STAGE TRAINING SCHEME FOR FURTHER ENHANCEMENT

In our preliminary study, we observed that with a randomly initialized encoder, HYPEBOYtends to rely too heavily on the projection heads rather than the encoder parameters, leading to sub-optimal training. Thus, we propose a **two-stage training scheme** to enhance the effectiveness of HYPEBOY. In a nutshell, we first train an HNN $f_\theta$ to reconstruct masked node features (inspired by Hou et al. (2022)). Then, we use the trained $f_\theta$ parameters for the initialization of HYPEBOY's encoder to learn hyperedge filling (Sections 4.1-4.3). This feature warm-up for the HNN encoder repeats the following three steps for a fixed number of epochs, which we set to 300 in all our experiments.

**Step 1: Encoding.** For each feature-reconstruction training epoch, we sample a certain number of nodes uniformly at random from $\mathcal{V}$, which we denote as $\mathcal{V}' \subseteq \mathcal{V}$. Then, we mask the features of sampled nodes by using a learnable input token $\boldsymbol{m}^{(I)} \in \mathbb{R}^d$ and use the resulting masked node feature matrix $\mathbf{X}'' \in \mathbb{R}^{|\mathcal{V}| \times d}$, as the input node feature matrix (i.e., $\mathbf{X}_i'' = \boldsymbol{m}^{(I)}, \forall v_i \in \mathcal{V}'$ and $\mathbf{X}_j'' = \mathbf{X}_j, \forall v_j \in \mathcal{V} \setminus \mathcal{V}'$). We also obtain augmented hyperedges $\mathcal{E}'$ by using the strategy described in Section 4.1. Then, we obtain node embeddings $\mathbf{Z}' \in \mathbb{R}^{|\mathcal{V}| \times d'}$ as follows: $\mathbf{Z}' = f_\theta(\mathbf{X}'', \mathcal{E}')$.

**Step 2: Decoding.** We again mask the embeddings of the nodes that we sampled earlier with a learnable embedding token $\boldsymbol{m}^{(E)} \in \mathbb{R}^{d'}$, obtaining the masked node embedding matrix $\mathbf{Z}''$ (i.e., $\mathbf{Z}_i'' = \boldsymbol{m}^{(E)}, \forall v_i \in \mathcal{V}'$ and $\mathbf{Z}_j'' = \mathbf{Z}_j', \forall v_j \in \mathcal{V} \setminus \mathcal{V}'$). Then, we acquire reconstructed node features $\hat{\mathbf{X}} \in \mathbb{R}^{|\mathcal{V}| \times d}$ by using a decoder HNN $g_\psi$ as follows: $\hat{\mathbf{X}} = g_\psi(\mathbf{Z}'', \mathcal{E}')$.

**Step 3: Updating.** After gaining reconstructed node features $\hat{\mathbf{X}}$, we maximize the similarity between the original and reconstructed features of the masked nodes. To this end, we minimize the following loss function, which is proposed by Hou et al. (2022): $\mathcal{L}' = (1/|\mathcal{V}'|) \sum_{v_i \in \mathcal{V}'} (1 - (\hat{\mathbf{X}}_i^T \mathbf{X}_i / \|\hat{\mathbf{X}}_i\| \|\mathbf{X}_i\|))$. Using gradient descent to minimize $\mathcal{L}'$, we update the parameters of the encoder HNN $f_\theta$, the decoder HNN $g_\psi$, the input token $\boldsymbol{m}^{(I)}$, and the embedding token $\boldsymbol{m}^{(E)}$.

## 5 EXPERIMENTAL RESULTS

We now evaluate the efficacy of HYPEBOY as techniques for (1) pre-training hypergraph neural networks (HNNs) for node classification (Section 5.1) and (2) learning general-purpose representations (Section 5.2). Then, we justify each of its components through an ablation study (Section 5.3).

**Datasets.** For experiments, we use 11 benchmark hypergraph datasets. The hypergraph datasets are from diverse domains, expressing co-citation, co-authorship, computer graphics, movie-actor, news, and political membership relations. In Appendix D, we detail their statistics and descriptions.

**Baselines methods.** We utilize 16 baseline methods. They include (a) 10 *(semi-)supervised HNNs*, including 2 state-of-the-art HNNs (ED-HNN (Wang et al., 2023a) and PhenomNN (Wang et al., 2023b)), (b) 2 *generative SSL* strategies for ordinary graphs (GraphMAE2 (Hou et al., 2023) and MaskGAE (Li et al., 2023)), and (c) 4 *SSL* strategies for hypergraph (TriCL (Lee & Shin, 2023a), HyperGCL (Wei et al., 2022), HyperGRL (Du et al., 2022), and H-GD, which is a direct extension of an ordinary graph SSL method (Zheng et al., 2022) to hypergraphs). We use UniGCNII (Huang & Yang, 2021) [2] and GCN (Kipf & Welling, 2017) as the backbone encoders for hypergraph SSL methods and ordinary graph SSL methods, respectively [3]. In Appendix E, we provide their details, including their implementations, training, and hyperparameters.

**HYPEBOY.** We utilize UniGCNII as an encoder of HYPEBOY, which is the same as that of other hypergraph SSL methods. For both node- and set projection heads, we use an MLP. Further details of HYPEBOY, including its implementations and hyperparameters, are provided in Appendix E.4.

---

[2]HyperGRL utilizes a GCN as its backbone encoder since its input is an ordinary graph.

[3]Results for alternative encoders can be found in Appendix F.4.

Table 1: Efficacy as pre-training techniques: AVG and STD of accuracy values in node classification under the **fine-tuning protocol.** The best and second-best performances are colored green and yellow, respectively. R.G. and A.R. denote the random guess and average ranking among all methods, respectively. O.O.T. means that training is not completed within 24 hours. HYPEBOY outperforms all the baseline methods in 8 datasets, and overall, it obtains the best average ranking.

| | Method | Citeseer | Cora | Pubmed | Cora-CA | DBLP-P | DBLP-A | AMiner | IMDB | MN-40 | 20News | House | A.R. |
|---|---|---|---|---|---|---|---|---|---|---|---|---|---|
| (Semi-)Supervised | R.G. | 18.1 (0.9) | 17.4 (1.0) | 36.0 (0.7) | 17.8 (0.7) | 18.9 (0.2) | 25.6 (1.0) | 10.2 (0.2) | 33.9 (0.7) | 3.7 (0.2) | 50.1 (1.3) | 26.7 (0.3) | 17.8 |
| | MLP | 32.5 (7.0) | 27.9 (7.0) | 62.1 (3.7) | 34.8 (5.1) | 73.5 (1.0) | 56.0 (4.9) | 22.3 (1.7) | 39.1 (2.4) | 89.4 (1.5) | 73.1 (1.4) | 72.2 (3.9) | 14.2 |
| | HGNN | 41.9 (7.8) | 50.0 (7.2) | 72.9 (5.0) | 50.2 (5.7) | 85.3 (0.8) | 67.1 (6.0) | 30.3 (2.5) | 42.2 (2.9) | 88.0 (1.4) | 76.4 (1.9) | 52.7 (3.8) | 10.7 |
| | HyperGCN | 31.4 (9.5) | 33.1 (10.2) | 63.5 (14.4) | 37.1 (9.1) | 53.5 (11.6) | 68.2 (14.4) | 26.4 (3.6) | 37.9 (4.5) | 55.1 (7.8) | 67.0 (9.4) | 49.8 (3.5) | 15.6 |
| | HNHN | 43.1 (8.7) | 50.0 (7.9) | 72.1 (5.4) | 48.3 (6.2) | 84.6 (0.9) | 62.6 (4.8) | 30.0 (2.4) | 42.3 (3.4) | 86.1 (1.6) | 74.2 (1.5) | 49.7 (2.2) | 12.6 |
| | UniGCN | 44.2 (8.1) | 49.1 (8.4) | 74.4 (3.9) | 51.3 (6.3) | 86.9 (0.6) | 65.1 (4.7) | 32.7 (1.8) | 41.6 (3.5) | 89.1 (1.0) | 77.2 (1.2) | 51.1 (2.4) | 9.6 |
| | UniGIN | 40.4 (9.1) | 47.8 (7.7) | 69.8 (5.6) | 48.3 (6.1) | 83.4 (0.8) | 63.4 (5.1) | 30.2 (1.4) | 41.4 (2.7) | 88.2 (1.8) | 70.6 (1.8) | 51.1 (3.0) | 13.9 |
| | UniGCNII | 44.2 (9.0) | 48.5 (7.4) | 74.1 (3.9) | 54.8 (7.5) | 87.4 (0.6) | 65.8 (3.9) | 32.5 (1.7) | 42.5 (3.9) | 90.8 (1.1) | 70.9 (1.0) | 50.8 (4.3) | 9.4 |
| | AllSet | 43.5 (8.0) | 47.6 (4.2) | 72.4 (4.5) | 57.5 (5.7) | 85.9 (0.6) | 65.3 (3.9) | 29.3 (1.2) | 42.3 (2.4) | 92.1 (0.6) | 71.9 (2.5) | 54.1 (3.4) | 10.4 |
| | ED-HNN | 40.3 (8.0) | 47.6 (7.7) | 72.7 (4.7) | 54.8 (5.4) | 86.2 (0.8) | 65.8 (4.8) | 30.0 (2.1) | 41.4 (3.0) | 90.7 (0.9) | 76.2 (1.2) | 71.3 (3.7) | 10.4 |
| | PhenomNN | 49.8 (9.6) | 56.4 (9.6) | 76.1 (3.5) | 60.8 (6.2) | 88.1 (0.4) | 72.3 (4.1) | 33.8 (2.0) | 44.1 (3.7) | 95.9 (0.8) | 74.0 (1.5) | 70.4 (5.6) | 3.9 |
| Self-supervised | GraphMAE2 | 41.1 (10.0) | 49.3 (8.3) | 72.9 (4.2) | 55.4 (8.4) | 86.6 (0.6) | 69.5 (4.4) | 32.8 (1.9) | 43.3 (2.7) | 90.1 (0.7) | 71.9 (1.3) | 52.8 (3.5) | 8.9 |
| | MaskGAE | 49.6 (10.1) | 57.1 (8.8) | 72.8 (4.3) | 57.8 (5.9) | 86.3 (0.5) | 74.8 (3.1) | 33.7 (1.6) | 44.5 (2.5) | 90.0 (0.9) | O.O.T. | 51.8 (3.3) | 7.7 |
| | TriCL | 51.7 (9.8) | 60.2 (7.9) | 76.2 (3.6) | 64.3 (5.5) | 88.0 (0.4) | 79.7 (2.9) | 33.1 (2.2) | 46.9 (2.9) | 90.3 (1.0) | 77.2 (1.0) | 69.7 (4.9) | 3.4 |
| | HyperGCL | 47.0 (9.2) | 60.3 (7.4) | 76.8 (3.7) | 62.0 (5.1) | 87.6 (0.5) | 79.7 (3.8) | 33.2 (1.6) | 43.9 (3.6) | 91.2 (0.8) | 77.8 (0.8) | 69.2 (4.9) | 3.5 |
| | H-GD | 45.4 (9.9) | 50.6 (8.2) | 74.5 (3.5) | 58.8 (6.2) | 87.3 (0.5) | 75.1 (3.6) | 32.6 (2.2) | 43.0 (3.3) | 90.0 (1.0) | 77.2 (1.0) | 69.7 (5.1) | 6.1 |
| | HyperGRL | 42.3 (9.3) | 49.1 (8.8) | 73.0 (4.3) | 55.8 (8.0) | 86.7 (0.6) | 70.8 (3.7) | 33.0 (1.8) | 43.1 (2.7) | 90.1 (0.8) | O.O.T. | 52.5 (3.3) | 9.2 |
| | HYPEBOY | 56.7 (9.8) | 62.3 (7.7) | 77.0 (3.4) | 66.3 (4.6) | 88.2 (0.4) | 80.6 (2.3) | 34.1 (2.2) | 47.6 (2.5) | 90.4 (0.9) | 77.6 (0.9) | 70.4 (4.8) | 1.7 |

## 5.1 EFFICACY AS A PRE-TRAINING TECHNIQUE (FINE-TUNED EVALUATION)

**Setup.** Following Wei et al. (2022), we randomly split the nodes into training/validation/test sets with the ratio of 1%/1%/98%, respectively.[4] For reliability, we assess each method on 20 data splits across 5 random model initializations, following Lee & Shin (2023a). We report the average (AVG) and standard deviation (STD) of test accuracy values on each dataset. Specifically, for each SSL strategy, including HYPEBOY, we pre-train a backbone encoder with the corresponding SSL scheme and then fine-tune the encoder in a (semi-)supervised manner.

**Results.** As shown in Table 1, HYPEBOY shows the best average ranking among all 18 methods. Two points stand out. First, pre-training an HNN with HYPEBOY generally improves node classification. Compared to the performance of UniGCNII (HYPEBOY's backbone encoder), HYPEBOY obtains performance gains up to 12.5 points in 10 out of 11 datasets. Second, HYPEBOY outperforms all other SSL strategies. Specifically, compared to the second-best method (TriCL), the accuracy gap is up to 5.0 points. In addition, the suboptimal performance of the state-of-the-art generative SSL strategies for graphs (i.e. GraphMAE2 and MaskGAE) implies the importance of preserving higher-order interactions in learning hypergraph representations. In summary, HYPEBOY serves as an effective SSL strategy to pre-train HNNs for node classification.

## 5.2 EFFICACY AS A GENERAL-PURPOSE EMBEDDING TECHNIQUE (LINEAR EVALUATION)

**Setup.** We assess the generalizability of learned representations from HYPEBOY in two downstream tasks: node classification and hyperedge prediction. Considering this objective, we limit the baseline methods to SSL strategies, which yield embeddings independent of downstream tasks, and the original node features (i.e., naive $\mathbf{X}$). We use the linear evaluation protocol (i.e., the embeddings are used as **fixed** inputs to the classifiers for each task). For node classification, we use the same settings described in Section 5.1. For hyperedge prediction, we split hyperedges into training/validation/test sets by the ratio of 60%/20%/20%. For its evaluation, we obtain the same number of negative hyperedge samples as that of the ground-truth hyperedges. We report the average (AVG) and standard deviation (STD) of test AUROC values on each dataset. Further experimental details about hyperedge prediction, including negative hyperedge sampling, are provided in Appendix E.3.

**Results.** As shown in Table 2, HYPEBOY has the best average ranking in both node classification and hyperedge prediction. Specifically, in node classification, compared to the second-best method (TriCL), the accuracy gap is up to 6.3 points. This demonstrates that HYPEBOY is more effective in learning general-purpose hypergraph representations than the other SSL strategies on hypergraphs.

---

[4]We ensure that a training set includes at least one node from each class.

Table 2: Efficacy as general-purpose embedding techniques: AVG and STD of accuracy/AUROC values in node classification and hyperedge prediction tasks under the **linear evaluation protocol**. In each downstream task, the best and second-best performances are colored green and yellow, respectively. A.R. denotes the average ranking among all methods. O.O.T. means that training is not completed within 24 hours. HYPEBOY obtains the best average ranking in both downstream tasks.

| | Method | Citeseer | Cora | Pubmed | Cora-CA | DBLP-P | DBLP-A | AMiner | IMDB | MN-40 | 20News | House | A.R. |
|---|---|---|---|---|---|---|---|---|---|---|---|---|---|
| Node classification | Naive $\mathbf{X}$ | 27.8 (7.0) | 32.4 (4.6) | 62.8 (2.8) | 31.9 (5.5) | 69.4 (0.7) | 54.7 (4.7) | 21.4 (1.2) | 38.1 (1.9) | 91.9 (1.1) | 70.6 (1.9) | 71.3 (5.4) | 5.5 |
| | GraphMAE2 | 29.2 (6.5) | 37.5 (7.0) | 55.5 (9.5) | 38.2 (9.1) | 75.6 (1.7) | 57.5 (5.6) | 27.3 (2.7) | 36.6 (3.5) | 89.1 (1.8) | 62.3 (2.3) | 51.7 (3.5) | 6.3 |
| | MaskGAE | 47.2 (11.1) | 56.8 (9.3) | 62.6 (5.5) | 56.0 (4.8) | 84.8 (0.7) | 75.1 (3.5) | 33.2 (2.0) | 44.1 (3.9) | 90.5 (0.9) | O.O.T. | 50.0 (2.8) | 4.5 |
| | TriCL | 53.3 (10.0) | 62.1 (8.8) | 74.5 (4.1) | 63.6 (5.2) | 87.1 (0.7) | 80.9 (3.2) | 35.0 (3.6) | 48.0 (3.2) | 80.0 (5.1) | 67.2 (4.0) | 69.1 (5.5) | 2.6 |
| | HyperGCL | 42.6 (8.6) | 61.8 (8.3) | 67.6 (8.0) | 58.1 (6.3) | 56.6 (5.2) | 79.8 (3.8) | 33.3 (2.2) | 47.5 (2.8) | 84.1 (2.8) | 71.2 (3.4) | 67.1 (5.4) | 4 |
| | H-GD | 35.6 (7.8) | 37.6 (6.8) | 58.0 (8.2) | 48.6 (7.4) | 73.3 (1.3) | 74.0 (3.3) | 33.8 (5.0) | 35.2 (2.9) | 76.6 (4.4) | 54.8 (7.4) | 68.3 (5.7) | 5.5 |
| | HyperGRL | 35.3 (8.2) | 35.4 (8.8) | 50.2 (8.7) | 39.4 (8.1) | 78.7 (1.2) | 62.7 (5.1) | 28.0 (2.8) | 34.8 (3.0) | 89.4 (1.5) | O.O.T. | 52.0 (3.7) | 6.1 |
| | HYPEBOY | 59.6 (9.9) | 63.5 (9.4) | 75.0 (3.4) | 66.0 (4.6) | 87.9 (0.5) | 81.2 (2.7) | 34.3 (3.2) | 48.8 (1.8) | 89.2 (2.2) | 75.7 (2.1) | 69.4 (5.4) | 1.5 |
| Hyperedge prediction | Naive $\mathbf{X}$ | 63.3 (2.1) | 75.5 (1.6) | 88.3 (0.6) | 55.0 (1.9) | 90.0 (0.4) | 72.1 (1.3) | 80.0 (1.1) | 39.5 (1.9) | 99.5 (0.1) | 97.7 (2.9) | 54.8 (5.0) | 6.4 |
| | GraphMAE2 | 73.3 (2.7) | 76.4 (1.7) | 81.6 (1.1) | 76.3 (3.1) | 85.2 (0.4) | 68.3 (1.8) | 80.7 (0.9) | 53.7 (2.6) | 99.5 (0.1) | 90.1 (5.7) | 62.9 (3.8) | 6.1 |
| | MaskGAE | 86.1 (1.6) | 88.5 (1.4) | 92.9 (0.5) | 81.8 (2.7) | 93.2 (0.5) | 79.3 (2.0) | 84.6 (0.1) | 58.1 (2.5) | 99.3 (0.1) | O.O.T. | 87.0 (3.4) | 4.1 |
| | TriCL | 90.5 (1.2) | 90.7 (1.3) | 91.9 (0.5) | 87.8 (1.5) | 94.8 (0.2) | 87.9 (1.4) | 90.4 (0.6) | 58.9 (2.1) | 99.6 (0.4) | 98.2 (3.0) | 90.0 (2.6) | 1.8 |
| | HyperGCL | 73.9 (2.6) | 85.4 (1.5) | 89.6 (0.5) | 81.1 (1.9) | 83.6 (0.6) | 83.5 (1.0) | 82.1 (7.6) | 53.8 (2.4) | 99.4 (0.1) | 96.7 (7.0) | 76.3 (6.3) | 5 |
| | H-GD | 72.2 (5.0) | 71.9 (3.1) | 87.2 (0.7) | 73.2 (4.0) | 91.6 (1.0) | 81.4 (1.9) | 84.9 (2.1) | 53.1 (1.8) | 99.5 (0.1) | 83.9 (2.1) | 87.9 (3.1) | 5.3 |
| | HyperGRL | 83.2 (8.0) | 84.0 (1.7) | 78.3 (1.8) | 80.0 (3.3) | 88.2 (0.3) | 80.8 (1.4) | 81.5 (0.7) | 54.7 (2.8) | 98.8 (0.4) | O.O.T. | 88.2 (3.3) | 5.5 |
| | HYPEBOY | 91.1 (1.1) | 91.9 (1.1) | 95.1 (0.3) | 88.1 (1.4) | 95.5 (0.1) | 87.3 (1.3) | 89.8 (0.5) | 59.4 (2.1) | 99.7 (0.1) | 99.0 (1.6) | 87.0 (2.8) | 1.5 |

Table 3: The ablation study with four variants of HYPEBOY on node classification under the fine-tuning protocol. The best and second-best performances are colored green and yellow, respectively. F.R., H.F., and P.H. denote Feature Reconstruction, Hyperedge Filling, and Projection Heads, respectively. A.R. denotes the average ranking among all methods. NA denotes no pre-training. HYPEBOY outperforms others in most datasets, justifying each of its components.

| | F. R. | H. F. | P. H. | Citeseer | Cora | Pubmed | Cora-CA | DBLP-P | DBLP-A | AMiner | IMDB | MN-40 | 20News | House | A.R. |
|---|---|---|---|---|---|---|---|---|---|---|---|---|---|---|---|
| NA | ✗ | ✗ | ✗ | 44.2 (9.0) | 48.5 (7.4) | 74.1 (3.9) | 54.8 (7.5) | 87.4 (0.6) | 65.8 (3.9) | 32.5 (1.7) | 42.5 (3.9) | 90.8 (1.1) | 70.9 (1.0) | 50.8 (4.3) | 5.5 |
| V1 | ✗ | ✔ | ✗ | 51.6 (11.2) | 60.7 (8.2) | 76.2 (3.6) | 63.5 (6.0) | 88.1 (0.5) | 78.5 (2.9) | 33.5 (2.8) | 46.8 (3.1) | 90.0 (1.1) | 77.4 (0.9) | 68.5 (4.5) | 4.3 |
| V2 | ✗ | ✔ | ✔ | 52.7 (9.6) | 59.7 (9.2) | 76.7 (3.2) | 63.5 (6.0) | 88.2 (0.5) | 79.1 (2.5) | 33.8 (2.2) | 46.9 (3.3) | 90.6 (1.0) | 77.0 (0.9) | 69.6 (4.9) | 3.3 |
| V3 | ✔ | ✗ | ✗ | 52.0 (9.3) | 58.9 (8.2) | 74.1 (3.9) | 61.2 (6.6) | 87.8 (0.4) | 79.9 (2.3) | 33.9 (2.1) | 46.3 (2.7) | 91.4 (0.9) | 77.5 (0.9) | 70.1 (4.8) | 3.6 |
| V4 | ✔ | ✔ | ✗ | 56.0 (9.9) | 61.8 (8.5) | 76.5 (3.1) | 65.3 (4.3) | 88.0 (0.4) | 80.3 (2.4) | 34.0 (2.0) | 47.5 (2.3) | 90.8 (1.0) | 77.4 (1.0) | 69.3 (5.0) | 2.5 |
| Ours | ✔ | ✔ | ✔ | 56.7 (9.8) | 62.3 (7.7) | 77.0 (3.4) | 66.3 (4.6) | 88.2 (0.4) | 80.6 (2.3) | 34.1 (2.2) | 47.6 (2.5) | 90.4 (0.9) | 77.6 (0.9) | 70.4 (4.8) | 1.4 |

## 5.3 ABLATION STUDY

We analyze the necessity for each component of HYPEBOY, specifically, (a) the hyperedge filling task (Section 3.1), (b) projection heads (Section 4.2), and (c) feature reconstruction warm-up (Section 4.4). To this end, we utilize four variants of HYPEBOY: **(V1)** without feature reconstruction warm-up and projection heads, **(V2)** without feature reconstruction warm-up [5], **(V3)** without the hyperedge filling process, and **(V4)** without projection heads. Here, projection heads are used only for methods with the hyperedge filling process. Note that **V3** is a method that only utilizes the feature reconstruction process, which is described in Section 4.4.

As shown in Table 3, HYPEBOY, equipped with all of its components, outperforms the others in most datasets, demonstrating the effectiveness of our design choices. There are two other notable results. First, the necessity of projection heads is evidenced by the superior performance of **V2** (compared to **V1**) and ours (compared to **V4**). Second, the advantage of the hyperedge filling task over feature reconstruction is manifested by the better average rank of **V2** compared to **V3**.

## 6 CONCLUSION

In this work, we conduct a comprehensive analysis of generative self-supervised learning on hypergraphs. Our contribution is three-fold. First, we formulate the hyperedge filling task, a generative self-supervised learning task on hypergraphs, and investigate the theoretical connection between the task and node classification (Section 3). Second, we present a generative SSL method HYPEBOY to solve the task (Section 4). Third, we demonstrate the superiority of HYPEBOY over existing SSL methods on hypergraphs through extensive experiments (Section 5).

---

[5]To mitigate an issue of over-relying on projection heads (Section 4.4), we have trained an encoder without projection heads at the beginning, and after some epochs, we train the encoder with projection heads.

ACKNOWLEDGEMENTS

This work was supported by Samsung Electronics Co., Ltd. and Institute of Information & Communications Technology Planning & Evaluation (IITP) grant funded by the Korea government (MSIT) (No. 2022-0-00871, Development of AI Autonomy and Knowledge Enhancement for AI Agent Collaboration) (No. 2019-0-00075, Artificial Intelligence Graduate School Program (KAIST)).

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

# A  APPENDIX

In this section, we provide a proof of each theorem. In addition, we extend the results from Section 3.2, which assume $v_i \in C_1$ is assumed, to include cases where $v_i \in C_0$.

## A.1  PROOF OF THEOREM 1

*Proof.* We first analyze a node $v_i$ that belongs to $C_1$ (i.e., $v_i \in C_1$). By definition, $\mathbb{E}_{\boldsymbol{x}}[\mathbf{1}_{\mathcal{F}(\boldsymbol{x}_i)=1}] = P_{\boldsymbol{x}}\left(f(\boldsymbol{x}_i; \boldsymbol{\mu}_1, \mathbf{I}) > f(\boldsymbol{x}_i; \boldsymbol{\mu}_0, \mathbf{I})\right)$ holds. Thus, $\mathbb{E}_{\boldsymbol{x}}[\mathbf{1}_{\mathcal{F}(\boldsymbol{x}_i)=1}]$ is determined by the distribution of $f(\boldsymbol{x}_i; \boldsymbol{\mu}_1, \mathbf{I}) > f(\boldsymbol{x}_i; \boldsymbol{\mu}_0, \mathbf{I})$, where a random variable is $\boldsymbol{x}_i$. We provide the exact formula for this inequality:

$$f(\boldsymbol{x}_i; \boldsymbol{\mu}_1, \mathbf{I}) > f(\boldsymbol{x}_i; \boldsymbol{\mu}_0, \mathbf{I}),$$
$$\equiv \exp(-(\boldsymbol{x}_i - \boldsymbol{\mu}_1)^T(\boldsymbol{x}_i - \boldsymbol{\mu}_1)) > \exp(-(\boldsymbol{x}_i - \boldsymbol{\mu}_0)^T(\boldsymbol{x}_i - \boldsymbol{\mu}_0)),$$
$$\equiv (\boldsymbol{x}_i - \boldsymbol{\mu}_1)^T(\boldsymbol{x}_i - \boldsymbol{\mu}_1) < (\boldsymbol{x}_i - \boldsymbol{\mu}_0)^T(\boldsymbol{x}_i - \boldsymbol{\mu}_0),$$
$$= \boldsymbol{x}_i^T\boldsymbol{x}_i - 2\boldsymbol{\mu}_1^T\boldsymbol{x}_i + \boldsymbol{\mu}_1^T\boldsymbol{\mu}_1 < \boldsymbol{x}_i^T\boldsymbol{x}_i - 2\boldsymbol{\mu}_0^T\boldsymbol{x}_i + \boldsymbol{\mu}_0^T\boldsymbol{\mu}_0,$$
$$\equiv (\boldsymbol{\mu}_1 - \boldsymbol{\mu}_0)^T\boldsymbol{x}_i + \boldsymbol{\mu}_0^T\boldsymbol{\mu}_0 - \boldsymbol{\mu}_1^T\boldsymbol{\mu}_1 > 0 \equiv \vec{\mathbf{1}}^T\boldsymbol{x}_i > 0, \quad \because \text{ definition of } \boldsymbol{\mu}_1 \text{ and } \boldsymbol{\mu}_0 \text{ in Assumption 1.}$$

Thus, $\mathbb{E}_{\boldsymbol{x}}[\mathbf{1}_{\mathcal{F}(\boldsymbol{x}_i)=1}]$ is equivalent to $P_{\boldsymbol{x}}(\vec{\mathbf{1}}^T\boldsymbol{x}_i > 0)$. In a similar sense, $\mathbb{E}_{\boldsymbol{x}}[\mathbf{1}_{\mathcal{F}(\boldsymbol{z}_i)=1}]$ is equivalent to $P_{\boldsymbol{x}}(\vec{\mathbf{1}}^T\boldsymbol{z}_i > 0)$, since as mentioned in Section 3.2, $\boldsymbol{z}_i$ is a function of $\boldsymbol{x}_k, \forall v_k \in \mathcal{V}$.

Now, we formalize $\boldsymbol{z}_i$. Note that $\boldsymbol{z}_i = \boldsymbol{x}_i - \gamma \nabla_{\boldsymbol{x}_i}\mathcal{L}$ holds by (**F2**). To further elaborate on this equation, we first detail $\nabla_{\boldsymbol{x}_i}\mathcal{L}$:

$$\nabla_{\boldsymbol{x}_i}\mathcal{L} = \frac{\partial\left(-\log(p_{(\mathbf{X},\mathcal{E},\Theta)}\left(v_i \mid q_{ij}\right))\right)}{\partial\boldsymbol{x}_i} = \frac{\partial\left(-\log\left(\frac{\exp(\boldsymbol{x}_i^T(\sum_{v_k \in q_{ij}}\boldsymbol{x}_k))}{\sum_{v_t \in \mathcal{V}}\exp(\boldsymbol{x}_t^T(\sum_{v_k \in q_{ij}}\boldsymbol{x}_k))}\right)\right)}{\partial\boldsymbol{x}_i}, \quad (6)$$

$$= -\frac{\partial\left(\boldsymbol{x}_i^T(\sum_{v_k \in q_{ij}}\boldsymbol{x}_k)\right)}{\partial\boldsymbol{x}_i} + \frac{\partial\left(\log\left(\sum_{v_t \in \mathcal{V}}\exp(\boldsymbol{x}_t^T(\sum_{v_k \in q_{ij}}\boldsymbol{x}_k))\right)\right)}{\partial\boldsymbol{x}_i}, \quad (7)$$

$$= -\left(\sum_{v_k \in q_{ij}}\boldsymbol{x}_k\right) + \frac{\exp(\boldsymbol{x}_i^T(\sum_{v_k \in q_{ij}}\boldsymbol{x}_k))}{\sum_{v_t \in \mathcal{V}}\exp(\boldsymbol{x}_t^T(\sum_{v_k \in q_{ij}}\boldsymbol{x}_k))}\left(\sum_{v_k \in q_{ij}}\boldsymbol{x}_k\right), \quad (8)$$

$$= -\left(1 - \frac{\exp(\boldsymbol{x}_i^T(\sum_{v_k \in q_{ij}}\boldsymbol{x}_k))}{\sum_{v_t \in \mathcal{V}}\exp(\boldsymbol{x}_t^T(\sum_{v_k \in q_{ij}}\boldsymbol{x}_k))}\right)\left(\sum_{v_k \in q_{ij}}\boldsymbol{x}_k\right). \quad (9)$$

By Eq (9), $\boldsymbol{z}_i$ can be expressed as a function of $\boldsymbol{x}_k, \forall v_k \in \mathcal{V}$, as follows:

$$\boldsymbol{z}_i = \boldsymbol{x}_i + \gamma \underbrace{\left(1 - \frac{\exp(\boldsymbol{x}_i^T(\sum_{v_k \in q}\boldsymbol{x}_k))}{\sum_{v_t \in \mathcal{V}}\exp(\boldsymbol{x}_t^T(\sum_{v_k \in q}\boldsymbol{x}_k))}\right)}_{\text{(Term 1)}}\left(\sum_{v_k \in q}\boldsymbol{x}_k\right). \quad (10)$$

Let $\boldsymbol{x}_q'$ denotes $\sum_{v_k \in q_{ij}}\boldsymbol{x}_k$. Furthermore, denote (Term 1) in Eq (10) as $f(\boldsymbol{x}) \in (0, 1)$. Then, we rewrite Eq (10) as $\boldsymbol{z}_i = \boldsymbol{x}_i + \gamma\left(1 - f(\boldsymbol{x})\right)\boldsymbol{x}_q'$. We finally rewrite $\vec{\mathbf{1}}^T\boldsymbol{z}_i$ as follows:

$$\vec{\mathbf{1}}^T\boldsymbol{z}_i = \vec{\mathbf{1}}^T\boldsymbol{x}_i + \gamma(1 - f(\boldsymbol{x}))\vec{\mathbf{1}}^T\boldsymbol{x}_q'. \quad (11)$$

Let $\beta$ denotes $\vec{\mathbf{1}}^T\boldsymbol{x}_q'$. Note that by the statement of Theorem 1, $\vec{\mathbf{1}}^T\boldsymbol{x}_q' > 0 \equiv \beta > 0$ holds. Thus, our main interest, which is $P_{\boldsymbol{x}}(\vec{\mathbf{1}}^T\boldsymbol{z}_i > 0)$, can be rewritten as follows:

$$P_{\boldsymbol{x}}\left(\vec{\mathbf{1}}^T\boldsymbol{z}_i > 0\right) = P_{\boldsymbol{x}}\left(\left(\vec{\mathbf{1}}^T\boldsymbol{x}_i + \gamma(1 - f(\boldsymbol{x}))\beta\right) > 0\right). \quad (12)$$

Note that when $\vec{\mathbf{1}}^T \boldsymbol{x}_i > 0$ holds, $\vec{\mathbf{1}}^T \boldsymbol{z}_i > 0$ holds as well, since $\gamma(1 - f(\boldsymbol{x}))\beta > 0$ holds. Thus, Eq (12) is split as follows:

$$P_{\boldsymbol{x}}\left(\vec{\mathbf{1}}^T \boldsymbol{z}_i > 0\right) = P_{\boldsymbol{x}}\left(\vec{\mathbf{1}}^T \boldsymbol{x}_i > 0\right) + P_{\boldsymbol{x}}\left(-\gamma(1 - f(\boldsymbol{x}))\beta < \vec{\mathbf{1}}^T \boldsymbol{x}_i < 0\right),$$

$$= \mathbb{E}_{\boldsymbol{x}}[\mathbf{1}_{\mathcal{F}(\boldsymbol{z}_i)=1}] = \underbrace{\mathbb{E}_{\boldsymbol{x}}[\mathbf{1}_{\mathcal{F}(\boldsymbol{x}_i)=1}]}_{\text{(a) Expected accuracy of naive } \boldsymbol{x}_i} + \underbrace{P_{\boldsymbol{x}}\left(-\gamma\beta < \frac{\vec{\mathbf{1}}^T \boldsymbol{x}_i}{(1 - f(\boldsymbol{x}))} < 0\right)}_{\text{(b) Additional gain via hyperedge filling}}. \quad (13)$$

Note that the gain term, which is the (b) term of Eq (13), is always greater than zero.

**Generalization to $v_i \in C_0$.** Now, we analyze a node $v_i$ that belongs to $C_0$ (i.e., $v_i \in C_0$). In this case, the previous condition $\vec{\mathbf{1}}^T \boldsymbol{x}'_q > 0 \equiv \beta > 0$ becomes $\vec{\mathbf{1}}^T \boldsymbol{x}'_q < 0 \equiv \beta < 0$. In a similar sense, for the expected accuracy: $P_{\boldsymbol{x}}\left(\vec{\mathbf{1}}^T \boldsymbol{z}_i > 0\right)$ is changed as $P_{\boldsymbol{x}}\left(\vec{\mathbf{1}}^T \boldsymbol{z}_i < 0\right)$. In this setting, we can directly extend the result of Eq (12) as follows:

$$P_{\boldsymbol{x}}\left(\vec{\mathbf{1}}^T \boldsymbol{z}_i < 0\right) = P_{\boldsymbol{x}}\left(\left(\vec{\mathbf{1}}^T \boldsymbol{x}_i + \gamma(1 - f(\boldsymbol{x}))\beta\right) < 0\right). \quad (14)$$

By employing the above proof, we can obtain the following result (note that $\beta < 0$ in this case):

$$P_{\boldsymbol{x}}\left(\vec{\mathbf{1}}^T \boldsymbol{z}_i < 0\right) = P_{\boldsymbol{x}}\left(\vec{\mathbf{1}}^T \boldsymbol{x}_i < 0\right) + P_{\boldsymbol{x}}\left(0 < \vec{\mathbf{1}}^T \boldsymbol{x}_i < -\gamma(1 - f(\boldsymbol{x}))\beta\right),$$

$$= \mathbb{E}_{\boldsymbol{x}}[\mathbf{1}_{\mathcal{F}(\boldsymbol{z}_i)=0}] = \underbrace{\mathbb{E}_{\boldsymbol{x}}[\mathbf{1}_{\mathcal{F}(\boldsymbol{x}_i)=0}]}_{\text{(a) Expected accuracy of naive } \boldsymbol{x}_i} + \underbrace{P_{\boldsymbol{x}}\left(-\gamma\beta < \frac{\vec{\mathbf{1}}^T \boldsymbol{x}_i}{(1 - f(\boldsymbol{x}))} < 0\right)}_{\text{(b) Additional gain via hyperedge filling}}. \quad (15)$$

Thus, we can also derive the same result for $v_i \in C_0$. $\qquad\square$

### A.2 PROOF OF THEOREM 2.

**Remark.** *We have denoted the affinity parameter in Assumption 2 as $\mathcal{P}$ to distinguish it from the hyperedge filling probability $p_{(\mathbf{X},\mathcal{E},\Theta)}$. In this proof, by allowing a slight duplication in notation, we let $\mathcal{P} := p$, since we do not use $p_{(\mathbf{X},\mathcal{E},\Theta)}$ in this proof. In addition, we let $e_j := e$ and $q_{ij} := q$.*

*Proof.* We first derive the functional form of $P_{\boldsymbol{x},e}\left(\vec{\mathbf{1}}^T\left(\sum_{v_k \in q} \boldsymbol{x}_k\right) > 0 \,\Big|\, p\right)$. By $v_i \in e_j \cap C_1$, which is the condition of the theorem, the following holds:

$$P_{\boldsymbol{x},e}\left(\vec{\mathbf{1}}^T\left(\sum_{v_k \in q} \boldsymbol{x}_k\right) > 0 \,\Big|\, p\right) = \frac{P_{\boldsymbol{x},e}\left(\vec{\mathbf{1}}^T\left(\sum_{v_k \in q} \boldsymbol{x}_k\right) > 0, v_i \in e \,\Big|\, p\right)}{P_e\left(v_i \in e \mid p\right)}. \quad (16)$$

It is important to note that if the number of nodes that belong to both $C_1$ and $e_j$ is decided, denoted by $s$ (i.e., $s := |e \cap C_1|$), the distribution of $\vec{\mathbf{1}}^T\left(\sum_{v_k \in q} \boldsymbol{x}_k\right)$ is automatically decided, since $\boldsymbol{x}_t, \forall v_t \in \mathcal{V}$ is independently generated from Gaussian distribution, whose mean vector is decided according to the class of $v_t$. This is because of $\sum_{v_k \in q} \boldsymbol{x}_k \sim \mathcal{N}((2s - 1 - S)\boldsymbol{\mu}_1, (S - 1)\mathbf{I})$, where $S$ is a size of $e$ for a given $s$ (i.e., $|e| = S$). From this result, the following two results are induced:

$$\vec{\mathbf{1}}^T\left(\sum_{v_k \in q} \boldsymbol{x}_k\right) \sim \mathcal{N}\left(\frac{(2s - 1 - S)d}{2}, (S - 1)d\right), \quad (17)$$

$$P_{\boldsymbol{x}}\left(\vec{\mathbf{1}}^T\left(\sum_{v_k \in q} \boldsymbol{x}_k\right) > 0 \,\Big|\, s\right) = \Phi\left((2s - S - 1)\sqrt{\frac{d}{4(S - 1)}}\right). \quad (18)$$

where $\Phi(\cdot)$ is a C.D.F. of the standard Gaussian distribution. Then, we rewrite Eq (16) as follows:

$$\equiv \sum_{s=0}^{S} \frac{P_{\boldsymbol{x},e}\left(\vec{\mathbf{1}}^T \left(\sum_{v_k \in q} \boldsymbol{x}_k\right) > 0, s, v_i \in e\right)}{P_e(v_i \in e \mid p)}, \tag{19}$$

$$= \sum_{s=0}^{S} \frac{P_{\boldsymbol{x}}\left(\vec{\mathbf{1}}^T \left(\sum_{v_k \in q} \boldsymbol{x}_k\right) > 0 \mid s\right) \times P_e(s, v_i \in e)}{P_e(v_i \in e \mid p)}. \tag{20}$$

By Assumption 2, each $P_e(s, v_i \in e)$ is derived as follows [6]:

$$P_e(s, v_i \in e \mid p) = \binom{S}{s} \left(\underbrace{\frac{p^s(1-p)^{S-s}}{2}}_{\text{prob. of } s \text{ at } c=1} + \underbrace{\frac{(1-p)^s p^{S-s}}{2}}_{\text{prob. of } s \text{ at } c=0}\right) \underbrace{\left(1 - \frac{\binom{N-1}{s}}{\binom{N}{s}}\right)}_{\text{prob. of } v_i \in C_1 \cap e}. \tag{21}$$

By using Eq (21), we further detail $P_e(v_i \in e \mid p)$ as follows:

$$P_e(v_i \in e \mid p) = \sum_{s=0}^{S} \binom{S}{s} \left(\frac{p^s(1-p)^{S-s}}{2} + \frac{(1-p)^s p^{S-s}}{2}\right) \left(1 - \frac{\binom{N-1}{s}}{\binom{N}{s}}\right), \tag{22}$$

$$= \sum_{s=0}^{S} \binom{S}{s} \left(\frac{p^s(1-p)^{S-s}}{2} + \frac{(1-p)^s p^{S-s}}{2}\right) \frac{s}{N}, \tag{23}$$

$$= \frac{1}{N} \sum_{s=0}^{S} \binom{S}{s} \left(\underbrace{\frac{sp^s(1-p)^{S-s}}{2}}_{s \sim B(S,p)} + \underbrace{\frac{s(1-p)^s p^{S-s}}{2}}_{s \sim B(S,1-p)}\right), \tag{24}$$

$$= \frac{1}{N} \left(\frac{Sp}{2} + \frac{S(1-p)}{2}\right), \because \text{ expectation of Binomial distribution,} \tag{25}$$

$$= \frac{S}{2N}. \tag{26}$$

With the result of Eq (26) and Eq (18), we rewrite Eq (20) as follows:

$$\equiv \frac{2N}{S} \sum_{s=0}^{S} \binom{S}{s} \left(\frac{p^s(1-p)^{S-s}}{2} + \frac{(1-p)^s p^{S-s}}{2}\right) \frac{s}{N} \Phi\left((2s-1-S)\sqrt{\frac{d}{4(S-1)}}\right), \tag{27}$$

$$= \frac{1}{S} \sum_{s=0}^{S} \binom{S}{s} s \left(p^s(1-p)^{S-s} + (1-p)^s p^{S-s}\right) \Phi\left((2s-1-S)\sqrt{\frac{d}{4(S-1))}}\right). \tag{28}$$

Note that our main function, which is $P_{\boldsymbol{x},e}\left(\vec{\mathbf{1}}^T \left(\sum_{v_k \in q} \boldsymbol{x}_k\right) > 0 \mid p\right)$, is equal to Eq (28).

It is important to note that Eq (28) is symmetric w.r.t. 0.5 in $p \in [0, 1]$. Thus, if we prove Eq (28) is strictly increasing on $p \in [0.5, 1]$, the proof of the statement that Eq (28) is strictly decreasing on $p \in [0, 0.5]$ is straight forward. Hence, we prove the second statement of the theorem by showing that Eq (28) is strictly increasing on $p \in [0.5, 1]$.

Now, we show the first and second statements of the theorem. Here, we first show the second statement, and from the result, we further prove the first statement.

**Second statement.** We first rewrite Eq (28) by adequately mixing two binomial functions. For simplicity, we denote $\Phi((2s-1-S)\sqrt{\frac{d}{4(S-1)}})$ as $\phi(s)$, since the value that changes in $\Phi(\cdot)$ is only

---

[6]The term *prob.* indicates probability.

$s$ and other terms are fixed constants. From this, Eq (28) is rewritten as follows:

$$= \frac{1}{S} \sum_{s=0}^{S} \binom{S}{s} s \left( p^s(1-p)^{S-s} + (1-p)^s p^{S-s} \right) \phi(s), \tag{29}$$

$$= \frac{1}{S} \underbrace{\sum_{s=0}^{S} \binom{S}{s} s p^s(1-p)^{S-s} \phi(s)}_{\text{(Term 1)}} + \frac{1}{S} \underbrace{\sum_{s=0}^{S} \binom{S}{s} s(1-p)^s (p)^{S-s} \phi(s)}_{\text{(Term 2)}}. \tag{30}$$

We define $s' := (S-s)$. Due to the symmetric characteristic of the binomial function and standard Gaussian distribution, we can rewrite (Term 2) in Eq (30) as follows:

$$\equiv \frac{1}{S} \sum_{s'=0}^{S} \binom{S}{s'} p^{s'}(1-p)^{S-s'}(S-s') \Phi\left( (S-2s'-1) \sqrt{\frac{d}{4(S-1)}} \right). \tag{31}$$

Since we can regard $s$ and $s'$ as equivalent terms in our framework, we add (Term 1) of Eq (30) and Eq (31) as follows:

$$\sum_{s=0}^{S} \binom{S}{s} \frac{p^s(1-p)^{S-s}}{S} \underbrace{\left[ (S-s)\Phi\left( (S-2s-1)\sqrt{\frac{d}{4(S-1)}} \right) + s\Phi\left( (2s-S-1)\sqrt{\frac{d}{4(S-1)}} \right) \right]}_{\text{Term (a)}}. \tag{32}$$

We aim to show that Eq (32) is a strictly increasing function w.r.t. $p \in [0.5, 1]$. For simplicity, we let $K := \sqrt{d/(4(S-1))}$. We first discard a constant term $1/S$ from Eq (32) for simplicity. Note that without Eq (32) Term (a), Eq (32) is equivalent to 1, and this is a sum of binomial probability mass functions.

That is, we can think of Eq (32) as a weighted average of the following values:

$$(S-s)\Phi\left((S-2s-1)K\right) + s\Phi\left((2s-S-1)K\right), \forall s \in \{0, 1, \cdots, S\}. \tag{33}$$

Specifically, we let $\binom{S}{s} p^s(1-p)^{S-s}$ as a weight of $\left[ (S-s)\Phi\left((S-2s-1)K\right) + s\Phi\left((2s-S-1)K\right) \right]$. Note that values of Eq (33) and their weights are both symmetric about $s = S/2$: which means $s$ and $S-s$ have the same weight value. Thus, we can rewrite Eq (32) for an odd number $S$ as:

$$\equiv \sum_{s=\lfloor S/2 \rfloor}^{S} \binom{S}{s} (p^s(1-p)^{S-s} + (1-p)^s p^{S-s})$$
$$\times \left[ (S-s)\Phi\left((S-2s-1)K\right) + s\Phi\left((2s-S-1)K\right) \right]. \tag{34}$$

While this formulation is for an odd number $S$, we can extend further results to an even number $S$.

We first show that Eq (33) is an increasing function w.r.t. $s \in \{\lfloor S/2 \rfloor, \lfloor S/2 \rfloor + 1, \cdots, S\}$. For two cases where $s = k$ and $s = k+1$, their Eq (33) is expressed as follows:

$$\left[ (S-k-1)\Phi\left(S-2k-3\right) + (k+1)\Phi\left(2k-S+1\right) \right] \tag{35}$$
$$> \left[ (S-k)\Phi\left(S-2k-1\right) + k\Phi\left(2k-S-1\right) \right], \tag{36}$$
$$\equiv k\left(\Phi\left(2k-S+1\right) - \Phi\left(2k-S-1\right)\right) + \Phi\left(2k-S+1\right) \tag{37}$$
$$> (S-k)\left(\Phi\left(S-2k-1\right) - \Phi\left(S-2k-3\right)\right) + \Phi\left(S-2k-3\right), \tag{38}$$

By the condition of $s \geq \lfloor S/2 \rfloor$, $k > S - k$ holds. In addition, by the characteristic of the CDF of the standard normal distribution, $(\Phi\left(2k-S+1\right) - \Phi\left(2k-S-1\right)) > (\Phi\left(S-2k-1\right) - \Phi\left(S-2k-3\right))$ also holds. Moreover, since $s \geq \lfloor S/2 \rfloor$, $\Phi\left(2k-S+1\right) > \Phi\left(S-2k-3\right)$ also holds. In sum, Eq (35) > Eq (36) holds, and this result implies that Eq (33) is an increasing function w.r.t. $s \in \{\lfloor S/2 \rfloor, \lfloor S/2 \rfloor + 1, \cdots, S\}$.

Since greater $s$ has a higher value of Eq (33), we can naturally conclude that Eq (34), which is our goal function, is an increasing function w.r.t. $p$ if weights are more assigned to the terms

related to the higher $s$ as $p$ increases. From this rationale, we show the following statement: $\exists s^* \in \{\lfloor S/2 \rfloor, \cdots, S\}$, s.t. $\partial w(p,s)/\partial p > 0, \forall s \geq s^*$ and $\partial w(p,s)/\partial p < 0, \forall s \leq s^*$, where $w(p,s) = \binom{S}{s}\left(p^s(1-p)^{S-s} + (1-p)^s p^{S-s}\right)$. We first derive the first derivative of $w(p,s)$:

$$\frac{\partial \omega(p,s)}{\partial p} = \binom{S}{s}\Big(\underbrace{sp^{s-1}(1-p)^{S-s}}_{(a)} - \underbrace{(S-s)p^s(1-p)^{S-s-1}}_{(b)} \tag{39}$$

$$-\underbrace{s(1-p)^{s-1}p^{S-s}}_{(c)} + \underbrace{(S-s)(1-p)^s p^{S-s-1}}_{(d)}\Big). \tag{40}$$

Then, we show that: $\exists s^* \in \{\lfloor S/2 \rfloor, \cdots, S\}$, s.t. $((a)-(b)) > ((c)-(d)), \forall s \geq s^*$ and $((a)-(b)) < ((c)-(d)), \forall s \leq s^*$. We elaborate on $(a)-(b)$ and $(c)-(d)$ as follows:

$$(a)-(b) = p^{s-1}(1-p)^{S-s-1}(s(1-p)-(S-s)p), \tag{41}$$

$$= p^{s-1}(1-p)^{S-s-1}(s-sp-Sp+sp), \tag{42}$$

$$(c)-(d) = (1-p)^{s-1}p^{S-s-1}(-sp+(S-s)(1-p)), \tag{43}$$

$$= (1-p)^{s-1}p^{S-s-1}(-sp+(S-s)-Sp+sp). \tag{44}$$

**Remark.** *When $p = 1/2, ((a)-(b)) - ((c)-(d)) = 0$ holds, which indicates that if Eq (32) turns out to be an increasing function w.r.t. $p$, Eq (32) has its minimum value at $p = 1/2$. Thus, we narrow down the range of $p$ as $p > 1/2$.*

We first formalize $(a)-(b) = (c)-(d)$ equation. To this end, we utilize the logarithm function to simplify the computation (note that $\binom{S}{s}$ is omitted).

$$\equiv p^{s-1}(1-p)^{S-s-1}(s-Sp) = (1-p)^{s-1}p^{S-s-1}(s-S+Sp), \tag{45}$$

$$\equiv (s-1)\log p + (S-s-1)\log(1-p) + \log(s-Sp),$$

$$= (s-1)\log(1-p) + (S-s-1)\log p + \log(s-S+Sp),$$

$$= (s-1)\log\left(\frac{p}{1-p}\right) + (S-s-1)\log\frac{1-p}{p} + \log\frac{s-Sp}{s-S(1-p)} = 0,$$

$$= \underbrace{(2s-S)\log\frac{p}{1-p}}_{\text{Term (a)}} + \underbrace{\log\frac{s-Sp}{s-S(1-p)}}_{\text{Term (b)}} = 0. \tag{46}$$

To show the required conditions, it is enough to show that (A) Eq (46) $< 0$ holds at $s = S/2$, and (B) Eq (46) $> 0$ holds at $s = S$. This is because since Eq (46) is an increasing function w.r.t. $s$, the above two conditions imply that there exists $s^*$, which is our interest, between $S/2$ and $S$. When plugging $S/2$ to $s$, $2s - S$ gets zero. Since $p > 1/2$ is given (see the above **Remark**), $\log\frac{s-Sp}{s-S(1-p)}$ becomes negative. Thus, (A) is proven. We now plug $S$ into $s$ and obtain the following result:

$$\equiv (2S-S)\log\frac{p}{1-p} + \log\frac{S-Sp}{S-S(1-p)}, \tag{47}$$

$$\equiv S\log\frac{p}{1-p} + \log\frac{S(1-p)}{Sp} \equiv S\log\frac{p}{1-p} - \log\frac{p}{1-p}, \tag{48}$$

$$\equiv (S-1)\log\frac{p}{1-p} > 0, \because p > 1/2. \tag{49}$$

Thus, we can conclude that Eq (32), which is our interest, is an increasing function w.r.t. $p > 1/2$, and has its minimum value at $p = 1/2$.

Now, we show that the above proof can also be applied to an even number $S$. Note that for an even number $S$, Eq (32) is equal to the addition of Eq (34) that starts with $s = S/2 + 1$ and Eq (50),

$$w(p,s) = \binom{S}{\frac{S}{2}} \times \frac{S}{2} \times p^{\frac{S+1}{2}}(1-p)^{\frac{S+1}{2}}. \tag{50}$$

We derive $\partial w(p,s)/\partial p$ as follows:

$$\frac{\partial w(p,s)}{\partial p} = \left(\binom{S}{\frac{S}{2}} \times \frac{S}{2}\right) \times \left(\frac{S}{2}p^{\frac{S-2}{2}}(1-p)^{\frac{S-2}{2}}(1-2p)\right) \leq 0, \forall p \in [0.5, 1]. \tag{51}$$

Since $\partial w(p,s)/\partial w(p,s) < 0$ holds for $s = (S+1)/2$, our proof is still valid for an even number $S$. We finalize the proof of **Second statement**.

**First statement.** From the above proof, we now show the lower bound of Eq (27), which is the first statement. Note that the value of Eq (27) (equivalent to Eq (32)) is minimized at $p = 0.5$. Thus, we prove the statement by finding the lower bound of Eq (28) at $p = 0.5$. For simplicity, we again use the following notation $K := \sqrt{\frac{d}{4(S-1)}}$.

$$= \frac{1}{S} \sum_{s=0}^{S} \binom{S}{s} s \left(p^s(1-p)^{S-s} + (1-p)^s p^{S-s}\right) \Phi\left((2s-1-S)\sqrt{\frac{d}{4(S-1))}}\right), \quad (52)$$

$$= \frac{1}{S} \sum_{s=0}^{S} \binom{S}{s} 2s \left(\frac{1}{2}\right)^S \Phi\left((2s-S-1)K\right). \quad (53)$$

First, consider an even number $S$. Then, for $\Phi((2s-S-1)K), \forall s \in \{1, \cdots, S\}$, consider the below relations:

$$1 \underbrace{\Phi(-S+1)}_{(a1)} + 2 \underbrace{\Phi(-S+3)}_{(b1)} + \cdots + (S-1)\underbrace{\Phi(S-3)}_{(b2)} + S\underbrace{\Phi(S-1)}_{(a2)}. \quad (54)$$

In Eq (54), $(a1) + (a2) = 1$ holds, and $(b1) + (b2) = 1$ holds also. In this manner, we can pair all terms in Eq (54). In a general sense, the following holds: $k_{11}k_{12} + k_{21}k_{22}$ such that $k_{11} + k_{21} = S, k_{12} + k_{22} = 1$, $k_{11} < k_{12}$, and $k_{21} < k_{22}$ hold. Note that $k_{11}k_{12} + k_{21}k_{22}$ is lower-bounded by $\frac{k_{11}+k_{21}}{2}$ since distributing the weight of a larger value to a smaller value will decrease the overall value. From this result, we can obtain a lower bound of Eq (54) as follows:

$$\frac{1}{S} \sum_{s=0}^{S} \binom{S}{s} 2s \left(\frac{1}{2}\right)^S \Phi\left((2s-S-1)K\right) \geq \frac{1}{S} \sum_{s=0}^{S} \binom{S}{s} 2s \left(\frac{1}{2}\right)^S \frac{1}{2} = 0.5. \quad (55)$$

Now, we consider an odd number $S$. We can extend the proof of an even number $S$ case, while additionally considering the following term: $s = \frac{S+1}{2}; \Phi(0)$ that does not have any pair. On the other hand, since $\Phi(0) = 0.5$, we still ensure Eq (55) holds in this case as well. Thus, for both even- and odd-number $S$, the value of Eq (28) at $p = 0.5$ is lower-bounded by $0.5$, which is a global lower bound of Eq (28). We finalize the proof of **First statement**.

Moreover, as described, Eq (28) is symmetric w.r.t. $p = 0.5$ on $p \in [0,1]$. In conclusion, we have verified that Eq (28) is strictly decreasing at $\mathcal{P} \in [0, 0.5]$, by extending the result at $\mathcal{P} \in [0.5, 1.0]$.

**Generalization to $v_i \in C_0$.** Note that the required condition for the improvement in $v_i \in C_0$ is $\mathbf{1}^T \sum_{v_k \in q} \boldsymbol{x}_k < 0$. Thus, we further investigate $P_{\boldsymbol{x},e}\left(\mathbf{1}^T \sum_{v_k \in q} \boldsymbol{x}_k < 0 \mid \mathcal{P}\right)$, where $v_i \in e$ and $q = e \setminus \{v_i\}$ (refer to Remark). First, let $s$ denote the number of nodes that belong to both $e_j$ and $C_0$. Note that we aim to investigate the following probability:

$$P_{\boldsymbol{x},e}\left(\vec{\mathbf{1}}^T\left(\sum_{v_k \in q} \boldsymbol{x}_k\right) < 0 \mid p\right) = \frac{P_{\boldsymbol{x},e}\left(\vec{\mathbf{1}}^T\left(\sum_{v_k \in q} \boldsymbol{x}_k\right) < 0, v_i \in e \mid p\right)}{P_e\left(v_i \in e \mid p\right)}. \quad (56)$$

Here, the following holds:

$$\sum_{v_k \in q} \boldsymbol{x}_k \sim \mathcal{N}((2s-1-S)\boldsymbol{\mu}_0, (S-1)\mathbf{I}), \quad (57)$$

$$\vec{\mathbf{1}}^T\left(\sum_{v_k \in q} \boldsymbol{x}_k\right) \sim \mathcal{N}\left(-\frac{(2s-1-S)d}{2}, (S-1)d\right), \quad (58)$$

$$P_{\boldsymbol{x}}\left(\vec{\mathbf{1}}^T\left(\sum_{v_k \in q} \boldsymbol{x}_k\right) < 0 \mid s\right) = \Phi\left((2s-S-1)\sqrt{\frac{d}{4(S-1)}}\right). \quad (59)$$

Note that Eq. (59) is equivalent to Eq. (18). Moreover, the following also holds:

$$P_e(s, v_i \in e \mid p) = \binom{S}{s} \left( \underbrace{\frac{p^s(1-p)^{S-s}}{2}}_{\text{prob. of } s \text{ at } c=0} + \underbrace{\frac{(1-p)^s p^{S-s}}{2}}_{\text{prob. of } s \text{ at } c=1} \right) \underbrace{\left( 1 - \frac{\binom{N-1}{s}}{\binom{N}{s}} \right)}_{\text{prob. of } v_i \in C_0 \cap e}. \tag{60}$$

Here again, Eq. (21) is equivalent to Eq. (60). Finally, we guarantee that Eq. (56) is expressed as follows:

$$\equiv \sum_{s=0}^{S} \frac{P_{\boldsymbol{x}}\left( \vec{\mathbf{1}}^T \left( \sum_{v_k \in q} \boldsymbol{x}_k \right) > 0 \;\middle|\; s \right) \times P_e(s, v_i \in e)}{P_e(v_i \in e \mid p)}, \tag{61}$$

$$\equiv \frac{2N}{S} \sum_{s=0}^{S} \binom{S}{s} \left( \frac{p^s(1-p)^{S-s}}{2} + \frac{(1-p)^s p^{S-s}}{2} \right) \frac{s}{N} \Phi\left( (2s-1-S)\sqrt{\frac{d}{4(S-1)}} \right), \tag{62}$$

$$= \frac{1}{S} \sum_{s=0}^{S} \binom{S}{s} s \left( p^s(1-p)^{S-s} + (1-p)^s p^{S-s} \right) \Phi\left( (2s-1-S)\sqrt{\frac{d}{4(S-1))}} \right). \tag{63}$$

Importantly, Eq. (63) is equal to Eq. (28). This result implies that Theorem 2 and its proof is generalizable to the case of $v_i \in C_0$ also.

$\square$

### A.3 EMPIRICAL DEMONSTRATION ON THE PROOF.

To empirically validate the proof provided above, we conduct toy experiments. We consider every combination of $S \in \{2, 3, 4, 5, 6, 7, 8\}$, which is a size of $e_j$ (i.e., $|e_j|$), and $d \in \{2, 3, 4, 5, 6, 7, 8\}$, which is a dimension of input node feature $\boldsymbol{x}_i \in \mathbb{R}^d, \forall v_i \in \mathcal{V}$. Then, we visualize how Eq (27) varies w.r.t. $S$, $d$, and $\mathcal{P} \in [0, 1]$. Note that $X$-axis of each plot corresponds to $\mathcal{P}$. As shown in Figure 3, for all cases, we can verify that the value of $P_{\boldsymbol{x}, e}\left( \vec{\mathbf{1}}^T \left( \sum_{v_k \in q} \boldsymbol{x}_k \right) > 0 \;\middle|\; \mathcal{P} \right)$ is strictly-increasing w.r.t. $\mathcal{P} \in [0.5, 1]$ and -decreasing w.r.t. $\mathcal{P} \in [0, 1]$ (statement 2), and the value is lower bounded by 0.5 (statement 1). Thus, we can ensure that Theorem 2 is valid through empirical verification.

## B ADDITIONAL THEORETICAL ANALYSIS

### B.1 EXISTENCE OF REASONABLE SOLUTIONS IN THE HYPEREDGE FILLING TASK

In this section, we investigate the *existence of a reasonable solution* in the proposed hyperedge filling task. To this end, we first formalize the concept of *the hyperedge filling task is solved*.

**Definition 1** (Solved hyperedge filing task). *For a hypergraph $G = (\mathcal{V}, \mathcal{E})$, the hyperedge filling task is solved when the following holds for a probabilistic model $p_{(\mathbf{X}, \mathcal{E}, \Theta)}(\cdot)$:*

$$p_{(\mathbf{X}, \mathcal{E}, \Theta)}(v_i \mid e_j \setminus \{v_i\}) > p_{(\mathbf{X}, \mathcal{E}, \Theta)}(v_k \mid e_j \setminus \{v_i\}),$$
$$\forall v_i \in e_j, \forall v_k \in \{v_s : \{v_s\} \cap (e_j \setminus \{v_i\}) \notin \mathcal{E}, v_s \in \mathcal{V}\}, , \forall e_j \in \mathcal{E}.$$

Roughly, Definition 1 indicates that for all possible hyperedge filling cases in a hypergraph $G$, a probabilistic model $p$ always has the greater probability of filling a correct node for a given query subset than that of filling a wrong node. It is important to note that our goal is to identify whether there exists a *reasonable solution* for $p(\cdot)$, not a trivial solution. Thus, we formalize the concept of a reasonable solution as the following definition:

**Definition 2** (Reasonable solution). *Let $\mathcal{Q} := \{e_j \setminus \{v_i\} : \forall v_i \in e_j, \forall e_j \in \mathcal{E}\}$. For a hypergraph $G = (\mathcal{V}, \mathcal{E})$, a reasonable solution of the hyperedge filling task is a pair of node representations $\mathbf{Z} \in \mathbb{R}^{|\mathcal{V}| \times d}$ and query set representations $\mathbf{Q} \in \mathbb{R}^{|\mathcal{Q}| \times d}$ such that:*

$$\boldsymbol{z}_i^T \boldsymbol{q}_j > \boldsymbol{z}_k^T \boldsymbol{q}_j,$$
$$\forall v_i \in S_j, \forall v_k \in \mathcal{V} \setminus S_j, \forall q_j \in \mathcal{Q}, \text{ where } S_j = \{v_s : (\{v_s\} \cap q_j \in \mathcal{E}) \wedge (v_s \notin q_j), v_s \in \mathcal{V}\}.$$

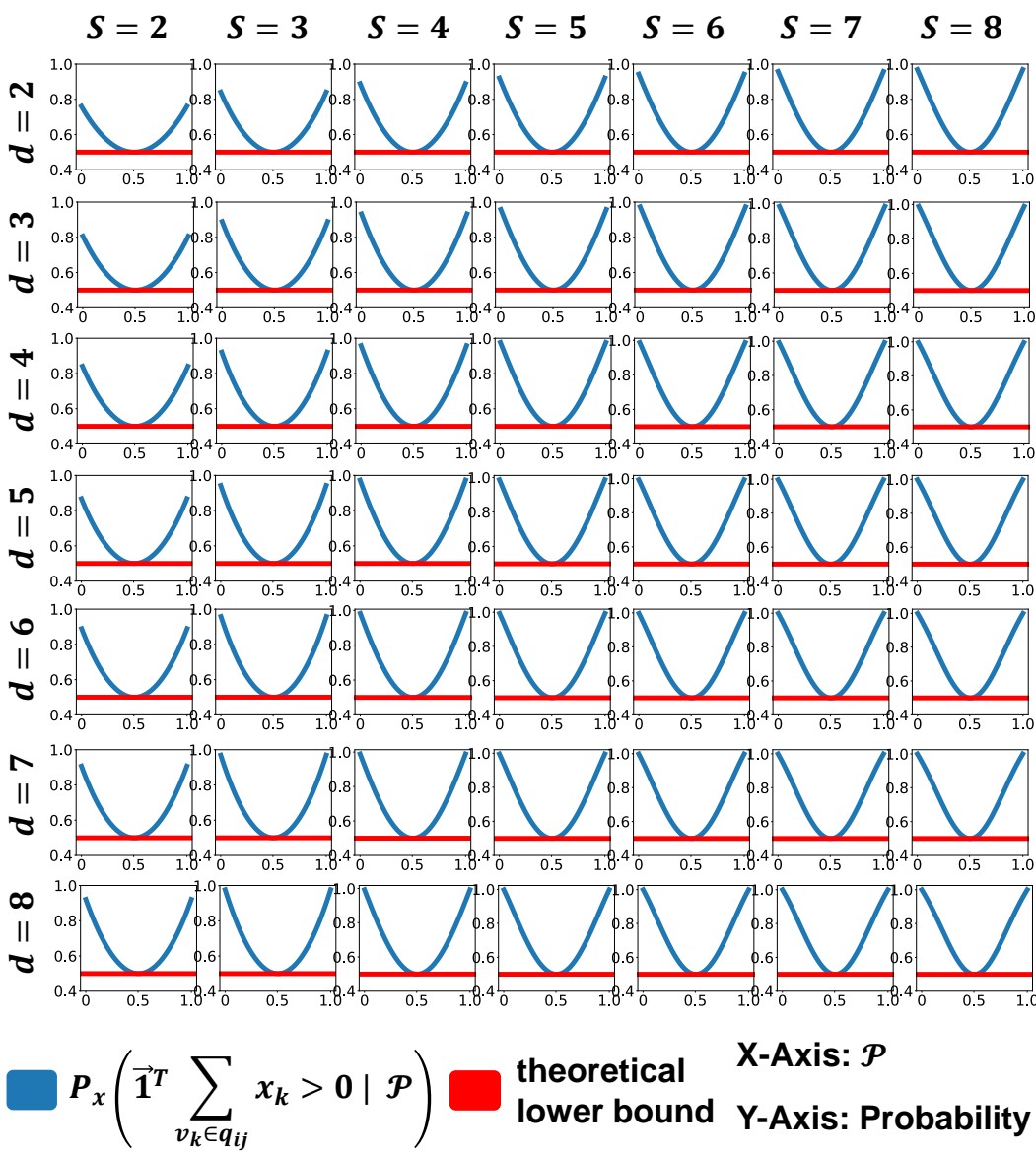

Figure 3: Empirical demonstration of Theorem 2. Note that $S$ denotes the size of a hyperedge, and $d$ denotes the dimension of features. As stated, $P_{\boldsymbol{x},e}\left(\vec{\mathbf{1}}^T\left(\sum_{v_k\in q}\boldsymbol{x}_k\right) > 0 \,\middle|\, \mathcal{P}\right)$ is strictly increasing in $\mathcal{P}\in[0.5,1]$ (statement 2), and lower bounded by $0.5$ (statement 1).

Now, we analyze whether there exists a reasonable solution for any hypergraph $G=(\mathcal{V},\mathcal{E})$.

**Theorem 3** (Existence of a reasonable solution). *For any hypergraph $G=(\mathcal{V},\mathcal{E})$, there exists a reasonable solution for some embedding dimension $d$.*

*Proof.* Consider a binary matrix $\mathbf{B}\in\{1,0\}^{|V|\times|\mathcal{Q}|}$, where each column $j\in\{1,\cdots,|\mathcal{Q}|\}$ represents one-hot encoding of $S_j$ over $\mathcal{V}$ (i.e., $\mathbf{B}_{ij}=\mathbf{1}[v_i\in S_j]$, where $\mathbf{1}[\cdot]$ is an indicator function). Here, we consider a matrix decomposition of $\mathbf{B}$, specifically, singular value decomposition of $\mathbf{B}$. Denote this singular value decomposition as $\mathbf{B}=\mathbf{U}\Sigma\mathbf{V}^T$, where $\mathbf{U}\in\mathbb{R}^{|\mathcal{V}|\times\text{rank}(\mathbf{B})}$, $\mathbf{V}\in\mathbb{R}^{\text{rank}(\mathbf{B})\times\text{rank}(\mathbf{B})}$, and $\mathbf{V}\in\mathbb{R}^{|\mathcal{Q}|\times\text{rank}(\mathbf{B})}$. Let $\mathbf{Z}:=\mathbf{U}\Sigma$ and $\mathbf{Q}:=\mathbf{V}$. In such a case, the following holds:

$$\boldsymbol{z}_i^T\boldsymbol{q}_j = \begin{cases} 1 & \text{if } v_i\in S_j, \\ 0 & \text{otherwise.} \end{cases} \tag{64}$$

Thus, $\boldsymbol{z}_i^T \boldsymbol{q}_j > \boldsymbol{z}_k^T \boldsymbol{q}_j, \forall v_i \in S_j, \forall v_k \in \mathcal{V} \setminus S_j, \forall q_j \in \mathcal{Q}$ always holds, which implies that $\mathbf{Z} := \mathbf{U}\Sigma$ and $\mathbf{Q} := \mathbf{V}$ are one of the reasonable solutions. $\square$

From Theorem 3, we demonstrate that there exists a reasonable solution for any hypergraph $G$.

### B.2 DIMENSIONAL COLLAPSE OF HYPEBOY WITHOUT PROJECTION HEADS.

In this subsection, we provide a theoretical analysis of why HYPEBOY without projection heads may suffer from dimensional collapse. Jing et al. (2022) have suggested that one of the reasons for the dimensional collapse in contrastive learning lies in the gradient flow (spec., mechanism of how representations are updated via contrastive loss). Specifically, consider a linear model that yields representations of a $n$ number of data by $\mathbf{Z} = \mathbf{XW}$, where $\mathbf{X} \in \mathbb{R}^{n \times d}$ are node features and $\mathbf{W} \in \mathbb{R}^{d \times k}$ is a learnable parameter matrix. When we update parameters $\mathbf{W}$ by using contrastive losses, $k$-th biggest singular value of weight matrices $\sigma_k$ evolve proportional to its own value (e.g., $\tilde{\sigma}_k \leftarrow \sigma_k \times c$, where $\tilde{\sigma}_k$ is a $k$-th biggest singular value of updated $\mathbf{W}$ and $c$ is a positive constant). Due to this mechanism, small single values grow slowly or diminish, causing dimensional collapse.

In this analysis, motivated by the analysis of Jing et al. (2022), we track how singular values of node representations change over learning hyperedge filling task. Following the setting of Jing et al. (2022), we consider a linear model: $\mathbf{Z} = \mathbf{XW}$. Note that $\mathbf{X}$ are given features of data points. Thus, the change of singular values of embeddings $\mathbf{Z}$ depends on $\mathbf{W}$.

We first define the loss of hyperedge filling as follows:

$$\mathcal{L} := -\sum_{e_j \in \mathcal{E}} \sum_{v_i \in e_j} \log \left( \frac{\exp\left( \boldsymbol{z}_i^T \left( \sum_{v_s \in q_{ij}} \boldsymbol{z}_s \right) \right)}{\sum_{v_t \in \mathcal{V}} \exp\left( \boldsymbol{z}_t^T \left( \sum_{v_s \in q_{ij}} \boldsymbol{z}_s \right) \right)} \right). \tag{65}$$

Now, we obtain the closed form of the following update equation, which is based on the gradient descent: $\mathbf{W} \leftarrow \mathbf{W} - \gamma \nabla_{\mathbf{W}} \mathcal{L}$, where $\gamma$ is a fixed learning rate. We first derive $(\partial \mathcal{L}/\partial \mathbf{Z})$, and utilize the chain rule: $(\partial \mathcal{L}/\partial \mathbf{Z}) \times (\partial \mathbf{Z}/\partial \mathbf{W})$.

The derivative of numerators of Eq (65) can be written as follows:

$$-\frac{\partial \left( \sum_{e_j \in \mathcal{E}} \sum_{v_i \in e_j} \boldsymbol{z}_i^T \left( \sum_{v_s \in q_{ij}} \boldsymbol{z}_s \right) \right)}{\partial \mathbf{Z}} = -\mathbf{EZ},$$

$$\text{where } \mathbf{E} \in \mathbb{R}^{|\mathcal{V}| \times |\mathcal{V}|}, \text{ s.t. } \mathbf{E}_{ij} = 2 \times |\{e_k : \{v_i, v_j\} \subseteq e_k, \forall e_k \in \mathcal{E}\}|.$$

We now derive the derivative of the denominators of Eq (65):

$$\frac{\partial \left( \sum_{e_j \in \mathcal{E}} \sum_{v_i \in e_j} \log \left( \sum_{v_t \in \mathcal{V}} \exp\left( \boldsymbol{z}_t^T \left( \sum_{v_s \in q_{ij}} \boldsymbol{z}_s \right) \right) \right) \right)}{\partial \mathbf{Z}}. \tag{66}$$

We denote a multiset of every possible query subset as $\mathcal{Q}$ (i.e., $\mathcal{Q} = \{e_j \setminus \{v_i\} : v_i \in e_j, e_j \in \mathcal{E}\}$). The reason for multiset is that given subsets can be duplicated due to the nature of overlapping hyperedges (e.g., when we omit a node $v_1$ from $\{v_1, v_2, v_3\}$ and $v_4$ from $\{v_4, v_2, v_3\}$, remaining sets become identical).

We split the derivation into the node level. In addition, we let a subset of $\mathcal{Q}$ such that includes a node $v_k$ as $\mathcal{Q}_k$. Then, for a node $v_k$, its gradient is computed as follows:

$$\sum_{q_j \in \mathcal{Q}} \frac{\exp\left( \boldsymbol{z}_k^T \left( \sum_{v_s \in q_j} \boldsymbol{z}_s \right) \right)}{\sum_{v_t \in \mathcal{V}} \exp\left( \boldsymbol{z}_t^T \left( \sum_{v_s \in q_j} \boldsymbol{z}_s \right) \right)} \times \left( \sum_{v_s \in q_j} \boldsymbol{z}_s \right) \tag{67}$$

$$+ \sum_{q_l \in \mathcal{Q}_k} \sum_{v_i \in \mathcal{V}} \frac{\exp\left( \boldsymbol{z}_i^T \left( \sum_{v_s \in q_l} \boldsymbol{z}_s \right) \right)}{\sum_{v_t \in \mathcal{V}} \exp\left( \boldsymbol{z}_t^T \left( \sum_{v_s \in q_l} \boldsymbol{z}_s \right) \right)} \boldsymbol{z}_i. \tag{68}$$

We define two more matrices to express Eq. (67) and (68) in a simple matrix form. By assigning an index $k = \{1, \cdots, |\mathcal{Q}|\}$ to $q_j \in \mathcal{Q}$, we define a binary matrix $\mathbf{Q} \in \mathbb{R}^{|\mathcal{Q}| \times |\mathcal{V}|}$ that represents $\mathcal{Q}$:

$$
\mathbf{Q}_{ji} = \begin{cases} 1 & \text{if } v_i \in q_j, \\ 0 & \text{otherwise.} \end{cases}
$$

In addition, we denote a score matrix $\mathbf{A} \in \mathbb{R}^{|\mathcal{Q}| \times |\mathcal{V}|}$ that models a probability of filling each node $v_i \in \mathcal{V}$ to each query subset $q_j \in \mathcal{Q}$, which is defined as follows:

$$
\mathbf{A}_{ki} = \frac{\exp\left(\boldsymbol{z}_i^T \left(\sum_{v_s \in q_j} \boldsymbol{z}_s\right)\right)}{\sum_{v_t \in \mathcal{V}} \exp\left(\boldsymbol{z}_t^T \left(\sum_{v_s \in q_j} \boldsymbol{z}_s\right)\right)}. \tag{69}
$$

By using $\mathbf{A}$ and $\mathbf{Q}$, we can express Eq (67) and (68) for all nodes as follows:

$$
\left( \underbrace{\mathbf{Q}^T \mathbf{A}}_{\text{Eq (67) of all nodes}} + \underbrace{\mathbf{A}^T \mathbf{Q}}_{\text{Eq (68) of all nodes}} \right) \mathbf{Z}. \tag{70}
$$

Finally, we express $(\partial \mathcal{L} / \partial \mathbf{Z})$ as follows:

$$
\frac{\partial \mathcal{L}}{\partial \mathbf{Z}} = \left( -\mathbf{E} + \mathbf{Q}^T \mathbf{A} + \mathbf{A}^T \mathbf{Q} \right) \mathbf{Z}. \tag{71}
$$

In addition, by using the chain rule and matrix derivative rules, $\partial \mathcal{L} / \partial \mathbf{W}$ is derived as follows:

$$
\frac{\partial \mathcal{L}}{\partial \mathbf{W}} = \mathbf{X}^T \left( -\mathbf{E} + \mathbf{Q}^T \mathbf{A} + \mathbf{A}^T \mathbf{Q} \right) \mathbf{X} \mathbf{W}. \tag{72}
$$

We denote updated $\mathbf{W}$ as $\tilde{\mathbf{W}}$. To sum up, update rule of $\tilde{\mathbf{W}} := \mathbf{W} - \gamma \nabla_{\mathbf{W}} \mathcal{L}$ is expressed as follows:

$$
\tilde{\mathbf{W}} := \left( \mathbf{I} + \gamma \mathbf{X}^T \left( \mathbf{E} - \mathbf{Q}^T \mathbf{A} + \mathbf{A}^T \mathbf{Q} \right) \mathbf{X} \right) \mathbf{W}. \tag{73}
$$

Here, we denote $k$-th biggest singular values of $\tilde{\mathbf{W}}$ and $\mathbf{W}$ as $\tilde{\sigma}_k$ and $\sigma_k$, respectively.

Note that $\left( \mathbf{I} + \gamma \mathbf{X}^T \left( \mathbf{E} - \mathbf{Q}^T \mathbf{A} + \mathbf{A}^T \mathbf{Q} \right) \mathbf{X} \right)$ is a function of $\mathbf{X}$, $\mathcal{E}$, and $\mathbf{W}$. Thus, we denote this term as $g(\mathbf{X}, \mathcal{E}, \mathbf{W}) \in \mathbb{R}^{d \times d}$.

If we take a closer look at $g(\mathbf{X}, \mathcal{E}, \mathbf{W})\mathbf{W}$, $\tilde{\mathbf{W}}$ is a near-linear transformation of $\mathbf{W}$ itself[7]. In such a case, from the min-max characterization rule, the following inequality holds:

$$
\tilde{\sigma}_k \le \sigma_1' \times \sigma_k, \text{ where } \sigma_1' \text{ is the biggest singular value of a matrix } g(\mathbf{X}, \mathcal{E}, \mathbf{W}). \tag{74}
$$

Eq (74) indicates that $\tilde{\sigma}_k$ is upper-bounded by the multiplication between the biggest singular value of $g(\mathbf{X}, \mathcal{E}, \mathbf{W})$ and $\sigma_k$. While it is not an exact equality, this result implies that the changed singular value is likely to be correlated with its previous singular value. Hence, small singular values are likely to grow slowly, similar to Jing et al. (2022) study, which may cause dimensional collapse.

## C DISCUSSIONS

### C.1 COMPARISON WITH HYPERGRL

In this section, we compare our HYPEBOY against HyperGRL (Du et al., 2022). Both methodologies utilize a strategy that leverages a node and a subset of a hyperedge for self-supervised learning. Specifically, this involves using a pair that consists of a node $v_i$ that belongs to a hyperedge $e_j$, and the subset of $e_j$ excluding $v_i$ ($e_j \setminus \{v_i\}$), for training purposes. However, HYPEBOY significantly differs from HyperGRL in various aspects.

---

[7]It is not a perfect linear transformation, since a function $g(\cdot)$ is also a function of $\mathbf{W}$.

**Motivation.** The motivation of HYPEBOY is markedly distinct from that of HyperGRL. The hyperedge filling task is a task that aims to correctly fill the missing node for a given (query) subset. The underlying motivation is to train a machine learning model to generate hyperedges, thereby identifying entities that are forming group interactions. This may enable the model to understand the complex, higher-order relationships within a given hypergraph. This objective aligns with the goals of question-answering approaches in NLP (Devlin et al., 2019). Conversely, the key motivation of HyperGRL, which performs node-level pretext task introduced by Du et al. (2022), is described as follows: "*the node inside one specific hyperedge should be distinguished from nodes outside this hyperedge given the context of node*". Here, the motivations behind Du et al. (2022) are akin to those of proximity-based approaches, including LINE (Tang et al., 2015).

**Method design.** The design choices of HYPEBOY and HyperGRL differ significantly. Below, we outline these technical distinctions.

- Input: HYPEBOY receives the original hypergraph structure, while HyperGRL receives a clique-expanded graph.
- Backbone encoder: HYPEBOY utilizes hypergraph neural networks, while HyperGRL employs graph neural networks.
- Augmentation: HYPEBOY augments node features and hyperedges, while HyperGRL does not utilize an augmentation strategy.
- Projection head: HYPEBOY utilizes different projection head parameters for node-projection and query subset-projection, while HyperGRL utilizes the same parameters for both.
- Negative samples: HYPEBOY utilizes the entire node as negative samples for each hyperedge filling case, while HyperGRL samples several nodes.
- SSL signal: HYPEBOY utilizes all possible (node, query subset) pairs of a given hypergraph, while HyperGRL samples a single pair for each hyperedge.
- Feature reconstruction: HYPEBOY utilizes node feature reconstruction warm-up, while HyperGRL utilizes hyperedge prediction loss as another SSL signal.

**Performance comparison.** It is important to note that HYPEBOY is superior to HyperGRL as an SSL strategy for hypergraphs. We provide empirical evidence below:

- Under the fine-tuning protocol, HYPEBOY outperforms HyperGRL in every dataset (refer to Section 5.1).
- Under the linear-evaluation protocol with the node classification task, HYPEBOY outperforms HyperGRL in every dataset (refer to Section 5.2).
- Under the linear-evaluation protocol with he hyperedge prediction task, HYPEBOY outperforms HyperGRL in 10 out of 11 datasets (refer to Section 5.2).

### C.2 POTENTIAL EXTENSIONS OF HYPEBOY

While this work mainly assumes static and undirected hypergraphs, many real-world group interactions have directions and temporal dynamics. Directed hypergraphs (Kim et al., 2023b) and temporal hypergraphs (Lee & Shin, 2023b) are utilized to include directional information and temporal information in hypergraphs, respectively. HYPEBOY can be extended to such hypergraphs by adequately considering their unique characteristics. Moreover, while we assume a transductive hypergraph setting, there is a recent effort to perform representation learning on hypergraphs under an inductive case (Behrouz et al., 2023). Considering this, HYPEBOY can also be extended to an inductive case, incorporating the idea of Behrouz et al. (2023). Lastly, HYPEBOY can be enhanced to capture more complex characteristics of a hypergraph (e.g., hyperedge-dependency of nodes (Choe et al., 2023)).

## D DATASETS

In this section, we provide details of the used datasets. In our experiments, we utilize 11 hypergraph benchmark datasets. Following Lee & Shin (2023a), we remove nodes that do not belong to any hyperedge. Data statistics of each dataset are reported in Table 4.

Table 4: Statistics of 11 datasets we utilize in our experiments.

|  | Citeseer | Cora | Pubmed | Cora-CA | DBLP-P | DBLP-A | AMiner | IMDB | MN-40 | 20News | House |
|---|---|---|---|---|---|---|---|---|---|---|---|
| $\|\mathcal{V}\|$ | 1,458 | 1,434 | 3,840 | 2,388 | 41,302 | 2,591 | 20,201 | 3,939 | 12,311 | 16,242 | 1,290 |
| $\|\mathcal{E}\|$ | 1,079 | 1,579 | 7,963 | 1,072 | 22,263 | 2,690 | 8,052 | 2,015 | 12,311 | 100 | 341 |
| $\sum_{e \in \mathcal{E}} \|e\|$ | 3,453 | 4,786 | 34,629 | 4,585 | 99,561 | 6,201 | 32,028 | 9,560 | 61,555 | 65,451 | 11,843 |
| # Classes | 6 | 7 | 3 | 7 | 6 | 4 | 12 | 3 | 40 | 4 | 2 |
| # Features | 3,703 | 1,433 | 500 | 1,433 | 1,425 | 334 | 500 | 3066 | 100 | 100 | 2 |

**Source of each dataset.** The sources of each dataset used in this work are as follows:

- The Cora, Citeseer, Pubmed, Cora-CA, and DBLP-P datasets are from the work of Yadati et al. (2019).

- The DBLP-A and IMDB datasets are from the work of Wang et al. (2019).

- The AMiner dataset is from the work of Zhang et al. (2019).

- The Mondelnet-40 (MN-40) dataset is from the work of Wu et al. (2015).

- The 20Newsgroups (20News) dataset is from the work of Dua et al. (2017).

- The House dataset is from the work of Chien et al. (2022).

**Co-citation datasets.** We utilize three co-citation datasets: Cora, Citeseer, and Pubmed. In these datasets, each node represents a publication and each hyperedge represents a set of publications co-cited by a publication. For example, if a publication has cited three publications that correspond to $v_i, v_j$, and $v_k$, respectively, these publications (nodes) are grouped as a hyperedge $\{v_i, v_j, v_k\} \in \mathcal{E}$. Node features are bag-of-words of the corresponding publication (node). Node class indicates the academic category of the corresponding publication.

**Co-authorship datasets.** We utilize four co-authorship datasets: Cora-CA, DBLP-P, DBLP-A, and AMiner. In Cora-CA, DBLP-P, and AMiner, each node represents a publication, and a set of publications that are written by the same author is grouped as a hyperedge. Node features are bag-of-words of the corresponding publication (node). Node class indicates the academic category of the corresponding publication. Conversely in DBLP-A, each node represents an author, and co-authors of a publication are grouped as a hyperedge. Node features are bag-of-words regarding the research keywords of the corresponding author [8]. Node class indicates the primary research area of the corresponding author.

**Computer graphic datasets.** We utilize one computer graphical dataset: Modelnet-40 (MN-40). In this dataset, each node represents a visual object, and hyperedges are synthetic hyperedges that have been created with a k-NN graph constructed according to the features of each data point, following Feng et al. (2019) and Chien et al. (2022). Node features are embeddings (obtained via GVCNN (Feng et al., 2018) and MVCNN (Su et al., 2015)) of the corresponding visual object. Node class indicates the category of the corresponding visual object.

**Movie-Actor dataset.** We utilize one movie-actor relational dataset: IMDB. In this dataset, each node indicates a movie, and the filmography of an actor is grouped as a hyperedge. Node features are bag-of-words of the plot of the corresponding movie. Node class indicates the genre of the movie.

**News dataset.** We utilize one news dataset: 20NewsGroups (20News). In this dataset, each node represents a document of 20 newsgroups and a set of documents containing a particular word is grouped as a hyperedge. Node features are TF-IDF representations of news messages of the corresponding news document. Node class indicates the category of the corresponding news document.

**Political membership dataset.** We utilize one political membership dataset: House. In this dataset, each node represents a member of the US House of Representatives, and members belonging to the same committee are grouped as a hyperedge. Node features are generated by adding noise to the one-hot encoded label vector, following Chien et al. (2022). Node class indicates the political party of the corresponding member.

---

[8]Please do not be confused with the academic corresponding author.

# E  EXPERIMENTAL DETAILS

In this section, we provide our experimental details. We first describe the machines we use in our experiments. Then, we provide details of HYPEBOY and baseline methods, including their implementation and hyperparameter settings.

## E.1  MACHINES

All experiments are conducted on a machine with NVIDIA RTX 8000 D6 GPUs (48GB memory) and two Intel Xeon Silver 4214R processors.

## E.2  DETAILS OF BASELINE METHODS

**Neural networks.** We utilize 10 (semi-)supervised baseline models: MLP, HGNN (Feng et al., 2019), HyperGCN (Yadati et al., 2019), HNHN (Dong et al., 2020), three unified hypergraph networks, (UniGCN, UniGIN, UniGCNII) (Huang & Yang, 2021), AllSet (Chien et al., 2022), ED-HNN (Wang et al., 2023a), and PhenomNN (Wang et al., 2023b). For all the methods, we fix their hidden dimension and model layers as 128 and 2, respectively.

**Graph SSL.** To utilize two graph-generative graph SSL methods, which are GraphMAE2 (Hou et al., 2023) and MaskGAE (Li et al., 2023), and one hypergraph-generative graph SSL method, which is HyperGRL (Du et al., 2022), we transform the input hypergraph into a graph by using the clique expansion, which converts each hyperedge into a clique of a graph (Dong et al., 2020). Moreover, we utilize a 2-layer GCN (Kipf & Welling, 2017) with hidden dimension 128 as a backbone encoder of these two SSL methods.

**H-GD.** One of our baseline methods H-GD directly extends the group discrimination approach, an SSL technique designed on ordinary graphs (Zheng et al., 2022). We strictly follow the overall structure suggested by Zheng et al. (2022), while the feature augmentation and topology augmentation have been replaced into $\tau_x$ and $\tau_e$ that are described in Section 4.1, respectively.

## E.3  DETAILS OF SETTINGS

**Hyperedge prediction setting.** Note that we need to obtain negative hyperedge samples to evaluate a model on the hyperedge prediction task. In our experiments, we utilize negative hyperedge samples that are generated by the Size Negative Sample strategy (Patil et al., 2020). Specifically, we first sample the size of the current hyperedge from the real-world hyperedge size distribution of the corresponding dataset. Then, we sample nodes uniformly at random and fill the corresponding hyperedge. By using the strategy, we obtain negative hyperedges with the same number of ground-truth hyperedges and utilize them for training/validation/test.

**Hyperedge embedding.** To perform hyperedge prediction, we require a representation of each hyperedge. On the other hand, since the output of utilized HNNs and GNNs are node embeddings, we need an additional technique to acquire hyperedge representation from node embeddings. To this end, we utilize a max-min pooling aggregation function (Yadati et al., 2020). For example, for a hyperedge $e_i = \{v_j, v_k, v_\ell\}$, whose constituent node embeddings are $z_j, z_k, z_\ell \in \mathbb{R}^d$, respectively, its hyperedge embedding $h_i^{(e)}$ is obtained as follows:

$$h_i = (\max_{t \in \{j,k,\ell\}} z_{ts} - \min_{t \in \{j,k,\ell\}} z_{ts})_{s=1}^d. \tag{75}$$

Finally, we utilize $h_i$ as a representation of $e_i$.

## E.4  DETAILS OF IMPLEMENTATION

**Feature reconstruction warm-up of HYPEBOY.** As described in Section 4.4, before training an encoder HNN with HYPEBOY, we perform a feature reconstruction process, which directly extends GraphMAE (Hou et al., 2022) to hypergraph structure. This process mitigates the encoder's over-reliance on the projection heads, improving the performance in downstream tasks (Section 5.3). For the decoder HNN $g_\psi$, we utilize a 2-layer UniGCNII (Huang & Yang, 2021), which has the same

Table 5: Hyperparameter combination of HYPEBOY that shows the best validation accuracy on each dataset. $p_v \in [0,1]$ indicates the magnitude of feature augmentation, $p_e$ indicates the magnitude of topological augmentation, and $SSLepochs$ indicates the training epoch of hyperedge filling task.

|  | Citeseer | Cora | Pubmed | Cora-CA | DBLP-P | DBLP-A | AMiner | IMDB | MN-40 | 20News | House |
|---|---|---|---|---|---|---|---|---|---|---|---|
| $p_v$ | 0.2 | 0.4 | 0.0 | 0.4 | 0.0 | 0.1 | 0.2 | 0.3 | 0.4 | 0.2 | 0.4 |
| $p_e$ | 0.9 | 0.9 | 0.5 | 0.8 | 0.6 | 0.9 | 0.6 | 0.9 | 0.5 | 0.5 | 0.8 |
| SSL epochs | 120 | 200 | 180 | 200 | 180 | 120 | 100 | 180 | 180 | 80 | 140 |

architecture as the encoder HNN. We mask $50\%$ of input nodes and set hyperedge augmentation ratio $p_e$ as 0.2.

**Projection heads of HYPEBOY.** We utilize a two-layer MLP model with a ReLU (Nair & Hinton, 2010) activation function without dropout- and normalization-layers, for both node projection head $f'_\phi$ and set projection head $f''_\rho$ (Section 4.2). Note that these projection heads are updated via the hyperedge filling task together with the encoder HNN.

**Training details and hyperparameters.** We train all models using the Adam optimizer (Kingma & Ba, 2015) with a fixed weight decay rate of $10^{-6}$. We fix the hidden dimension and dropout rate of all models as 128 and 0.5, respectively. When training any neural network for downstream tasks, we train a model for 200 epochs, and for every 10 epochs, we evaluate the validation accuracy of the model. Then, we utilize the model's checkpoint that yields the best validation accuracy to perform the final evaluation of the model on the test dataset.

For a linear evaluation protocol of node classification, we utilize a logistic classifier, with a learning rate 0.001. For a linear evaluation protocol of hyperedge classification, we utilize a two-layer MLP, with a learning rate of 0.001 and a dropout rate of 0.5.

Regarding hyperparameter tuning, for each model and dataset, we find the hyperparameter combination that gives the best validation performance. Then, with the selected hyperparameters, we train the model on a training dataset and evaluate the trained model on a test dataset.

For all the supervised models, we tune the learning rate as a hyperparameter within $\{0.05, 0.01, 0.005, 0.001, 0.0005, 0.0001\}$. Moreover, especially for PhenomNN, which is a state-of-the-art hypergraph neural network, we additionally tune their other hyperparameters. Specifically, along with the learning rate, we additionally tune $\lambda_0$ and $\lambda_1$ within $\{0, 1, 10\}$, which are weights of different message-passing functions, and $\alpha$ within $\{0, 1, 10\}$, which is the weight of a node's own feature.

For all the SSL baseline methods, we set broader search spaces, which are as follows:

- TriCL (Lee & Shin, 2023a): We tune their feature augmentation magnitude within $\{0.2, 0.3, 0.4\}$, hyperedge augmentation magnitude within $\{0.2, 0.3, 0.4\}$, and learning rate of its all components within $\{0.01, 0.001, 0.0001\}$.

- HyperGCL (Wei et al., 2022): We tune learning rate of an encoder within $\{0.01, 0.001, 0.0001\}$, that of a view generator within $\{0.01, 0.001, 0.0001\}$, and the weight of contrastive loss $\beta$ within $\{0.5, 1, 2\}$.

- HGD, a variant of GD (Zheng et al., 2022): We tune their feature augmentation magnitude within $\{0.2, 0.3, 0.4\}$, hyperedge augmentation magnitude within $\{0.2, 0.3, 0.4\}$, and learning rate of its all components within $\{0.01, 0.001, 0.0001\}$.

- GraphMAE2 (Hou et al., 2023): We tune mask ratio within $\{0.25, 0.5, 0.75\}$ and learning rate of its all components within $\{0.01, 0.001, 0.0001\}$.

- MaskGAE (Li et al., 2023) We tune degree loss weight $\alpha$ within $\{0.001, 0.002, 0.003\}$ and learning rate of its all components within $\{0.01, 0.001, 0.0001\}$.

- HyperGRL (Du et al., 2022) We tune negative-sampling adjacency matrix order $k$ within $\{1, 2, 3\}$ and learning rate of its all components within $\{0.01, 0.001, 0.0001\}$.

In addition, we set self-supervised learning epochs of the above methods as hyperparameters, searching within $\{50, 100, 150, \cdots, 450, 500\}$.

Table 6: Comparison between HYPEBOY (HYPEBOY (denoted by w/ Aug.)) and a variant of HY-PEBOY that does not use augmentation strategy (denoted by w/o Aug.) in the node classification task under two protocols. The best performance is colored as a green. In most of the settings, HYPEBOY outperforms its variant.

| Protocol | Method | Citeseer | Cora | Pubmed | Cora-CA | DBLP-P | DBLP-A | AMiner | IMDB | MN-40 | 20News | House |
|---|---|---|---|---|---|---|---|---|---|---|---|---|
| Fine | w/o Aug. | 53.5 (8.7) | 58.6 (8.3) | 75.5 (4.0) | 62.9 (6.0) | 87.8 (0.4) | 79.7 (2.3) | 33.9 (2.1) | 47.2 (2.5) | 90.7 (0.7) | 77.5 (0.9) | 69.9 (4.9) |
| tuning | w/ Aug. | 56.7 (9.8) | 62.3 (7.7) | 77.0 (3.4) | 66.3 (4.6) | 88.2 (0.4) | 80.6 (2.3) | 34.1 (2.2) | 47.6 (2.5) | 90.4 (0.9) | 77.6 (0.9) | 70.4 (4.8) |
| Linear | w/o Aug. | 55.8 (9.0) | 60.6 (7.3) | 72.7 (3.4) | 63.3 (4.6) | 87.6 (0.5) | 81.4 (2.5) | 34.2 (2.7) | 47.8 (2.0) | 89.7 (1.9) | 75.1 (1.6) | 67.5 (1.6) |
| evaluation | w/ Aug. | 59.6 (9.9) | 63.5 (9.4) | 75.0 (3.4) | 66.0 (4.6) | 87.9 (0.5) | 81.2 (2.7) | 34.3 (3.2) | 48.8 (1.8) | 89.2 (2.2) | 75.7 (2.1) | 69.4 (5.4) |

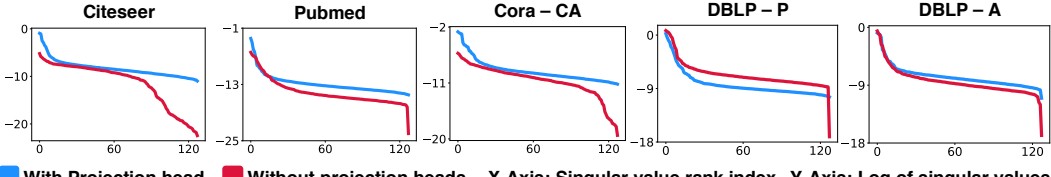

■ With Projection head   ■ Without projection heads   X-Axis: Singular value rank index   Y-Axis: Log of singular values

Figure 4: Analyzing dimensional collapse of HYPEBOY with/without projection heads on five benchmark datasets. While HYPEBOY does not suffer from dimensional collapse, its variant that does not utilize projection heads, suffers from this issue.

For HYPEBOY, we tune feature augmentation magnitude $p_x$ within $\{0.0, 0.1, 0.2, 0.3, 0.4\}$ and hyperedge augmentaiton magnitude $p_e$ within $\{0.5, 0.6, 0.7, 0.8, 0.9\}$. We fix the learning rate and training epochs of the feature reconstruction warm-up as $0.001$ and $300$, respectively.

Regarding the overall training process of HYPEBOY, we first train an encoder HNN with a feature reconstruction process for 300 epochs. Then, we train the encoder HNN with the hyperedge filling task. Specifically, the hyperedge filling training epoch is a hyperparameter, which we search within $\{20, 40, \cdots, 180, 200\}$. We report a hyperparameter combination of each dataset that shows the best validation accuracy in our fine-tuning protocol experiment (Section 5.1) in Table 5.

## F    ADDITIONAL EXPERIMENTAL RESULTS

In this section, we present additional experimental results that were excluded from the main paper due to space constraints.

### F.1    COMPARISON AGAINST HYPEBOY WITHOUT AUGMENTATION

As described in Section 4.1, augmentation (spec., masking) has played a key role in various generative SSL methods for obtaining effective representations. Motivated by these findings, we incorporate augmentation process $\tau_x$ and $\tau_e$ into HYPEBOY. To assess the impact of augmentation, we analyze a variant of HYPEBOY that omits the augmentation step. In this variant, the input to HYPEBOY's encoder HNN consists of ground-truth features and topology, $(\mathbf{X}, \mathcal{E})$. We then compare the performance of this variant against our proposed HYPEBOY in node classification tasks, under both fine-tuning and linear evaluation protocols. As shown in Table 6, HYPEBOY consistently outperforms the variant across most of the settings, demonstrating the necessity of augmentation in the model's performance.

### F.2    DIMENSIONAL COLLAPSE ANALYSIS

In Section 4.2, we discuss the role of projection heads in terms of dimensional collapse: projection heads encourage an encoder HNN to avoid dimensional collapse. We further analyze this phenomenon in five benchmark hypergraph datasets: Citeseer, Pubmed, Cora-CA, DBLP-P, and DBLP-A. As shown in Figure 4, across all these datasets, we verify that certain singular values of embeddings achieved via HYPEBOY without projection heads drop to zero. This empirically demonstrates that dimensional collapse has occurred in such cases (refer to red lines). Notably, such issues are not observed in HYPEBOY (refer to blue lines).

Table 7: Performance of SSL methods in various HNNs under fine-tuning protocol. The best and second-best performances are colored green and yellow, respectively. NA indicates the corresponding encoder without pre-training. HYPEBOY outperforms other hypergraph SSL methods in most of the settings.

| | Method | Citeseer | Cora | Pubmed | Cora-CA | DBLP-P |
|---|---|---|---|---|---|---|
| HGNN | NA | 41.9 (7.8) | 50.0 (7.2) | 72.9 (5.0) | 50.2 (5.7) | 85.3 (0.8) |
| | TriCL | 49.3 (10.3) | 56.9 (10.0) | 74.3 (4.1) | 57.2 (6.1) | 87.4 (5.0) |
| | HyperGCL | 47.2 (9.3) | 59.5 (7.6) | 76.3 (4.3) | 53.3 (6.9) | 85.8 (0.4) |
| | HYPEBOY | 52.1 (9.1) | 61.1 (9.8) | 76.8 (4.3) | 60.0 (6.0) | 87.4 (0.5) |
| MeanPoolingConv | NA | 39.9 (10.0) | 48.5 (8.9) | 72.4 (4.0) | 50.6 (7.5) | 85.7 (0.7) |
| | TriCL | 46.6 (8.3) | 59.8 (8.9) | 74.0 (3.8) | 57.3 (4.8) | 87.0 (5.0) |
| | HyperGCL | 43.2 (9.4) | 59.6 (7.6) | 74.2 (4.3) | 55.0 (6.7) | 85.7 (0.9) |
| | HYPEBOY | 54.5 (8.3) | 61.2 (7.8) | 77.3 (3.4) | 63.0 (4.4) | 86.8 (0.5) |
| SetGNN | NA | 43.6 (6.3) | 48.9 (9.2) | 69.8 (4.5) | 50.7 (6.7) | 82.8 (1.0) |
| | TriCL | 48.9 (7.6) | 57.7 (8.4) | 72.0 (4.5) | 57.8 (4.9) | 85.1 (0.5) |
| | HyperGCL | 46.8 (8.5) | 54.6 (7.5) | 73.1 (3.9) | 56.9 (5.3) | 83.6 (0.7) |
| | HYPEBOY | 53.8 (9.1) | 62.3 (6.4) | 73.4 (3.4) | 61.0 (3.9) | 84.2 (0.7) |

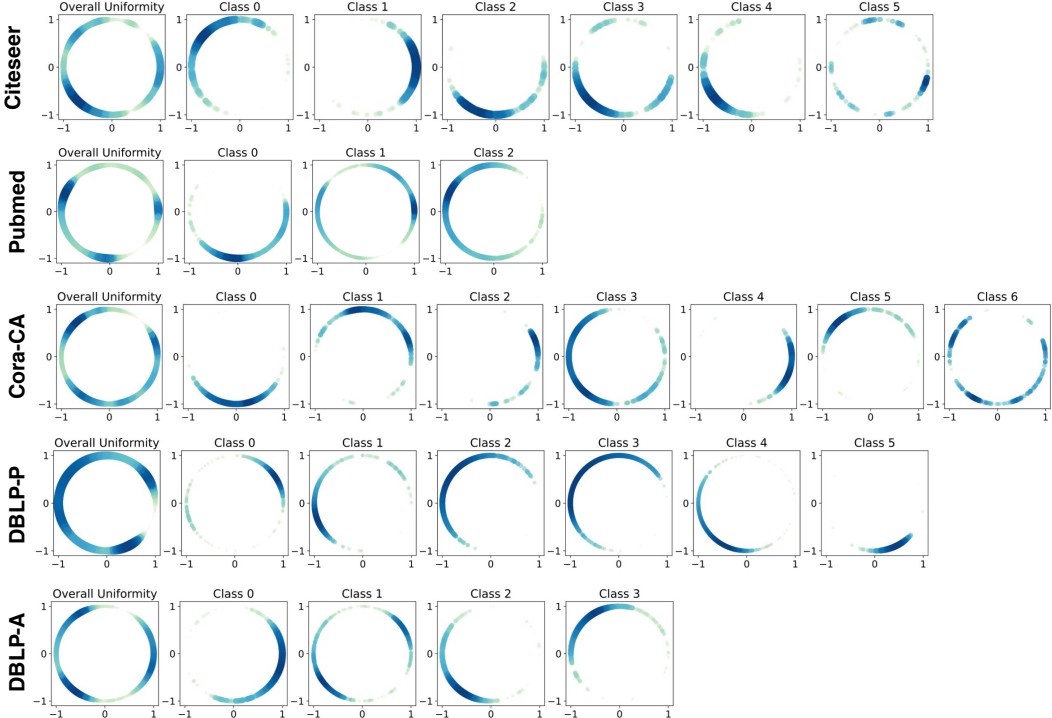

Figure 5: Analyzing alignment and uniformity of representations obtained via HYPEBOY. In most cases, representations obtained by HYPEBOY achieve both alignment and uniformity.

### F.3 ALIGNMENT AND UNIFORMITY ANALYSIS

In Section 4.3, we discuss the role of our $p_{(\mathbf{X}, \mathcal{E}, \Theta)}(\cdot)$ design choice in promoting alignment and uniformity of node representations. This characteristic is further analyzed across five datasets: Citeseer, Pubmed, Cora-CA, DBLP-P, and DBLP-A. As shown in Figure 5, node representations obtained from HYPEBOY achieve both uniformity (first column) and class alignment (other columns) in most of the cases.

Table 8: Efficacy of pretext tasks under fine-tuning protocol. The best performances are colored as green. Among the three tasks, the hyperedge filling task shows superior performance.

| Method | Citeseer | Cora | Pubmed | Cora-CA | DBLP-P |
|---|---|---|---|---|---|
| Naive contrastive learning | 59.2 (6.7) | 44.5 (10.2) | 75.2 (4.0) | 62.1 (5.3) | 86.7 (0.5) |
| Naive feature reconstruction | 58.6 (8.0) | 51.5 (9.2) | 74.7 (5.1) | 61.9 (7.3) | 87.3 (0.6) |
| Naive hyperedge filling | 60.7 (8.2) | 51.6 (11.8) | 76.2 (3.6) | 63.5 (6.0) | 88.1 (0.5) |

Table 9: Comparing HYPEBOY against other topological SSL methods under fine-tuning protocol. The best performances are colored as green. Among all the four methods, HYPEBOY shows the best performance.

| Data | Method | Citeseer | Cora | Pubmed | Cora-CA | DBLP-P |
|---|---|---|---|---|---|---|
| Graph | Edge filling | 44.1 (11.3) | 55.7 (8.2) | 73.6 (3.8) | 55.5 (8.2) | 86.9 (0.5) |
| Graph | Hyperedgeedge filling | 42.8 (9.2) | 56.4 (6.0) | 73.6 (4.0) | 55.0 (8.6) | 86.7 (0.5) |
| Hypergraph | Hyperedge prediction | 43.9 (7.6) | 56.9 (9.7) | 73.0 (4.9) | 59.1 (6.5) | 85.8 (0.6) |
| Hypergraph | Hypereedge filling (HYPEBOY) | 56.7 (9.8) | 62.3 (7.7) | 77.0 (3.4) | 66.3 (4.6) | 88.2 (0.4) |

## F.4 ENCODER ANALYSIS

In this section, we experimentally verify that the efficacy of HYPEBOY extends beyond UniGC-NII (Huang & Yang, 2021). For this purpose, we employ three HNNs: HGNN (Feng et al., 2019), MeanPoolingConv (Lee & Shin, 2023a), and SetGNN (Chien et al., 2022). Notably, the latter two encoders, MeanPoolingConv and SetGNN, serve as backbone encoders in TriCL (Lee & Shin, 2023a) and HyperGCL (Wei et al., 2022), respectively. The experimental setting mirrors that of the fine-tuning experiments detailed in Section 5.1. As shown in Table 7, HYPEBOY outperforms other hypergraph SSL methods in most of the settings, validating its broad effectiveness across a diverse range of HNNs.

## F.5 PRETEXT TASK ANALYSIS

In this section, we evaluate the efficacy of different SSL pretext tasks. Specifically, we assess our hyperedge filling task by comparing it with two other SSL tasks: contrastive learning and feature reconstruction. To ensure a fair comparison and minimize the impact of the underlying SSL method design choices, we streamline each method as described below:

- Naive contrastive learning: This method is a variant of TriCL (Lee & Shin, 2023a) that does not use projection head parameters. Other settings are all the same as that of TriCL (e.g., contrastive losses and view generation process).

- Naive feature reconstruction: This method is a variant of the hypergraph feature reconstruction method (Section 4.4), that does not use a decoder HNN.

- Naive hyperedge filling: This method is a variant of HYPEBOY that does not use feature reconstruction warm-up and projection heads. Specifically, this method is equivalent to **V1**, which is described in Section 5.3.

As shown in Table 8, the naive hyperedge filling method outperforms the other two methods across all settings, highlighting the superior effectiveness of the hyperedge filling task.

## F.6 TOPOLOGICAL GENERATIVE PRETEXT TASK ANALYSIS

In this section, we compare HYPEBOY with other topology generative SSL methods. To this end, we adopt the following three baseline methods:

- Graph edge filling: Utilizes edge filling on a clique-expanded graph (Appendix E.2), employing the same method HYPEBOY used to fill size-2 hyperedges.

- Graph hyperedge filling: Applies hyperedge filling on a clique-expanded graph (Appendix E.2), using the same approach as HYPEBOY for filling hyperedges.

- Hyperedge prediction: Undertakes the hyperedge prediction task as an SSL task, leveraging size negative samples (Patil et al., 2020).

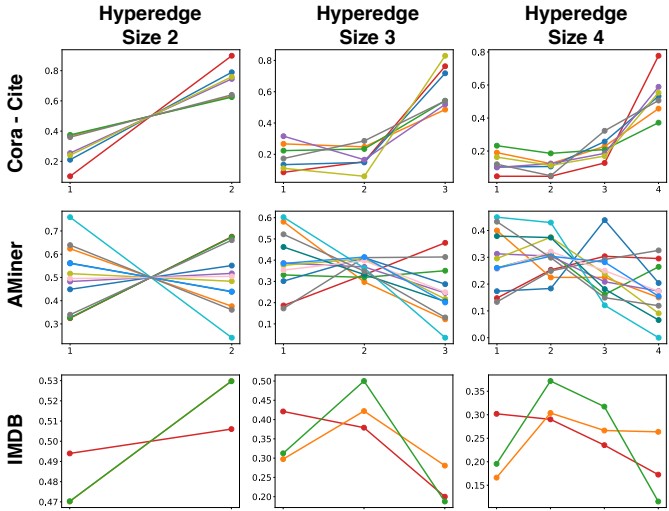

Figure 6: Homophily analysis of three benchmark datasets: Cora-Cite, AMiner, and IMDB. Each color represents the class (label) of nodes. An upward trend in a class indicates homophily, whereas a downward trend indicates non-homophily. Cora-Cite demonstrates homophily, IMDB shows non-homophily, and AMiner falls in between, with some classes exhibiting homophily and others not.

As shown in Table 9, HYPEBOY outperforms the other three topological generative SSL methods in all settings, highlighting the effectiveness of the hyperedge filling task.

### F.7    HOMOPHILY ANALYSIS

In this section, we demonstrate that HYPEBOY exhibits prominent node classification performance across various homophilic scenarios, including both homophilic and non-homophilic hypergraphs. To this end, we analyze both real-world hypergraphs and semi-synthetic hypergraphs.

**Real-world hypergraphs.** We analyze the homophily characteristic of three hypergraph benchmark datasets: Cora-Cite, AMiner, and IMDB, which are utilized in this work. Note that HYPEBOY shows the best performance among all the methods in these datasets (refer to Table 1).

For the homophily analysis, we utilize the method suggested by Veldt et al. (2023), which is a visualization-based approach. The analysis should be conducted for each hyperedge size, with the results for each hyperedge size presented in separate plots. In these plots, the line color represents the node class. A brief interpretation of the resulting plots is as follows:

- An upward trend line indicates that nodes of the corresponding class demonstrate higher-order homophilic characteristics within hyperedges of the same size.

- A downward trend line indicates that nodes of the corresponding class display higher-order non-homophilic characteristics within hyperedges of the same size.

As shown in Figure 6, the three datasets (Cora-cite, AMiner, and IMDB) exhibit different homophilic patterns. Specifically, the Cora-Cite dataset demonstrates homophilic tendencies, the IMDB dataset shows a non-homophilic pattern, and the AMiner dataset falls in between, with some classes being homophilic and others not. Given that HYPEBOY outperforms other methods in these datasets, we can conclude that its efficacy is valid for both homophilic and non-homophilic hypergraphs.

**Semi-synthetic hypergraphs.** We analyze how the performance of HYPEBOY varies as the homophilic extent of hypergraphs varies. To this end, we corrupt the hypergraph topology of two homophilic hypergraph datasets: Cora and Citeseer. Specifically, we utilize the node-swapping method. Details are provided in Algorithm 1. Note that the more we swap nodes in a hypergraph, the more the hypergraph gets non-homophilic.

Table 10: Analysis of performance changes under variations in homophilic extent. The smallest drop from the original performance is highlighted in green. NA indicates UniGCNII encoder without pre-training. As shown, HYPEBOY is the most robust method against homophily violation among the used hypergraph SSL methods.

| Dataset | Method | Original | $T = 50$ | $T = 100$ | $T = 150$ | $T = 200$ |
|---|---|---|---|---|---|---|
| Cora | NA | 48.5 (-) | 46.7 (-1.8) | 44.7 (-3.8) | 43.5 (-5.0) | 40.9 (-7.6) |
| | TriCL | 60.2 (-) | 59.1 (-1.1) | 56.1 (-4.1) | 55.7 (-4.5) | 53.6 (-6.5) |
| | HyperGCL | 60.3 (-) | 57.4 (-2.9) | 53.7 (-6.6) | 52.0 (-8.3) | 50.0 (-10.3) |
| | HYPEBOY | 62.3 (-) | 61.7 (-0.6) | 59.2 (-3.1) | 58.0 (-4.3) | 56.7 (-5.6) |
| Citeseer | NA | 44.2 (-) | 41.4 (-2.8) | 39.0 (-5.2) | 35.9 (-8.3) | 33.8 (-10.4) |
| | TriCL | 51.7 (-) | 51.1 (-0.6) | 47.5 (-4.2) | 45.2 (-6.5) | 41.8 (-9.9) |
| | HyperGCL | 47.0 (-) | 45.8 (-1.2) | 43.3 (-3.7) | 40.9 (-6.1) | 37.7 (-9.3) |
| | HYPEBOY | 56.7 (-) | 56.5 (-0.2) | 54.2 (-2.5) | 50.8 (-5.9) | 48.6 (-8.1) |

---

**Algorithm 1** Node swapping algorithm

**Input:** A set of hyperedges $\mathcal{E}$, Number of iterations $T$
**Output:** Shuffled hyperedges $\mathcal{E}'$

1: $\mathcal{E}' \leftarrow \mathcal{E}$
2: **for** $k \leftarrow 1$ to $T$ **do**
3:      Sample two hyperedges $e'_i$ and $e'_j$ from $\mathcal{E}'$ uniformly at random
4:      Sample a node $v'_i$ from $e'_i$ and a node $v'_j$ from $e'_j$ uniformly at random
5:      $e''_i \leftarrow (e'_i \setminus \{v'_i\}) \cup \{v'_j\}$
6:      $e''_j \leftarrow (e'_j \setminus \{v'_j\}) \cup \{v'_i\}$
7:      $\mathcal{E}' \leftarrow (\mathcal{E}' \setminus \{e'_i\} \setminus \{e'_j\}) \cup \{e''_i\} \cup \{e''_j\}$
**Return:** Swapped hyperedges $\mathcal{E}'$

---

As shown in Table 10, HYPEBOY is the most robust among the hypergraph SSL methods evaluated, exhibiting the smallest performance drop in scenarios where homophily is violated.

## F.8 ADDITIONAL DATA SPLITS

In this section, we investigate the efficacy of HYPEBOY in various node-label-split settings. To this end, we sample a certain number of nodes for each class as training nodes and validation nodes. Specifically, we utilize the following two settings:

- For each node class, use 5 nodes/5 nodes/the rest as train/validation/test nodes, respectively.

- For each node class, use 10 nodes/10 nodes/the rest as train/validation/test nodes, respectively.

Other experimental settings are the same as that of the fine-tuning experiments (Section 5.1). As shown in Table 11, HYPEBOY outperforms other hypergraph SSL methods in all the settings. Thus, we demonstrate that the efficacy of HYPEBOY is not restricted to a particular label split setting.

## F.9 HETEROGENOUS HYPERGRAPH EXPERIMENT

In this section, we investigate the efficacy of HYPEBOY as a pre-training strategy on the heterogeneous hypergraph dataset, which contains hyperedges of various semantics. To this end, we utilize the ACM dataset, where nodes correspond to publications. There are two types of hyperedges: authorship and subject relations.

- Authorship: Publications authored by the same person are represented as a hyperedge.

- Subject relations: Publications sharing a subject are represented as a hyperedge.

Other experimental settings are the same as that of the fine-tuning experiments (Section 5.1). As shown in Table 12, HYPEBOY outperforms other hypergraph SSL methods in the ACM dataset, demonstrating its efficacy beyond homogeneous hypergraphs.

Table 11: Results in the various label-split settings under the **fine-tuning protocol**. The best and second-best performances are colored green and yellow, respectively. NA indicates UniGCNII encoder without pre-training. As shown, HYPEBOY outperforms other hypergraph SSL methods in all the settings.

|  | Method | Citeseer | Cora | Pubmed | Cora-CA | DBLP-P |
|---|---|---|---|---|---|---|
| 5 nodes | NA | 53.7 (6.8) | 63.7 (3.8) | 71.6 (4.6) | 60.7 (4.9) | 77.8 (4.7) |
|  | TriCL | 60.1 (5.2) | 71.0 (3.1) | 71.9 (3.8) | 67.6 (2.9) | 80.5 (2.5) |
|  | HyperGCL | 57.0 (5.6) | 70.3 (2.9) | 72.2 (4.4) | 61.5 (4.4) | 79.5 (3.7) |
|  | HYPEBOY | 63.1 (4.6) | 72.0 (2.4) | 72.9 (3.2) | 67.8 (2.4) | 81.5 (2.4) |
| 10 nodes | NA | 62.7 (3.8) | 72.8 (2.2) | 73.7 (3.0) | 67.7 (3.0) | 82.4 (1.9) |
|  | TriCL | 66.3 (2.6) | 75.7 (2.3) | 74.5 (3.1) | 69.7 (2.2) | 83.6 (1.9) |
|  | HyperGCL | 64.4 (3.9) | 74.9 (2.1) | 74.3 (3.1) | 69.0 (3.4) | 83.6 (1.6) |
|  | HYPEBOY | 67.7 (2.6) | 75.8 (7.8) | 74.9 (2.7) | 70.9 (1.7) | 84.6 (1.5) |

Table 12: Results in the ACM dataset under the **fine-tuning protocol**. The best performances are colored green. Among the four methods, HYPEBOY shows the best performance.

| Method | UniGCNII | TriCL | HyperGCL | HYPEBOY |
|---|---|---|---|---|
| Accuracy | 40.0 (3.1) | 40.3 (2.4) | 40.8 (3.4) | 41.7 (2.6) |

# G   POTENTIAL APPLICATIONS OF HYPEREDGE FILLING

In this section, we provide two potential applications of the hyperedge filling task.

- **Email recipient recommendation**: Given recipients of an email, recommend a user likely to be added as a recipient of the email. Nodes in a hypergraph indicate users, and each hyperedge indicates an email, consisting of a sender, recipients, and CCs.
- **Item recommendation**: Given items in a shopping cart recommend an item to be co-purchased with the items. Nodes in a hypergraph indicate items, and each hyperedge contains a set of co-purchased items.

