# HypeBoy: Generative Self-Supervised Representation Learning on Hypergraphs

## Abstract

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

**Hypergraph neural networks (HNNs).** HNNs learn hypergraph representation. Converting hyperedges into cliques (fully connected subgraphs) allows graph neural networks to be applied to hypergraphs (Feng et al., 2019; Yadati et al., 2019). Such conversion, however, may result in topological information loss, since high-order interactions (hyperedges) are reduced to pair-wise interactions (edges). As such, most HNNs pass messages through hyperedges to encode hypergraphs. Some notable examples include HNHN (Dong et al., 2020) with a hyperedge encoder, UniGNN (Huang & Yang, 2021) with generalized message passing for graphs and hypergraphs, AllSet (Chien et al., 2022) with a set encoder, ED-HNN (Wang et al., 2023a) with permutation-equivariant diffusion operators, and PhenomNN (Wang et al., 2023b) with hypergraph-regularized energy functions. ***Note***: The discussed HNNs are hypergraph encoders, with no dedication to a particular loss objective. We clarify that our interest is in designing a generative SSL strategy for hypergraphs, not new encoders.

**Self-supervised learning (SSL).** SSL strategies aim to learn representation from the input data itself, without relying on external labels. They can largely be categorized into contrastive or generative types. Contrastive SSL aims to maximize the agreement between data obtained from diverse views (Chen et al., 2020; Grill et al., 2020; You et al., 2020). Generative SSL, on the other hand, predicts or reconstructs parts of the input data. Success of generative SSL demonstrates its strengths in learning complex input data, in domains including natural language processing (Devlin et al., 2019; OpenAI, 2023) and computer vision (He et al., 2022; Tong et al., 2022). Recently, generative SSL for graphs has gained significant attention, with their main focuses on reconstructing edges (Tan et al., 2023; Li et al., 2023) or node features (Hou et al., 2022; 2023). ***Note***: Extending feature reconstruction from graphs to hypergraphs can be direct, which serves as our baseline method. However, it is non-trivial to extend the edge reconstruction methods for hyperedges (refer to Section 3.1).

**Self-supervised learning on hypergraphs.** The interest in SSL for hypergraphs is on the rise. Early hypergraph SSL strategies mainly targeted specific downstream tasks, such as group (Zhang et al., 2021) and session-based recommendation (Xia et al., 2022). Recent ones aim to obtain general-purpose representation. TriCL (Lee & Shin, 2023) utilizes a tri-directional contrastive loss, which consists of node-, hyperedge-, and membership-level contrast. Kim et al. (2023) enhances the scalability of TriCL with a partitioning technique. HyperGCL (Wei et al., 2022) generates views for contrast and empirically demonstrates its superiority over rule-based augmentation methods. ***Note***: (1) All the hypergraph SSL strategies are contrastive (rather than generative) and (2) no prior works establish a clear theoretical connection between their SSL strategies and downstream tasks.

## 3 PROPOSED TASK AND THEORETICAL ANALYSIS

In this section, after providing some preliminaries, we propose a novel generative SSL task on hypergraphs, *hyperedge filling*. Then, we establish a theoretical connection between hyperedge filling and node classification, which is a commonly-considered important downstream task.

**Preliminaries.** A hypergraph $\mathcal{G} = (\mathcal{V}, \mathcal{E})$ is defined by a node set $\mathcal{V}$ and a hyperedge set $\mathcal{E}$. Each hyperedge $e_j \in \mathcal{E}$ is a non-empty set of nodes (i.e., $\emptyset \neq e_j \subseteq \mathcal{V}, \forall e_j \in \mathcal{E}$). Each node $v_i \in \mathcal{V}$ is

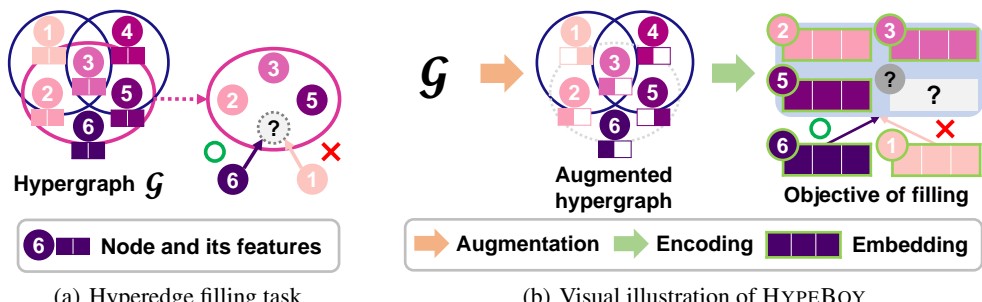

(a) Hyperedge filling task        (b) Visual illustration of HYPEBOY

Figure 1: Overview of (a) the hyperedge filling task and (b) HYPEBOY, the proposed SSL method based on the task. The goal of the task is to find the missing node for a given query subset (i.e., the other nodes in a hyperedge). HYPEBOY trains HNNs aiming to correctly predict the missing node.

equipped with a feature vector $\boldsymbol{x}_i \in \mathbb{R}^d$, and $\mathbf{X} \in \mathbb{R}^{|\mathcal{V}| \times d}$ denotes the node feature matrix where the $i$-th row $\mathbf{X}_i$ corresponds to $\boldsymbol{x}_i$.

A hypergraph neural network (HNN) $f_\theta$ is a function that receives a node feature matrix $\mathbf{X}$ and a set of hyperedges $\mathcal{E}$ as inputs to return nodes embeddings $\mathbf{Z} \in \mathbb{R}^{|\mathcal{V}| \times k}$ (i.e., $\mathbf{Z} = f_\theta(\mathbf{X}, \mathcal{E})$, where $\theta$ is the set of learnable parameters).[1] *Note*: Hypergraph self-supervised learning (SSL) aims to train $f_\theta$ by utilizing only $\mathbf{X}$ and $\mathcal{G}$, without any supervision from downstream task-related labels.

## 3.1 PROPOSED TASK: HYPEREDGE FILLING

We propose *hyperedge filling*, a generative SSL task for hypergraph representation learning. We first define the task and discuss the superiority of the proposed task over alternatives. An illustration of the hyperedge filling task is provided in Figure 1(a).

**Task definition.** Given a set of nodes, hyperedge filling aims to predict a node that is likely to form a hyperedge with it. Specifically, for each hyperedge $e_j \in \mathcal{E}$, we divide it into a (missing) node $v_i \in e_j$ and a (query) subset $q_{ij} = e_j \setminus \{v_i\}$. Then, the target of the task is to correctly fill the missing node $v_i$ for each given subset $q_{ij}$. This can be formalized by maximizing the probability of $v_i$ correctly completing $q_{ij}$, which is denoted as $p_{(\mathbf{X}, \mathcal{E}, \Theta)}(v_i \mid q_{ij})$, where $\Theta$ is a set of parameters we aim to optimize in this task. We will further elaborate on our design of $p_{(\mathbf{X}, \mathcal{E}, \Theta)}(\cdot)$ in Section 4.3.

**Advantage over alternatives.** Potential alternatives include naive extensions of generative SSL tasks for ordinary graphs: (a) generating hyperedges from scratch and (b) classifying given sets of nodes into real and fake hyperedges. Compared to (a), by shifting the focus of prediction from the set level (hyperedge itself) to the node level, the hyperedge filling task reduces the prediction space from computationally prohibitive $O(2^{|\mathcal{V}|})$ to affordable $O(|\mathcal{V}|)$. Compared to (b), the hyperedge filling task provides richer and more diversified generative SSL signals. Specifically, when considering a single hyperedge $e_j$, our task offers $|e_j|$ distinct node-subset combinations that can serve as SSL signals. In contrast, predicting the mere existence of $e_j$ yields a singular and, thus, limited signal.

## 3.2 THEORETICAL RESULTS ON HYPEREDGE FILLING

To demonstrate the effect of hyperedge filling as a general SSL task, we present its theoretical connection to node classification. In essence, we demonstrate that node representations optimized for the hyperedge filling task can improve node classification accuracy. Then, we briefly discuss the theoretical difficulty of the task, showing that it has reasonable solutions for every hypergraph.

### 3.2.1 BASIC SETTING

First, we assume a data model of a hypergraph $\mathcal{G} = (\mathcal{V}, \mathcal{E})$, where (1) each node belongs to a single class, (2) the features of each node are generated from a Gaussian distribution, and (3) each hyper-

---

[1]In this paper, we assume HNNs return only vector representations of nodes unless otherwise stated, while we acknowledge that some HNNs return embeddings of hyperedges as well.

edge is generated according to a given homophilic ratio $\mathcal{P} \in [0.5, 1]$.[2] Specifically, if a hyperedge has a higher homophilic ratio $\mathcal{P}$, the hyperedge is more likely to contain nodes of the same class.

**Assumption 1** (Node classes and features). *Assume that there are $2N$ nodes and node classes $C_1$ and $C_0$ such that $C_1 \cup C_0 = \mathcal{V}, C_1 \cap C_0 = \emptyset$, and $|C_1| = |C_0| = N$. Each node feature vector $\boldsymbol{x}_i$ is independently generated from $\mathcal{N}(\boldsymbol{x}; \boldsymbol{\mu}_1, \boldsymbol{\Sigma})$ if $v_i \in C_1$, and $\mathcal{N}(\boldsymbol{x}; \boldsymbol{\mu}_0, \boldsymbol{\Sigma})$ if $v_i \in C_0$. For simplicity, we assume $\boldsymbol{\mu}_1 = (0.5)_{i=1}^d$, $\boldsymbol{\mu}_0 = (-0.5)_{i=1}^d$, and $\boldsymbol{\Sigma} = \mathbf{I}$, where $\mathbf{I}$ is the $d$-by-$d$ identity matrix.*

**Assumption 2** (Hypergraph topology). *Assume that the number of hyperedges and the size of each hyperedge are given. There is no singleton hyperedge (i.e., $|e_j| \geq 2, \forall e_j \in \mathcal{E}$). Let $B$ denote the binomial distribution. Each hyperedge $e_j \in \mathcal{E}$ has a membership $c_j \in \{0, 1\}$, where $c_j \sim B(1, 0.5)$. Given the number $|e_j|$ of nodes and the class $c_j \in \{0, 1\}$ of a hyperedge, the number of its members belonging to $C_1$ satisfies $|e_j \cap C_1| \sim B(|e_j|, \mathcal{P}^{c_j}(1 - \mathcal{P})^{1-c_j})$.*

Second, we describe how node representations are updated via the hyperedge filling task. In this theoretical analysis, we define the updating process of node representations as follows:

(**F1**) Filling probability $p_{(\mathbf{X}, \mathcal{E}, \Theta)}(\cdot)$ is defined on each node-subset pair as follows:

$$p_{(\mathbf{X}, \mathcal{E}, \Theta)}(v_i \mid q_{ij}) := \frac{\exp(\boldsymbol{x}_i^T(\sum_{v_k \in q_{ij}} \boldsymbol{x}_k))}{\sum_{v_t \in \mathcal{V}} \exp(\boldsymbol{x}_t^T(\sum_{v_k \in q_{ij}} \boldsymbol{x}_k))}. \tag{1}$$

(**F2**) Node representation $\boldsymbol{z}_i$ is obtained via gradient descent with respect to $\boldsymbol{x}_i$ from $\mathcal{L}$, which is the negative log-likelihood of Eq. (1), (i.e., $\mathcal{L} = -\log p_{(\mathbf{X}, \mathcal{E}, \Theta)}(v_i \mid q_{ij})$). For ease of analysis, we assume $\boldsymbol{z}_i = \boldsymbol{x}_i - \gamma \nabla_{\boldsymbol{x}_i} \mathcal{L}$, where $\gamma \in \mathbb{R}^+$ is a fixed constant.

At last, we assume a Gaussian naive Bayes classifier $\mathcal{F}$ (Bishop, 2006), which is defined as:

$$\mathcal{F}(\boldsymbol{x}_i) = \arg\max_{k \in \{0,1\}} f(\boldsymbol{x}_i; \boldsymbol{\mu}_k, \mathbf{I}), \text{ where } f \text{ is the P.D.F. of } \mathcal{N}(\boldsymbol{x}; \boldsymbol{\mu}_k, \Sigma). \tag{2}$$

### 3.2.2 HYPEREDGE FILLING HELPS NODE CLASSIFICATION

Our goal is to show that for accurate classification of $v_i$, the representation $\boldsymbol{z}_i$, which is obtained for hyperedge filling as described in (**F1**) and (**F2**), is more effective than the original feature $\boldsymbol{x}_i$. First, we assume a node $v_i$ belonging to the class $C_1$ (i.e., $v_i \in C_1$), and we later generalize the result to $C_0$. Then, the effectiveness of an original feature is defined as the expected accuracy of a classifier $\mathcal{F}$ with $\boldsymbol{x}_i$ (i.e., $\mathbb{E}_{\boldsymbol{x}}[\mathbf{1}_{\mathcal{F}(\boldsymbol{x}_i)=1}] := P_{\boldsymbol{x}}(f(\boldsymbol{x}_i; \boldsymbol{\mu}_1, \mathbf{I}) > f(\boldsymbol{x}_i; \boldsymbol{\mu}_0, \mathbf{I}))$). Similarly, that with a derived representation is defined as $\mathbb{E}_{\boldsymbol{z}}[\mathbf{1}_{\mathcal{F}(\boldsymbol{z}_i)=1}] = P_{\boldsymbol{z}}(f(\boldsymbol{z}_i; \boldsymbol{\mu}_1, \mathbf{I}) > f(\boldsymbol{z}_i; \boldsymbol{\mu}_0, \mathbf{I}))$. Below, we show that the effectiveness of a derived representation $\boldsymbol{z}_i$ is higher than that of an original feature $\boldsymbol{x}_i$.

**Theorem 1** (Improvement in effectiveness). *Assume a hyperedge $e_j$ s.t. $e_j \cap C_1 \neq \emptyset$ and node features $\mathbf{X}$ that are generated under Assumption 1. For a node $v_i \in e_j \cap C_1$, the following holds:*

$$\boxed{\vec{\mathbf{1}}^T \sum_{v_k \in q_{ij}} \boldsymbol{x}_k > 0} \Rightarrow \mathbb{E}_{\boldsymbol{z}}[\mathbf{1}_{\mathcal{F}(\boldsymbol{z}_i)=1}] > \mathbb{E}_{\boldsymbol{x}}[\mathbf{1}_{\mathcal{F}(\boldsymbol{x}_i)=1}], \text{ where } \vec{\mathbf{1}} \text{ denotes } (1)_{k=1}^d. \tag{3}$$

*Proof.* Full proof is provided in Appendix A.1. $\square$

Theorem 1 states that when a certain condition (boxed in Eq. (3)) is met, the effectiveness of $\boldsymbol{z}_i$ is greater than that of $\boldsymbol{x}_i$. This result implies that node representations, when refined using the objective function associated with the hyperedge filling task, are more proficient in performing accurate node classification compared to the original node features.

While the finding in Theorem 1 demonstrates the usefulness of the hyperedge filling task in node classification, its validity relies on the specific condition. We further analyze the probability that the condition is met by a stochastic $G$ under Assumptions 1 and 2 for a given $\mathcal{P}$.

**Theorem 2** (Realization of condition). *Assume node features $\mathbf{X}$ and a hyperedge $e_j$ s.t. (i) generated under Assumption 1 and 2 respectively, and (ii) $e_j \cap C_1 \neq \emptyset$. For any $q_{ij}$ where $v_i \in C_1 \cap e_j$, the following holds:*

---

[2]We set $\mathcal{P} \in [0.5, 1]$ because generation process (Assumption 2) is symmetric about $\mathcal{P} = 0.5$ at $\mathcal{P} \in [0, 1]$ under binary class setting.

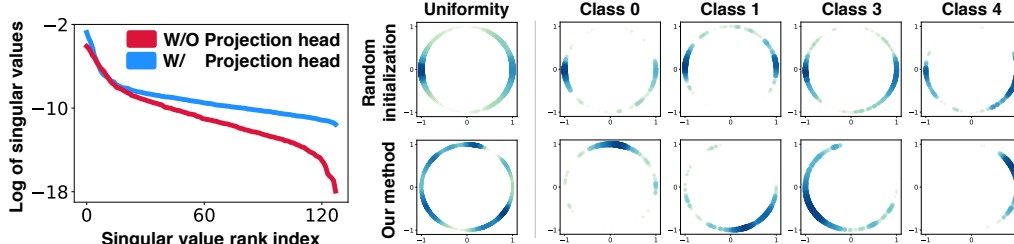

(a) Representation spectrum analysis. A sudden singular-value drop implies **dimensional collapse**.

(b) Representations on the unit hypersphere. Uniformly distributed representations achieve **uniformity**, and representations of nodes of the same class located close to each other achieve **alignment**.

Figure 2: Analysis regarding Property 2 (avoiding dimensional collapse) and Property 3 (representation uniformity and alignment) of HYPEBOY. As shown in (a), while HYPEBOY without projection heads (red) suffers from the dimensional collapse, HYPEBOY (blue) does not, demonstrating the necessity of the projection head. Furthermore, as shown in (b), representations from an HNN trained by HYPEBOY meet both uniformity and alignment, justifying our design choice of the loss function. Experiments are conducted on the Cora dataset.

1. $P_{\boldsymbol{x},e}\left(\mathbf{1}^T \sum_{v_k \in q_{ij}} \boldsymbol{x}_k > 0 \mid \mathcal{P}\right) \geq 0.5, \forall \mathcal{P} \in [0.5, 1]$.

2. $P_{\boldsymbol{x},e}\left(\mathbf{1}^T \sum_{v_k \in q_{ij}} \boldsymbol{x}_k > 0 \mid \mathcal{P}\right)$ *is a strictly increasing function w.r.t.* $\mathcal{P} \in [0.5, 1]$.

*Proof.* Full proof is provided in Appendix A.2. □

Theorem 2 states that the probability of the condition being satisfied is at least 0.5, if the homophilic ratio $\mathcal{P}$ is at least 0.5. Moreover, the likelihood of satisfying the condition strictly increases with respect to $\mathcal{P}$. Notably, many real-world group interactions exhibit homophilic traits (Laakasuo et al., 2020; Khanam et al., 2023). Therefore, the hyperedge filling task can improve node classification in many real-world scenarios, as evidenced by our theoretical findings and real-world characteristics.

**Generalization to the class** $C_0$**.** The above results can be easily generalized to the class $C_0$ due to the symmetry. Specifically, for each node $v_i \in C_0$, the effectiveness (spec., the expected accuracy of the classifier $\mathcal{F}$) of a derived representation $\boldsymbol{z}_i$ is greater than that of an original feature $\boldsymbol{x}_i$ under a certain condition. The probability of such a condition holding strictly increases from $0.5$ with respect to the homophilic ratio $\mathcal{P} \in [0.5, 1]$. We theoretically show this in Appendices A.1 and A.2.

### 3.2.3 EXISTENCE OF REASONABLE SOLUTIONS

If a task is too difficult, it could be excessively challenging for a model to learn meaningful representations from the task. Fortunately, with a sufficiently large embedding dimension, our hyperedge filling task has "reasonable" solutions for every hypergraph, as shown theoretically in Appendix B.1.

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

### 4.4 Two-stage training scheme for further enhancement

We propose a novel two-stage training scheme to further enhance the effectiveness of HYPEBOY.

**Challenges: heavy reliance on projection heads.** In our preliminary studies, we observed that during training, HYPEBOY often relies heavily on projection heads rather than the parameters of encoder HNNs. Consequently, HNNs are pre-trained suboptimally.

**Solution: warming-up encoders via feature reconstruction.** To reduce this reliance on projection heads, we introduce a *warm-up* training stage. Firstly, we train the parameters (i.e., the parameter of an encoder HNN and projection heads), aiming for node-feature reconstruction, where projection heads play a less prominent role. Then, the parameters of the encoder HNN (excluding the projection heads) are employed to initialize the HNN encoder of HYPEBOY. This initialization strengthens the HNN encoders, thereby reducing the reliance on projection heads. For details and effectiveness of this warm-up stage, refer to Appendix D.3 and Section 5.3, respectively.

## 5 Experimental Results

We now evaluate the efficacy of HYPEBOY as techniques for (1) pre-training hypergraph neural networks (HNNs) for node classification (Section 5.1) and (2) learning general-purpose representations (Section 5.2). Then, we justify each of its component through an ablation study (Section 5.3).

**Datasets.** For experiments, we use 11 benchmark hypergraph datasets. The hypergraph datasets are from diverse domains, expressing co-citation, co-authorship, computer graphics, movie-actor, news, and political membership relations. In Appendix C, we detail their statistics and descriptions.

**Baselines methods.** We utilize 15 baseline methods. They include (a) 10 *(semi-)supervised HNNs*, including ED-HNN (Wang et al., 2023a) and PhenomNN (Wang et al., 2023b), (b) 2 latest *generative SSL* strategies for ordinary graphs (GraphMAE2 (Hou et al., 2023) and MaskGAE (Liu et al., 2022)), and (3) 3 *contrastive SSL* strategies for hypergraph (TriCL (Lee & Shin, 2023), HyperGCL (Wei et al., 2022), and H-GD, which is a direct extension of a graph SSL method (Zheng et al., 2022) to hypergraphs). We use UniGCNII (Huang & Yang, 2021) and GCN (Kipf & Welling, 2017) as the encoders for hypergraph SSL methods and graph SSL methods, respectively.[3] In Appendix D, we provide their details, including their implementations, training, and hyperparameters.

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

---

[5]Representing higher-order interactions with a graph can cause the information loss (Dong et al., 2020).

Table 2: Efficacy as general-purpose embedding techniques: AVG and STD of accuracy/AUROC values in node-classification/hyperedge-prediction under the **linear evaluation protocol**. In each downstream task, the best and second-best performances are colored green and yellow, respectively. A.R. denotes the average ranking among all methods. O.O.T. means that training is not completed within 24 hours. HYPEBOY obtains the best average ranking in both downstream tasks.

| | Method | Citeseer | Cora | Pubmed | Cora-CA | DBLP-P | DBLP-A | AMiner | IMDB | MN-40 | 20News | House | A.R. |
|---|---|---|---|---|---|---|---|---|---|---|---|---|---|
| Node classification | Naive **X** | 27.8 (7.0) | 32.4 (4.6) | 62.8 (2.8) | 31.9 (5.5) | 69.4 (0.7) | 54.7 (4.7) | 21.4 (1.2) | 38.1 (1.9) | 91.9 (1.1) | 70.6 (1.9) | 71.3 (5.4) | 5.0 |
| | GraphMAE2 | 29.2 (6.5) | 37.5 (7.0) | 55.5 (9.5) | 38.2 (9.1) | 75.6 (1.7) | 57.5 (5.6) | 27.3 (2.7) | 36.6 (3.5) | 89.1 (1.8) | 62.3 (2.3) | 51.7 (3.5) | 5.6 |
| | MaskGAE | 47.2 (11.1) | 56.8 (9.3) | 62.6 (5.5) | 56.0 (4.8) | 84.8 (0.7) | 75.1 (3.5) | 33.2 (2.0) | 44.1 (3.9) | 90.5 (0.9) | O.O.T. | 50.0 (2.8) | 4.4 |
| | TriCL | 53.3 (10.0) | 62.1 (8.8) | 74.5 (4.1) | 63.6 (5.2) | 87.1 (0.7) | 80.9 (3.2) | 35.0 (3.6) | 48.0 (3.2) | 80.0 (5.1) | 67.2 (4.0) | 69.1 (5.5) | 2.5 |
| | HyperGCL | 42.6 (8.6) | 61.8 (8.3) | 67.6 (8.0) | 58.1 (6.3) | 56.6 (5.2) | 79.8 (3.8) | 33.3 (2.2) | 47.5 (2.8) | 84.1 (2.8) | 71.2 (3.4) | 67.1 (5.4) | 3.8 |
| | H-GD | 35.6 (7.8) | 37.6 (6.8) | 58.0 (8.2) | 48.6 (7.4) | 73.3 (1.3) | 74.0 (3.3) | 33.8 (5.0) | 35.2 (2.9) | 76.6 (4.4) | 54.8 (7.4) | 68.3 (5.7) | 5.3 |
| | HYPEBOY | 59.6 (9.9) | 63.5 (9.4) | 75.0 (3.4) | 66.0 (4.6) | 87.9 (0.5) | 81.2 (2.7) | 34.3 (3.2) | 48.8 (1.8) | 89.2 (2.2) | 75.7 (2.1) | 69.4 (5.4) | 1.4 |
| Hyperedge prediction | Naive **X** | 63.3 (2.1) | 75.5 (1.6) | 88.3 (0.6) | 55.0 (1.9) | 90.0 (0.4) | 72.1 (1.3) | 80.0 (1.1) | 39.5 (1.9) | 99.5 (0.1) | 97.7 (2.9) | 54.8 (5.0) | 5.8 |
| | GraphMAE2 | 73.3 (2.7) | 76.4 (1.7) | 81.6 (1.1) | 76.3 (3.1) | 85.2 (0.4) | 68.3 (1.8) | 80.7 (0.9) | 53.7 (2.6) | 99.5 (0.1) | 90.1 (5.7) | 62.9 (3.8) | 5.5 |
| | MaskGAE | 86.1 (1.6) | 88.5 (1.4) | 92.9 (0.5) | 81.8 (2.7) | 93.2 (0.5) | 79.3 (2.0) | 84.6 (0.1) | 58.1 (2.5) | 99.3 (0.1) | O.O.T. | 87.0 (3.4) | 4.0 |
| | TriCL | 90.5 (1.2) | 90.7 (1.3) | 91.9 (0.5) | 87.8 (1.5) | 94.8 (0.2) | 87.9 (1.4) | 90.4 (0.6) | 58.9 (2.1) | 99.6 (0.1) | 98.2 (3.0) | 90.0 (2.6) | 1.8 |
| | HyperGCL | 73.9 (2.6) | 85.4 (1.5) | 89.6 (0.5) | 81.1 (1.9) | 83.6 (0.6) | 83.5 (1.0) | 82.1 (7.6) | 53.8 (2.4) | 99.4 (0.1) | 96.7 (0.7) | 76.3 (6.3) | 4.5 |
| | H-GD | 72.2 (5.0) | 71.9 (3.1) | 87.2 (0.7) | 73.2 (4.0) | 91.6 (1.0) | 81.4 (1.9) | 84.9 (2.1) | 53.1 (1.8) | 99.5 (0.1) | 83.9 (2.1) | 87.9 (3.1) | 4.9 |
| | HYPEBOY | 91.1 (1.1) | 91.9 (1.1) | 95.1 (0.3) | 88.1 (1.4) | 95.5 (0.1) | 87.3 (1.3) | 89.8 (0.5) | 59.4 (2.1) | 99.7 (0.1) | 99.0 (1.6) | 87.0 (2.8) | 1.4 |

Table 3: The ablation study with four variants of HYPEBOY on node classification under the fine-tuning protocol. The best and second-best performances are colored green and yellow, respectively. F.R., H.F., and P.H. denote Feature Reconstruction, Hyperedge Filling, and Projection Heads, respectively. A.R. denotes the average ranking among all methods. HYPEBOY outperforms others in most datasets, justifying each of its components.

| | F. R. | H. F. | P. H. | Citeseer | Cora | Pubmed | Cora-CA | DBLP-P | DBLP-A | AMiner | IMDB | MN-40 | 20News | House | A.R. |
|---|---|---|---|---|---|---|---|---|---|---|---|---|---|---|---|
| **V1** | ✗ | ✔ | ✗ | 51.6 (11.2) | 60.7 (8.2) | 76.2 (3.6) | 63.5 (0.6) | 88.1 (0.5) | 78.5 (2.9) | 33.5 (2.8) | 46.8 (3.1) | 90.0 (1.1) | 77.4 (0.9) | 68.5 (4.5) | 4.1 |
| **V2** | ✗ | ✔ | ✔ | 52.7 (9.6) | 59.7 (9.2) | 76.7 (3.2) | 63.5 (0.6) | 88.2 (0.5) | 79.1 (2.5) | 33.8 (2.2) | 46.9 (3.3) | 90.6 (1.0) | 77.0 (0.9) | 69.6 (4.9) | 3.2 |
| **V3** | ✔ | ✗ | ✗ | 52.0 (9.3) | 58.9 (8.2) | 74.1 (3.9) | 61.2 (6.6) | 87.8 (0.4) | 79.9 (2.3) | 33.9 (2.1) | 46.3 (2.7) | 91.4 (0.9) | 77.5 (0.9) | 70.1 (4.8) | 3.6 |
| **V4** | ✔ | ✔ | ✗ | 56.0 (9.9) | 61.8 (8.5) | 76.5 (3.1) | 65.3 (4.3) | 88.0 (0.4) | 80.3 (2.4) | 34.0 (2.0) | 47.5 (2.3) | 90.8 (1.0) | 77.4 (1.0) | 69.3 (5.0) | 2.5 |
| **Ours** | ✔ | ✔ | ✔ | 56.7 (9.8) | 62.3 (7.7) | 77.0 (3.4) | 66.3 (4.6) | 88.2 (0.4) | 80.6 (2.3) | 34.1 (2.2) | 47.6 (2.5) | 90.4 (0.9) | 77.6 (0.9) | 70.4 (4.8) | 1.3 |

(Section 4.4). To this end, we utilize four variants of HYPEBOY: **(V1)**: without feature reconstruction warm-up and projection heads, **(V2)**: without feature reconstruction warm-up[6], **(V3)**: without the hyperedge filling process, and **(V4)**: without projection heads. Here, projection heads are used only for methods with the hyperedge filling process. Note that **(V3)** is a direct extension of feature reconstructing SSL methods for graphs to hypergraphs.

As shown in Table 3, HYPEBOY, equipped with all of its components, outperforms the others in most datasets, demonstrating the effectiveness of our design choices. There are two other notable results. First, the necessity of projection heads is evidenced by the superior performance of **V2** (compared to **V1**) and ours (compared to **V4**). Second, the advantage of the hyperedge filling task over feature reconstruction is manifested by the better average rank of **V2** compared to **V3**.

## 6  CONCLUSION

In this work, we conduct a comprehensive analysis of generative self-supervised learning on hypergraphs. Our contribution is three-fold. First, we propose the hyperedge filling task, a generative self-supervised learning task on hypergraphs, and investigate the theoretical connection between the task and node classification (Section 3). Second, we present a generative SSL method HYPEBOY to solve the proposed task (Section 4). Third, we demonstrate the superiority of HYPEBOY over existing SSL methods on hypergraphs through extensive experiments (Section 5). Code and datasets are available at `link`.

---

[6]In order to mitigate an issue of over-relying on projection heads (Section 4.4), we have trained an encoder without projection heads at the beginning, and after some epochs, we train the encoder with projection heads.

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

_x\left(\vec{\mathbf{1}}^T z_i > 0\right) = P_x\left(\vec{\mathbf{1}}^T x_i > 0\right) + P_x\left(-\gamma(1 - f(x))\beta < \vec{\mathbf{1}}^T x_i < 0\right),$$

$$= \mathbb{E}_x[\mathbf{1}_{\mathcal{F}(z_i)=1}] = \underbrace{\mathbb{E}_x[\mathbf{1}_{\mathcal{F}(x_i)=1}]}_{\text{(a) Expected accuracy of naive } x_i} + \underbrace{P_x\left(-\gamma\beta < \frac{\vec{\mathbf{1}}^T x_i}{(1 - f(x))} < 0\right)}_{\text{(b) Additional gain via hyperedge filling}}. \tag{13}$$

Note that the gain term, which is the (b) term of Eq (13), is always greater than zero.

**Generalizing to $v_i \in C_0$.** Now, we analyze a node $v_i$ that belongs to $C_0$ (i.e., $v_i \in C_0$). In this case, the previous condition $\vec{\mathbf{1}}^T x'_q > 0 \equiv \beta > 0$ becomes $\vec{\mathbf{1}}^T x'_q < 0 \equiv \beta < 0$. In a similar sense, for the expected accuracy: $P_x\left(\vec{\mathbf{1}}^T z_i > 0\right)$ is changed as $P_x\left(\vec{\mathbf{1}}^T z_i < 0\right)$. In this setting, we can directly extend the result of Eq (12) as follows:

$$P_x\left(\vec{\mathbf{1}}^T z_i < 0\right) = P_x\left(\left(\vec{\mathbf{1}}^T x_i + \gamma(1 - f(x))\beta\right) < 0\right). \tag{14}$$

By employing the above proof, we can obtain the following result (note that $\beta < 0$ in this case):

$$P_x\left(\vec{\mathbf{1}}^T z_i < 0\right) = P_x\left(\vec{\mathbf{1}}^T x_i < 0\right) + P_x\left(0 < \vec{\mathbf{1}}^T x_i < -\gamma(1 - f(x))\beta\right),$$

$$= \mathbb{E}_x[\mathbf{1}_{\mathcal{F}(z_i)=0}] = \underbrace{\mathbb{E}_x[\mathbf{1}_{\mathcal{F}(x_i)=0}]}_{\text{(a) Expected accuracy of naive } x_i} + \underbrace{P_x\left(-\gamma\beta < \frac{\vec{\mathbf{1}}^T x_i}{(1 - f(x))} < 0\right)}_{\text{(b) Additional gain via hyperedge filling}}. \tag{15}$$

Thus, we can derive the same result for $v_i \in C_0$ also. $\qquad\square$

## A.2 PROOF OF THEOREM 2.

**Remark.** *We have denoted the homophilic ratio in Assumption 2 as $\mathcal{P}$ to distinguish it from the hyperedge filling probability $p_{(\mathbf{X}, \mathcal{E}, \Theta)}$. In this proof, by allowing a slight duplication in notation, we let $\mathcal{P} := p$, since we do not use $p_{(\mathbf{X}, \mathcal{E}, \Theta)}$ in this proof. In addition, we let $e_j := e$ and $q_{ij} := q$.*

*Proof.* We first derive the functional form of $P_{x,e}\left(\vec{\mathbf{1}}^T\left(\sum_{v_k \in q} x_k\right) > 0 : p\right)$. By the condition of the theorem such that $v_i \in e_j \cap C_1$, the following holds:

$$P_{x,e}\left(\vec{\mathbf{1}}^T\left(\sum_{v_k \in q} x_k\right) > 0 : p\right) = \frac{P_{x,e}\left(\vec{\mathbf{1}}^T\left(\sum_{v_k \in q} x_k\right) > 0, v_i \in e : p\right)}{P_e\left(v_i \in e : p\right)}. \tag{16}$$

It is important to note that if the number of nodes that belong to both $C_1$ and $e_j$ is decided, denoted by $s$ (i.e., $s = |e \cap C_1|$), the distribution of $\vec{\mathbf{1}}^T\left(\sum_{v_k \in q} x_k\right)$ is automatically decided, since $x_t, \forall v_t \in \mathcal{V}$ is independently generated from Gaussian distribution, whose mean vector is decided according to the class of the corresponding node. This is because for a given $s$, and by letting the size of the hyperedge $e$ as $S$ (i.e., $|e| = S$), $\sum_{v_k \in q} x_k \sim \mathcal{N}((2s - 1 - S)\mu_1, (S - 1)\mathbf{I})$ holds. From this result, the following two results are induced:

$$\vec{\mathbf{1}}^T\left(\sum_{v_k \in q} x_k\right) \sim \mathcal{N}\left(\frac{(2s - 1 - S)d}{2}, (S - 1)d\right), \tag{17}$$

$$P_x\left(\vec{\mathbf{1}}^T\left(\sum_{v_k \in q} x_k\right) > 0 : s\right) = \Phi\left((2s - S - 1)\sqrt{\frac{d}{4(S - 1)}}\right). \tag{18}$$

Thus, we rewrite Eq (16) as follows:

$$\equiv \sum_{s=0}^{S} \frac{P_{\boldsymbol{x},e}\left(\vec{\mathbf{1}}^T\left(\sum_{v_k \in q} \boldsymbol{x}_k\right) > 0, s, v_i \in e\right)}{P_e\left(v_i \in e : p\right)}, \tag{19}$$

$$= \sum_{s=0}^{S} \frac{P_{\boldsymbol{x}}\left(\vec{\mathbf{1}}^T\left(\sum_{v_k \in q} \boldsymbol{x}_k\right) > 0 : s\right) \times P_e(s, v_i \in e)}{P_e\left(v_i \in e : p\right)}. \tag{20}$$

By Assumption 2, each $P_e(s, v_i \in e)$ is derived as follows[7]:

$$P_e(s, v_i \in e : p) = \binom{S}{s} \left(\underbrace{\frac{p^s(1-p)^{S-s}}{2}}_{\text{prob. of } s \text{ at } c=1} + \underbrace{\frac{(1-p)^s p^{S-s}}{2}}_{\text{prob. of } s \text{ at } c=0}\right) \underbrace{\left(1 - \frac{\binom{N-1}{s}}{\binom{N}{s}}\right)}_{\text{prob. of } v_i \in C_1 \cap e}. \tag{21}$$

By using Eq (21), we derive $P_e(v_i \in e : p)$ as follows:

$$P_e(v_i \in e : p) = \sum_{s=0}^{S} \binom{S}{s}\left(\frac{p^s(1-p)^{S-s}}{2} + \frac{(1-p)^s p^{S-s}}{2}\right)\left(1 - \frac{\binom{N-1}{s}}{\binom{N}{s}}\right), \tag{22}$$

$$= \sum_{s=0}^{S} \binom{S}{s}\left(\frac{p^s(1-p)^{S-s}}{2} + \frac{(1-p)^s p^{S-s}}{2}\right)\frac{s}{N}, \tag{23}$$

$$= \frac{1}{N}\sum_{s=0}^{S} \binom{S}{s}\left(\underbrace{\frac{sp^s(1-p)^{S-s}}{2}}_{s \sim B(S,p)} + \underbrace{\frac{s(1-p)^s p^{S-s}}{2}}_{s \sim B(S,1-p)}\right), \tag{24}$$

$$= \frac{1}{N}\left(\frac{Sp}{2} + \frac{S(1-p)}{2}\right), \because \text{each is an expectation function}, \tag{25}$$

$$= \frac{S}{2N}. \tag{26}$$

With the result of Eq (26) and Eq (18), we rewrite Eq (20) as follows:

$$\equiv \frac{2N}{S}\sum_{s=0}^{S}\binom{S}{s}\left(\frac{p^s(1-p)^{S-s}}{2} + \frac{(1-p)^s p^{S-s}}{2}\right)\frac{s}{N}\Phi\left((2s-1-S)\sqrt{\frac{d}{4(S-1)}}\right), \tag{27}$$

$$= \frac{1}{S}\sum_{s=0}^{S}\binom{S}{s}s\left(p^s(1-p)^{S-s} + (1-p)^s p^{S-s}\right)\Phi\left((2s-1-S)\sqrt{\frac{d}{4(S-1))}}\right). \tag{28}$$

Note that our main function, which is $P_{\boldsymbol{x},e}\left(\vec{\mathbf{1}}^T\left(\sum_{v_k \in q} \boldsymbol{x}_k\right) > 0 : p\right)$, is equal to Eq (28). Here, one can easily verify that Eq (28) is equivalent to the first statement of the theorem.

Now, we show the first and second statements of the theorem. Here, we first show the second statement, and from the result, we further prove the first statement.

**Second statement.** Now, we further rewrite Eq (28) by adequately mixing two binomial functions. For simplicity, we denote $\Phi((2s-1-S)\sqrt{\frac{d}{4(S-1)}})$ as $\phi(s)$, since $\Phi(\cdot)$ only depends on $s$, and other terms are fixed constants. We rewrite Eq (28) as follows:

$$= \frac{1}{S}\sum_{s=0}^{S}\binom{S}{s}s\left(p^s(1-p)^{S-s} + (1-p)^s p^{S-s}\right)\phi(s), \tag{29}$$

$$= \underbrace{\frac{1}{S}\sum_{s=0}^{S}\binom{S}{s}sp^s(1-p)^{S-s}\phi(s)}_{\text{(Term 1)}} + \underbrace{\frac{1}{S}\sum_{s=0}^{S}\binom{S}{s}s(1-p)^s(p)^{S-s}\phi(s)}_{\text{(Term 2)}}. \tag{30}$$

---

[7]prob. indicates probability.

We define $s' := (S - s)$. Due to the symmetric characteristic of the binomial function and standard Gaussian distribution, we can rewrite (Term 2) in Eq (30) as follows:

$$\equiv \frac{1}{S} \sum_{s'=0}^{S} \binom{S}{s'} p^{s'} (1-p)^{S-s'} (S-s') \Phi\left((S - 2s' - 1)\sqrt{\frac{d}{4(S-1)}}\right). \tag{31}$$

Since we can regard $s$ and $s'$ as equivalent terms in our framework, we add (Term 1) of Eq (30) and Eq (31) as follows:

$$= \frac{1}{S} \sum_{s=0}^{S} \binom{S}{s} p^s (1-p)^{S-s} \underbrace{\left[(S-s)\Phi\left((S-2s-1)\sqrt{\frac{d}{4(S-1)}}\right) + s\Phi\left((2s-S-1)\sqrt{\frac{d}{4(S-1)}}\right)\right]}_{\text{Term (a)}}.$$
$$\tag{32}$$

We aim to show that Eq (32) is a strictly increasing function w.r.t. $p \in [0.5, 1]$. For simplicity, we let $K := \sqrt{d/(4(S-1))}$. We first discard $1/(S+1)$ from Eq (32) for simplicity, since $1/(S+1)$ is a constant, which is independent of the increasing/decreasing of values. When we do not consider Eq (32) Term (a), Eq (32) is equivalent to 1, and this is a probability distribution of binomial distribution.

That is, we can think of Eq (32) as a weighted average of the following values:

$$\left[(S-s)\Phi\left((S - 2s - 1)K\right) + s\Phi\left((2s - S - 1)K\right)\right], \forall s \in \{0, 1, \cdots, S\}. \tag{33}$$

Specifically, we denote $\binom{S}{s} p^s (1-p)^{S-s}$ as a weight of $\left[(S-s)\Phi\left((S-2s-1)K\right) + s\Phi\left((2s - S - 1)K\right)\right]$. Note that values of Eq (33) and their weights are both symmetric about $s = S/2$: $s$ and $S - s$ have the same value. Thus, we can rewrite Eq (32) for an even number $S$ as:

$$\equiv \sum_{s=\lfloor S/2 \rfloor}^{S} \binom{S}{s} \left(p^s(1-p)^{S-s} + (1-p)^s p^{S-s}\right)$$
$$\times \left[(S-s)\Phi\left((S - 2s - 1)K\right) + s\Phi\left((2s - S - 1)K\right)\right]. \tag{34}$$

While this formulation is for an odd number $S$, we can extend further results to an even number $S$.

We first show that Eq (33) is an increasing function w.r.t. $s \in \{\lfloor S/2 \rfloor, \lfloor S/2 \rfloor + 1, \cdots, S\}$. For two cases where $s = k$ and $s = k + 1$, their Eq (33) is compared as follows:

$$\left[(S - k - 1)\Phi\left(S - 2k - 3\right) + (k+1)\Phi\left(2k - S + 1\right)\right] \tag{35}$$
$$> \left[(S - k)\Phi\left(S - 2k - 1\right) + k\Phi\left(2k - S - 1\right)\right], \tag{36}$$
$$\equiv k\left(\Phi\left(2k - S + 1\right) - \Phi\left(2k - S - 1\right)\right) + \Phi\left(2k - S + 1\right) \tag{37}$$
$$> (S - k)\left(\Phi\left(S - 2k - 1\right) - \Phi\left(S - 2k - 3\right)\right) + \Phi\left(S - 2k - 3\right), \tag{38}$$

By the condition of $s \geq \lfloor S/2 \rfloor$, $k > S - k$ holds. Furthermore, by the characteristic of the CDF of standard normal distribution, $(\Phi\left(2k - S + 1\right) - \Phi\left(2k - S - 1\right)) > (\Phi\left(S - 2k - 1\right) - \Phi\left(S - 2k - 3\right))$ holds. Moreover, since $s \geq \lfloor S/2 \rfloor$, $\Phi\left(2k - S + 1\right) > \Phi\left(S - 2k - 3\right)$ also holds. In sum, Eq (35) > Eq (36) holds, and this result implies that Eq (33) is an increasing function w.r.t. $s \in \{\lfloor S/2 \rfloor, \lfloor S/2 \rfloor + 1, \cdots, S\}$.

Since greater $s$ has a higher value of Eq (33), we can naturally conclude that Eq (34), which is our goal function, is an increasing function w.r.t. $p$ if weights are more assigned to the terms related to the higher $s$ as $p$ increases. This idea is formalized as: $\exists s^* \in \{\lfloor S/2 \rfloor, \cdots, S\}$, s.t. $\partial w(p, s)/\partial p > 0, \forall s \geq s^*$ and $\partial w(p, s)/\partial p < 0, \forall s \leq s^*$, where $w(p, s) = \binom{S}{s}\left(p^s(1-p)^{S-s} + (1-p)^s p^{S-s}\right)$. To this end, we derive the first derivative of $w(p, s)$:

$$\frac{\partial \omega(p, s)}{\partial p} = \binom{S}{s}\left(\underbrace{sp^{s-1}(1-p)^{S-s}}_{(a)} - \underbrace{(S-s)p^s(1-p)^{S-s-1}}_{(b)}\right. \tag{39}$$
$$\left. - \underbrace{s(1-p)^{s-1}p^{S-s}}_{(c)} + \underbrace{(S-s)(1-p)^s p^{S-s-1}}_{(d)}\right). \tag{40}$$

We then compute $(a) + (b)$ and $(c) + (d)$ respectively, and sum them up as follows:

$$(a) + (b) = p^{s-1}(1-p)^{S-s-1}(s(1-p) - (S-s)p), \tag{41}$$

$$= p^{s-1}(1-p)^{S-s-1}(s - sp - Sp + sp), \tag{42}$$

$$(c) + (d) = (1-p)^{s-1}p^{S-s-1}(-sp + (S-s)(1-p)), \tag{43}$$

$$= (1-p)^{s-1}p^{S-s-1}(-sp + (S-s) - Sp + sp), \tag{44}$$

$$(a) + (b) + (c) + (d) = p^{s-1}(1-p)^{S-s-1}(s - Sp) + (1-p)^{s-1}p^{S-s-1}(S - s - Sp). \tag{45}$$

**Remark.** *When $p = 1/2$, Eq (45) $= 0$ holds, which means that if Eq (32) turns out to be an increasing function w.r.t. $p$, Eq (32) has its minimum value at $p = 1/2$. Thus, we narrow down the range of $p$ as $p > 1/2$.*

Moving on to our main object, first, we find $s^*$ such that Eq (45) $= 0$ holds. We utilize the logarithm function to simplify the computation (note that $\binom{S}{s}$ is canceled out).

$$\equiv p^{(s-1)}(1-p)^{S-s-1}(s - Sp) = (1-p)^{(s-1)}p^{S-s-1}(s - S + Sp), \tag{46}$$

$$\equiv (s-1)\log p + (S-s-1)\log(1-p) + \log(s - Sp) =$$
$$(s-1)\log(1-p) + (S-s-1)\log p + \log(s - S + Sp),$$

$$= (s-1)\log\left(\frac{p}{1-p}\right) + (S-s-1)\log\frac{1-p}{p} + \log\frac{s - Sp}{s - S(1-p)} = 0,$$

$$= \underbrace{(2s - S)\log\frac{p}{1-p}}_{\text{Term (a)}} + \underbrace{\log\frac{s - Sp}{s - S(1-p)}}_{\text{Term (b)}} = 0. \tag{47}$$

However, it is hard to obtain the exact solution of Eq (47). Instead, we analyze the formula of Term (a) + Term (b) in Eq (47), which is expressed as below:

$$(2s - S)\log\frac{p}{1-p} + \log\frac{s - Sp}{s - S(1-p)}. \tag{48}$$

To show the required conditions, it is enough to show that Eq (48) $< 0$ holds at $s = S/2$, and Eq (48) $> 0$ holds at $s = S$. This is because since Eq (48) is an increasing function w.r.t. $s$, the above two conditions imply that there exists a $s^*$, which is our interest, between $S/2$ and $S$. Thus, we finalize the proof by showing that at $s = S/2$, Eq (48) $< 0$ holds and at $s = S$, Eq (48) $> 0$ holds.

When plugging $S/2$ to $s$, $2s - S$ gets zero. Since $p > 1/2$ is given, $\log\frac{s-Sp}{s-S(1-p)}$ becomes negative. Thus, we have demonstrated the first claim. We now plug $S$ into $s$ and obtain the following result:

$$\equiv (2S - S)\log\frac{p}{1-p} + \log\frac{S - Sp}{S - S(1-p)} > 0, \tag{49}$$

$$\equiv S\log\frac{p}{1-p} + \log\frac{S(1-p)}{Sp} > 0, \tag{50}$$

$$\equiv S\log\frac{p}{1-p} - \log\frac{p}{1-p} > 0, \tag{51}$$

$$\equiv (S-1)\log\frac{p}{1-p} > 0. \tag{52}$$

Thus, we can conclude that Eq (32), which is our interest, is an increasing function w.r.t. $p > 1/2$, and has its minimum value at $p = 1/2$.

Now, we show that the above proof can also be applied to an even number $S$. Note that for an even number $S$, Eq (32) is equal to the addition of Eq (34) that starts with $s = S/2 + 1$ and Eq (53),

$$w(p, s) = \binom{S}{\frac{S}{2}} \times \frac{S}{2} \times p^{\frac{S+1}{2}}(1-p)^{\frac{S+1}{2}}. \tag{53}$$

Specifically, if we obtain $\partial w(p, s)/\partial p$, we get

$$\frac{\partial w(p, s)}{\partial p} = \left( \binom{S}{\frac{S}{2}} \times \frac{S}{2} \right) \times \left( \frac{S}{2} p^{\frac{S-2}{2}} (1-p)^{\frac{S-2}{2}} (1-2p) \right) \leq 0, \forall p \in [0.5, 1]. \quad (54)$$

Since the sign of $\partial w(p, s)/\partial w(p, s)$ remains unchanged for $s = S + 1/2$, our proof is still valid for an even number $S$.

**First statement.** From the above proof, we now show the lower bound of Eq (27), which is the first statement. As described on Remark in A.2, the value of Eq (27) (equivalent to Eq (32)) is minimized at $p = 0.5$. Thus, we show the statement by finding the lower bound of Eq (28) at $p = 0.5$, by again denoting $K := \sqrt{\frac{d}{4(S-1)}}$, as follows:

$$= \frac{1}{S} \sum_{s=0}^{S} \binom{S}{s} s \left( p^s (1-p)^{S-s} + (1-p)^s p^{S-s} \right) \Phi \left( (2s - 1 - S) \sqrt{\frac{d}{4(S-1))}} \right), \quad (55)$$

$$= \frac{1}{S} \sum_{s=0}^{S} \binom{S}{s} 2s \left( \frac{1}{2} \right)^S \Phi \left( (2s - S - 1)K \right). \quad (56)$$

First, consider an even number $S$. Then, for $\Phi((2s - S - 1)K), \forall s \in \{1, \cdots, S\}$, consider the below relations:

$$1 \underbrace{\Phi(-S+1)}_{(a1)} + 2 \underbrace{\Phi(-S+3)}_{(b1)} + \cdots + (S-1) \underbrace{\Phi(S-3)}_{(b2)} + S \underbrace{\Phi(S-1)}_{(a2)}. \quad (57)$$

In Eq (57), $(a1) + (a2) = 1$ holds, and $(b1) + (b2) = 1$ holds also. In this manner, we can pair all terms in Eq 57. If we take a closer look at this, pairs have a relation of: $k_{11}k_{12} + k_{21}k_{22}$ such that $k_{11} + k_{21} = S, k_{12} + k_{22} = 1, k_{11} < k_{12}$, and $k_{21} < k_{22}$ hold. Note that $k_{11}k_{12} + k_{21}k_{22}$ is lower-bounded by $\frac{k_{11}+k_{21}}{2}$ since distributing the weight of a larger value to a smaller value will decrease the overall value. From this result, we can obtain a lower bound of Eq (57) as follows:

$$\frac{1}{S} \sum_{s=0}^{S} \binom{S}{s} 2s \left( \frac{1}{2} \right)^S \Phi \left( (2s - S - 1)K \right) \geq \frac{1}{S} \sum_{s=0}^{S} \binom{S}{s} 2s \left( \frac{1}{2} \right)^S \frac{1}{2} = 0.5. \quad (58)$$

Now, we consider an odd number $S$. Similar things also happen in this case, but there is a term $s = \frac{S+1}{2}; \Phi(0)$ that does not have any pair. On the other hand, since $\Phi(0) = 0.5$, we can still ensure Eq (58) holds in this case. Thus, we have derived that regardless of the fact whether $S$ is an odd or even number,

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

 hyperedges represent sets of publications co-cited by particular publications. For example, if a particular publication has cited nodes (publications) $v_i, v_j$, and $v_k$, these nodes are grouped as a hyperedge $\{v_i, v_j, v_k\} \in \mathcal{E}$. Node features are bag-of-words of the corresponding publication. Node classes indicate categories of the publication.

**Co-authorship datasets.** We utilize four co-authorship datasets: Cora-CA, DBLP-P, DBLP-A, and AMiner. In Cora-CA, DBLP-P, and AMiner, each node represents a publication, and a set of publications that are written by a particular author is grouped as a hyperedge. Features are bag-of-words of the corresponding publication, and classes indicate categories of the publication. Conversely in DBLP-P, each node represents an author, and co-authors of a particular publication are grouped as a hyperedge. Node features are bag-of-words regarding the research keywords of the author. Node classes indicate the research area of the author.

**Computer graphic datasets.** We utilize one computer graphical dataset: Modelnet-40 (MN-40). In this dataset, each node represents a visual object, and hyperedges are synthetic hyperedges that have been created with a k-NN graph constructed according to the features of each data point, following Feng et al. (2019) and Chien et al. (2022). Node features are embeddings of each visual object obtained via GVCNN (Feng et al., 2018) and MVCNN (Su et al., 2015). Node classes indicate the categories of the corresponding visual object.

**Movie-Actor dataset.** We utilize one movie-actor dataset: IMDB. In this dataset, each node indicates a movie, and the filmography of a particular actor is grouped as a hyperedge. Node features are bag-of-words of the plot of the movie. Node classes indicate the genre of the movie.

**News dataset.** We utilize one news dataset: 20NewsGroups (20News). In this dataset, each node represents documents of 20 newsgroups and a set of documents containing a particular word is grouped as a hyperedge. Node features are TF-IDF representations of news messages. Node classes indicate the categories of the corresponding document.

**Political membership dataset.** We utilize one political membership dataset: House. In this dataset, each node represents a member of the US House of Representatives, and members belonging to the same committee are grouped as a hyperedge. Node features are created by adding noise to the one-hot encoded label vector, as suggested by Chien et al. (2022). Node classes indicate the political party of the representatives.

## D  EXPERIMENTAL DETAILS

In this appendix section, we provide details of our experiments. We first describe the machines we have used in our experiments. Then, we provide details of baseline methods and implementations.

Table 5: Hyperparameter combination of HYPEBOY that shows the best validation accuracy on each dataset. $p_v \in [0, 1]$ indicates magnitude of feature augmentation, $p_e$ indicates magnitude of topological augmentation, and $SSLepochs$ indicates the training epoch of hyperedge filling task.

| | Citeseer | Cora | Pubmed | Cora-CA | DBLP-P | DBLP-A | AMiner | IMDB | MN-40 | 20News | House |
|---|---|---|---|---|---|---|---|---|---|---|---|
| $p_v$ | 0.2 | 0.4 | 0.0 | 0.4 | 0.0 | 0.1 | 0.2 | 0.3 | 0.4 | 0.2 | 0.4 |
| $p_e$ | 0.9 | 0.9 | 0.5 | 0.8 | 0.6 | 0.9 | 0.6 | 0.9 | 0.5 | 0.5 | 0.8 |
| SSL epochs | 120 | 200 | 180 | 200 | 180 | 120 | 100 | 180 | 180 | 80 | 140 |

## D.1 MACHINES

All experiments are conducted on a machine with NVIDIA RTX 8000 D6 GPUs (48GB memory) and two Intel Xeon Silver 4214R Processors.

## D.2 DETAILS OF BASELINE METHODS

**Neural networks.** We utilize 10 supervised baseline models: MLP, HGNN (Feng et al., 2019), HyperGCN (Yadati et al., 2019), HNHN (Dong et al., 2020), three unified hypergraph networks, (UniGCN, UniGIN, UniGCNII) (Huang & Yang, 2021), AllSet (Chien et al., 2022), ED-HNN (Wang et al., 2023a), and PhenomNN (Wang et al., 2023b).

**Graph SSL.** In order to run two graph generative SSL methods, which are GraphMAE2 (Hou et al., 2023) and MaskGAE (Li et al., 2023), we transform the input hypergraph into a graph by using the clique expansion, which converts each hyperedge into a clique of a graph (Dong et al., 2020).

**H-GD.** One of our baseline methods H-GD directly extends the Group Discrimination method, an SSL technique designed on a graph (Zheng et al., 2022). We strictly follow the overall structure suggested by Zheng et al. (2022), while the feature augmentation and topology augmentation have been replaced into $\tau_x$ and $\tau_e$ that are described in Section 4.1, respectively.

## D.3 DETAILS OF SETTINGS AND IMPLEMENTATIONS

**Hyperedge prediction setting.** Note that we need to create negative hyperedge samples to evaluate the performance of a model on the hyperedge prediction task. In our experiment, we have created negative hyperedge samples based on SNS (size negative samples) (Patil et al., 2020). Specifically, to create a negative hyperedge, we sample a size of the current hyperedge from the real-world hyperedge size distribution of the corresponding dataset. Then, we sample nodes uniformly at random and fill the corresponding hyperedge. In this manner, we create negative hyperedges with the same number of ground-truth hyperedges.

**Feature reconstruction warm-up of HYPEBOY.** As described in Section 4.4, before training an encoder HNN with HYPEBOY, we utilize a feature reconstruction process, which directly extends GraphMAE (Hou et al., 2022) to hypergraph structure. This process mitigates the encoder's over-reliance on the projection heads, improving the performance in downstream tasks (Section 5.3).

To this end, we first mask a certain portion of input node features with a learnable mask token. We have fixed the augmentation portion as 0.5 for all the cases. In addition, we augmentation the input hyperedge by using $\tau_{\mathcal{E}}$, which is defined in Section 4.1. We have fixed the magnitude of augmentation $p_e$ as 0.2 for all the cases. Then, we obtain node embeddings by using an encoder HNN. We again mask embeddings of nodes that have been masked at the input level by using another learnable mask. Subsequently, we obtain reconstructed features with an HNN decoder, which has an architecture that is the same as that of the encoder HNN. Finally, we utilize the cosine similarity between reconstructed features and original features as a loss, specifically, one minus cosine similarity. Specifically, we utilize UniGCNII (Huang & Yang, 2021) for encoder and decoder.

**Projection heads of HYPEBOY.** For projection heads (Section 4.2), we utilize a two-layer MLP model with a ReLU (Nair & Hinton, 2010) activation function for both node projection head $f'_\phi$ and set projection head $f'_\rho$. Note that these projection heads are updated via the hyperedge filling task together with an encoder HNN.

Table 6: Comparison between HYPEBOY without augmentation step (denoted by w/o Aug.) and HYPEBOY (denoted by w/ Aug.) in the node classification task under two protocols. The best performance is colored as a green. In most of the settings, HYPEBOY outperforms its variant.

| Protocol | Method | Citeseer | Cora | Pubmed | Cora-CA | DBLP-P | DBLP-A | AMiner | IMDB | MN-40 | 20News | House |
|---|---|---|---|---|---|---|---|---|---|---|---|---|
| Fine | w/o Aug. | 53.5 (8.7) | 58.6 (8.3) | 75.5 (4.0) | 62.9 (6.0) | 87.8 (0.4) | 79.7 (2.3) | 33.9 (2.1) | 47.2 (2.5) | 90.7 (0.7) | 77.5 (0.9) | 69.9 (4.9) |
| tuning | w/ Aug. | 56.7 (9.8) | 62.3 (7.7) | 77.0 (3.4) | 66.3 (4.6) | 88.2 (0.4) | 80.6 (2.3) | 34.1 (2.2) | 47.6 (2.5) | 90.4 (0.9) | 77.6 (0.9) | 70.4 (4.8) |
| Linear | w/o Aug. | 55.8 (9.0) | 60.6 (7.3) | 72.7 (3.4) | 63.3 (4.6) | 87.6 (0.5) | 81.4 (2.5) | 34.2 (2.7) | 47.8 (2.0) | 89.7 (1.9) | 75.1 (1.6) | 67.5 (1.6) |
| evaluation | w/ Aug. | 59.6 (9.9) | 63.5 (9.4) | 75.0 (3.4) | 66.0 (4.6) | 87.9 (0.5) | 81.2 (2.7) | 34.3 (3.2) | 48.8 (1.8) | 89.2 (2.2) | 75.7 (2.1) | 69.4 (5.4) |

**Training details and hyperparameters.** We utilize the Adam optimizer (Kingma & Ba, 2015) with a fixed weight decay rate of $10^{-6}$ to train all models. We fix the hidden dimension of all models and the dropout rate of them as 128 and 0.5, respectively. When training any neural network for downstream tasks, we train the model for 200 epochs, and for every 10 epochs, we evaluate the validation accuracy of the model. Then, we utilize the model checkpoint that yields the best validation accuracy to make a final evaluation of the model on the test dataset.

For a linear evaluation protocol of node classification, we utilize a logistic classifier, with a learning rate $10^{-3}$. For a linear evaluation protocol of hyperedge classification, we utilize a two-layer MLP, with a learning rate of $10^{-3}$ and the dropout rate of 0.5.

Regarding hyperparameter tuning, for each model and dataset, we find the hyperparameter combination that gives the best validation performance. Then, with the selected hyperparameters, we train the model on a training dataset and evaluate the performance of the trained model on a test dataset.

For all the supervised models we have used, we tune the learning rate as a hyperparameter within $\{0.05, 0.01, 0.005, 0.001, 0.0005, 0.0001\}$. Moreover, especially for SOTA HNNs PhenomNN, we additionally tune their other hyperparameters. Specifically, along with the learning rate, we additionally tune $\lambda_0$ and $\lambda_1$ within $\{0, 1, 10\}$, which are weights of different message-passing functions, and $\alpha$ within $\{0, 1, 10\}$ which is a weight of node's own representation during encoding.

For all SSL baseline methods, we set a broader search space, which is as follows:

- TriCL (Lee & Shin, 2023): We tune their feature augmentation magnitude within $\{0.2, 0.3, 0.4\}$, hyperedge augmentation magnitude within $\{0.2, 0.3, 0.4\}$, and learning rate of an encoder within $\{0.01, 0.001, 0.0001\}$.

- HyperGCL (Wei et al., 2022): We tune learning rate of an encoder within $\{0.01, 0.001, 0.0001\}$, that of a generator $\{0.01, 0.001, 0.0001\}$, and weight of contrastive loss $\beta$ within $\{0.5, 1, 2\}$.

- HGD, variant of GD (Zheng et al., 2022): We tune their feature augmentation magnitude within $\{0.2, 0.3, 0.4\}$, hyperedge augmentation magnitude within $\{0.2, 0.3, 0.4\}$, and learning rate of an encoder within $\{0.01, 0.001, 0.0001\}$.

- GraphMAE2 (Hou et al., 2023): We tune mask ratio within $\{0.25, 0.5, 0.75\}$ and learning rate of an encoder within $\{0.01, 0.001, 0.0001\}$.

- MaskGAE (Li et al., 2023) We tune degree loss weight $\alpha$ within $\{0.001, 0.002, 0.003\}$ and learning rate of an encoder within $\{0.01, 0.001, 0.0001\}$.

In addition, we set self-supervised learning epochs of the above methods as hyperparameters, searching within $\{50, 100, 150, \cdots, 450, 500\}$.

For HYPEBOY, we tune feature augmentation magnitude $p_x$ within $\{0.0, 0.1, 0.2, 0.3, 0.4\}$, and hyperedge augmentaiton magnitude $p_e$ within $\{0.5, 0.6, 0.7, 0.8, 0.9\}$. We have fixed the learning rate and training epochs of the feature reconstruction warm-up as 0.001 and 300, respectively. In addition, we first train an encoder with a feature reconstruction process for 300 epochs. Then, we train the encoder with the hyperedge filling, and the specific epoch is a hyperparameter searched within $\{20, 40, \cdots, 180, 200\}$. We report a hyperparameter combination of each dataset that shows the best validation accuracy in our fine-tuning experiment (Section 5.1) in Table 5.

# E ADDITIONAL EXPERIMENTAL RESULTS

In this section, we present several experimental results that are omitted in the main paper.

Figure 4: Analyzing dimensional collapse of HYPEBOY with/without projection heads on five benchmark datasets. While HYPEBOY does not suffer from dimensional collapse, its variant that does not utilize projection heads, suffers from this issue.

### E.1 COMPARISON AGAINST HYPEBOY WITHOUT AUGMENTATION

As described in Section 4.1, augmentation (masking) has played a key role in various generative SSL methods in obtaining effective representations. Motivated by their findings, we have also adopted augmentation strategy $\tau_x$ and $\tau_e$ in HYPEBOY also. To demonstrate the effectiveness of augmentation, we have employed one variant of HYPEBOY, where the augmentation step is omitted. In other words, the input of a target HNN $f_\theta$ is always a clean feature and topology, $(\mathbf{X}, \mathcal{E})$. Then, we compare this variant with our proposed HYPEBOY, in the node classification task under fine-tuning protocol and linear evaluation protocol. As shown in Table 6, HYPEBOY outperforms its variant, which does not have an augmentation step in most of the datasets of both protocols. Thus, we can conclude that augmentation is crucial for mitigating the issue of over-emphasizing proximity information and creating effective representations.

### E.2 DIMENSIONAL COLLAPSE ANALYSIS

In Section 4.2, we discuss the role of projection heads in terms of dimensional collapse: projection heads encourage an encoder HNN to avoid dimensional collapse. We further anlyze this phenomenon in five more benchmark hypergraph datasets: Citeseer, Pubmed, Cora-CA, DBLP-P, and DBLP-A. As shown in Figure 4, in all five datasets, we have verified that small singular values of embeddings that are created via HYPEBOY without projection heads drop to zero, empirically demonstrating that the dimensional collapse has occurred (red lines). Notably, such an issue has not been observed in HYPEBOY (blue lines).

### E.3 ALIGNMENT AND UNIFORMITY ANALYSIS

In Section 4.3, we discuss the role of our design choice of $p_{(\mathbf{X}, \mathcal{E}, \Theta)}(\cdot)$ in promoting alignment and uniformity of representations. We further demonstrate this characteristic in five more benchmark hypergraph datasets: Citeseer, Pubmed, Cora-CA, DBLP-P, and DBLP-A. As shown in Figure 5, representations obtained from HYPEBOY achieve both uniformity (first column) and alignment (other columns) in most of the cases.

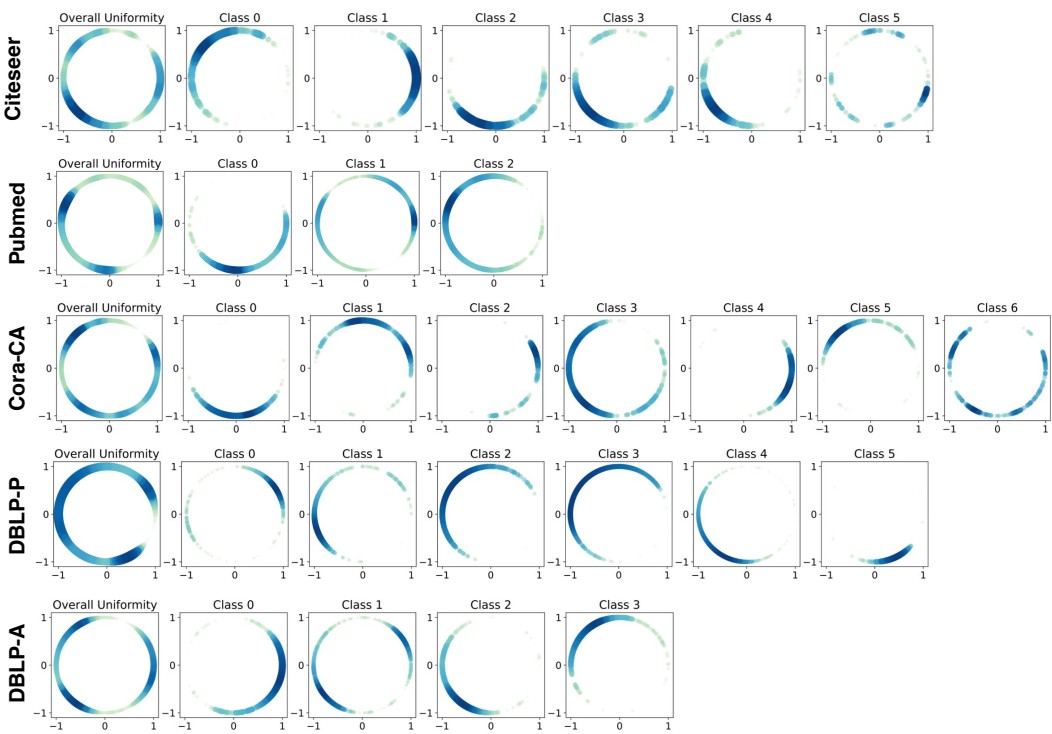

Figure 5: Analyzing alignment and uniformity of representations obtained via HYPEBOY. In most cases, representations obtained by HYPEBOY achieve both alignment and uniformity.