# OpenReview forum: "HypeBoy: Generative Self-Supervised Representation Learning on Hypergraphs"
_ICLR.cc/2024/Conference — ICLR 2024 poster_

### Official Review · Reviewer_EPYj · 2023-10-28

**Soundness:** 3 good
**Presentation:** 3 good
**Contribution:** 2 fair
**Rating:** 5
**Confidence:** 4

**Summary:**

This article proposes a novel self-supervised learning (SSL) task, hyperedge filling. The authors also give the relationship between hyperedge filling and node classification. Based on the SSL task, the authors propose HypeBoy, which is composed of 3 steps, hypergraph augmentation, hypergraph encoding, and hypergraph filling. The authors demonstrate the effectiveness of HypeBoy under multiple SSL tasks on multiple datasets.

**Strengths:**

1. The article proposes a novel SSL task, hyperedge filling, which is a supplement to the general node classification task.
2. The hyperedge-filling task proposed in the article can obtain more accurate node features, thereby helping other tasks.

**Weaknesses:**

1. In the hypothesis, the premise of the article is that each hyperedge contains most of the same category vertices. However, there are hyperedges that do not satisfy this situation. For example, a hyperedge of size 10 contains 5 nodes of category A and 5 category B.
2. One of the main contributions of the paper is hyperedge filling, but using only hyperedge filling (v1) in the ablation experiment is the worst among v1-v4. The effectiveness of using hyperedge filling alone is questionable, or it needs to be bound to feature reconstruction and projection heads.

**Questions:**

1. In BASIC SETTING, one of the assumptions is that the homophily ratio of each hyperedge is in [0.5, 1]. In graphs, the homophily rate is defined as the proportion of intra-class edges to all edges. So how does this definition apply to the hypergraph? Do all datasets in the experiments satisfy this assumption?
2. I noticed that in section 5.1, the article uses a setting of 1% of the training set, which is different from most articles. Most other papers use fixed division or 5/10 nodes per category. I'm curious about what the considerations are for such an experimental setup?

---

> ### Author Response · Authors · 2023-11-17
> **Dear Reviewer EPYj**
>
> Dear Reviewer EPYj,
>
> We appreciate your constructive review. We provide our responses to each of your comments below. First, however, please let us make clarifications about the homophily ratio, which would serve to better clarify our responses.

---

> ### Author Response · Authors · 2023-11-17
> **Clarification for Homophily Ratio $\mathscr{P}$**
>
> - **Clarification 1.** Our theoretical analysis covers both homophily and non-homophily scenarios.
>     - Specifically, our theorems (Theorems 1 and 2) are valid in both homophilic and non-homophilic cases.
>         - The (expected) general-sense homophily (i.e. the same class nodes tend to interact with each other) is in a symmetric U-curve relation w.r.t. $\mathscr{P} \in [0,1]$.
>             - Homophily decreases at $\mathscr{P} \in [0, 0.5]$ and increases at $\mathscr{P} \in [0.5, 1]$.
>             - Homophily is maximized at either $\mathscr{P} =1$ or$\mathscr{P} = 0$ and minimized at $\mathscr{P} = 0.5$.
>         - Our theorems (Theorems 1 and 2) hold for any $\mathscr{P} \in [0,1]$, i.e., both homophilic and non-homophilic cases.
> - **Clarification 2.** $\mathscr{P}$ is a parameter that controls the number of particular class nodes in a hyperedge.
>     - Specifically, homophily does not monotonically increase as $\mathscr{P}$ increases (see Clarification 1 above).
>     - We apologize for the confusion seemingly caused by our verbal expression of $\mathscr{P}$ (we used the term “homophily ratio”).
>     - We have renamed the $\mathscr{P}$ as “***affinity parameter***” and modified it in the revised manuscript with additional descriptions.
>     - Details regarding $\mathscr{P}$ are described below:
>         - In our theoretical analysis, we assume two types of hyperedges (Type 0 and Type 1).
>         - (Homophilic) As $\mathscr{P} \rightarrow 1$, Type 0 hyperedges tend to be filled with class-0 nodes, and Type 1 hyperedges are filled with class-1 nodes.
>         - (Homophilic) As $\mathscr{P} \rightarrow 0$, Type 0 hyperedges tend to be filled with class-1 nodes, and Type 0 hyperedges are filled with class-0 nodes.
>         - (Non-Homophilic) As $\mathscr{P} \rightarrow 0.5$, for each hyperedge, half of it tends to be filled with class-1 nodes, and the rest half tends to be filled with class-0 nodes.
>     - Our data model is symmetric about $\mathscr{P} = 0.5$. In other words, $\mathscr{P}$ and $1 - \mathscr{P}$ are expected to produce equally homophilic hypergraphs.
>         - Considering the entire hyperedges, the tendency of node classes being mixed within hyperedges is equivalent at hypergraphs that are generated with $\mathscr{P}$ and those with $1-\mathscr{P}$.

---

> ### Author Response · Authors · 2023-11-17
> **Response to Weakness 1 [Premise on Homophily]**
>
> ### Comment
>
> `In the hypothesis, the premise of the article is that each hyperedge contains most of the same category vertices. However, there are hyperedges that do not satisfy this situation. For example, a hyperedge of size 10 contains 5 nodes of category A and 5 category B.`
>
> ---
>
> ### Response
>
> - We apologize for the confusion. The detailed descriptions regarding $\mathscr{P}$ are provided in “Clarification for Homophily Ratio”.
> - **Clarification W1.1:** Our theoretical analysis covers both general-sense homophily and non-homophily (i.e. homophily means that the same-class nodes tend to interact with each other).
>     - Both Theorem 1 and 2 hold for any $\mathscr{P} \in [0, 1]$, i.e., both homophilic and non-homophilic cases.
>     - With $\mathscr{P} = 0.5$, each hyperedge is expected to contain half class-0 nodes and class-1 nodes.
> - **Experiment W1.2**: To demonstrate HypeBoy’s efficacy when the homophily assumption of input data is violated, we conduct an additional experiment, described in the tables below.
>     - **Results**: HypeBoy is the most robust to decreasing homophily. Its performance gap to the SOTA baselines tends to increase as the homophily assumption of the input data is increasingly violated.
>     - The experimental details are as follows:
>         - We randomly choose two hyperedges ($e_{1}$ and $e_{2}$), and choose one node each from the chosen hyperedges ($v_{1} \in e_{1}$ and $v_{2}\in e_{2}$).
>         - Then, we swap two chosen nodes: $e^\prime_1 = (e_1 \setminus \{v_1\})  \cup \{v_2\}$ and $e^\prime_2 = (e_2 \setminus \{v_2\}) \cup \{v_1\}$, then $e_{1} = e^\prime_{1}$ and $e_{2} = e^\prime_{2}$.
>         - We repeat this process. With more swapping, a hypergraph is expected to get more non-homophilic.
>     - The results are added in the revised manuscript as Table 10.
> - **Argument W1.3**: HypeBoy is more effective than its competitors in less homophilic hypergraphs.
>     - The proposed method (HypeBoy) outperforms the baseline methods in non-homophilic benchmark datasets. See more details in the response for Q1.
>
> ---
>
> ### Node Swapping Experimental Results
>
> **Cora (Value in a bracelet indicates the performance gap from that of the original hypergraph)**
>
> |  | Original | Shuffle 50 | Shuffle 100 | Shuffle 150 | Shuffle 200 |
> | --- | --- | --- | --- | --- | --- |
> | No Pretraining | 0.485 | 0.467 (-0.018) | 0.447 (-0.038) | 0.435 (-0.050) | 0.409 (-0.076) |
> | TriCL | 0.602 | 0.591 (-0.011) | 0.561 (-0.041) | 0.557 (-0.045) | 0.536 (-0.065) |
> | HyperGCL | 0.603 | 0.574 (-0.029) | 0.537 (-0.066) | 0.520 (-0.083) | 0.500 (-0.103) |
> | HypeBoy (Ours) | 0.623 | 0.617 (**-0.006**) | 0.592 (**-0.031**) | 0.580 (**-0.043**) | 0.567 (**-0.056**) |
>
> **Citeseer (Value in a bracelet indicates the performance gap from that of the original hypergraph)**
>
> |  | Original | Shuffle 50 | Shuffle 100 | Shuffle 150 | Shuffle 200 |
> | --- | --- | --- | --- | --- | --- |
> | No Pretraining | 0.442 | 0.414 (-0.028)  | 0.390 (-0.052) | 0.359 (-0.083) | 0.338 (-0.104) |
> | TriCL | 0.517 | 0.511 (-0.006) | 0.475 (-0.042) | 0.452 (-0.065) | 0.418 (-0.099) |
> | HyperGCL | 0.470 | 0.458 (-0.012) | 0.433 (-0.037) | 0.409 (-0.061) | 0.377 (-0.093) |
> | HypeBoy (Ours) | 0.567 | 0.565 (**-0.002**) | 0.542 (**-0.025**) | 0.508 (**-0.059**) | 0.486 (**-0.081**) |

---

> > ### Comment · Reviewer_EPYj · 2023-11-22
> >
> > Thank the authors for the response, and I have read via the feedback. I maintain my review score.

---

> > > ### Author Response · Authors · 2023-11-22
> > > **Thank you**
> > >
> > > Dear Reviewer EPYj,
> > >
> > > Thanks again for your careful review and valuable comments, which helped us improve our submission.
> > >
> > > Best, \
> > > The Authors

---

> ### Author Response · Authors · 2023-11-17
> **Response to Weakness 2 [Effectiveness of Hyperedge Filling Alone]**
>
> ### Comment
>
> `One of the main contributions of the paper is hyperedge filling, but using only hyperedge filling (v1) in the ablation experiment is the worst among v1-v4. The effectiveness of using hyperedge filling alone is questionable, or it needs to be bound to feature reconstruction and projection heads.`
>
> ---
>
> ### Response Summary
>
> - We show that the *hyperedge filling* task is a better SSL task compared to other SSL tasks, including feature reconstruction and contrastive learning.
> - We clarify that it is common to employ additional parameters (e.g., projection heads) to complement an SSL task, and our competing SSL methods also utilize such additional parameters.
>
> ### Response
>
> - We clarify the relationship between additional parameters (e.g. projection heads) and SSL methods:
>     - **Clarification W2.1**:  Projection heads are widely used tools to complement SSL methods, enhancing their performances.
>     - **Clarification W2.2**: Typical SSL pretext tasks, *contrastive learning* and *feature reconstruction*, also require additional parameters such as projection heads, decoders, and view generators.
> - Under such context, we have conducted two additional experiments to address the reviewer’s concerns:
>     - We compared the *hyperedge filling* task with other SSL tasks under two settings: (1) encoder only, i.e., without additional parameters; (2) encoder + additional parameters. Under both settings, hyperedge filling is more effective than other SSL tasks.
>     - **Experiment W2.3 [Encoder-Only]**:
>         - **Summary**: When an encoder alone is employed, the *hyperedge filling* task is more effective than other hypergraph SSL tasks.
>         - **Setting**: We compare various SSL methods by removing their additional parameters.
>             - *Hyperedge filling*: It employs an encoder without projection heads (equivalent to V1 in the main paper).
>             - *Feature reconstruction*: We remove a decoder from the feature reconstruction method. Specifically, we make the embedding dimension of the encoder the same as the node feature dimension (other parts are the same as our feature reconstruction scheme).
>             - *Contrastive learning*: We remove projection heads from TriCL [1] (contrastive loss and augmentation steps are equivalent to TriCL).
>         - **Outcome details**: As shown in the below table, *hyperedge filling* without any parameters (V1) outperformed (1) *feature reconstruction* (V3) or (2) *contrastive learning* methods.
>         - **Revision**: We have added the results to the revised manuscript as Table 8.
>     - **Experiment W2.4 [Encoder + Additional Parameters]**:
>         - **Summary**: When additional parameters are also employed, the *hyperedge filling* task is more effective than other hypergraph SSL tasks.
>         - **Setting**: We compare various SSL methods that employ additional parameters.
>             - *Hyperedge filling*: This method employs an encoder with projection heads (equivalent to V2 in the main paper).
>             - *Feature reconstruction*: We employ both encoder and decoder (equivalent to V3 in the main paper).
>             - *Contrastive learning*: We employ TriCL and HyperGCL [2].
>         - **Outcome details**: By comparing the performance of *hyperedge filling* + projection head (V2) with that of V3, TriCL, and HyperGCL, we have verified that V2 outperforms other methods in terms of the average rank (this result can be obtained from Table 1 and 3).
>
> ---
>
> ### SSL methods W/O additional parameters
>
> |  | Cora | Citeseer | Pubmed | Cora-CA | DBLP |
> | --- | --- | --- | --- | --- | --- |
> | Contrastive learning W/O projection heads | 0.592 (0.067) | 0.445 (0.102) | 0.752 (0.040) | 0.621 (0.053) | 0.867 (0.005) |
> | Feature reconstruction W/O decoder | 0.586 (0.080) | 0.515 (0.092) | 0.747 (0.051) | 0.619 (0.073) | 0.873 (0.006) |
> | V1 (Hyperedge filling W/O projection heads) | **0.607 (0.082)** | **0.516 (0.112)** | **0.762 (0.036)** | **0.635 (0.060)** | **0.881 (0.005)** |

---

> ### Author Response · Authors · 2023-11-17
> **Response to Question 1 [Hyperedge Homophily Ratio]**
>
> ### Comment
>
> `In BASIC SETTING, one of the assumptions is that the homophily ratio of each hyperedge is in [0.5, 1]. In graphs, the homophily rate is defined as the proportion of intra-class edges to all edges. So how does this definition apply to the hypergraph? Do all datasets in the experiments satisfy this assumption?`
>
> ---
>
> ### Response Summary
>
> - Our theoretical analyses hold for both general-sense homophily- and non-homophily (i.e. homophily means that the same-class nodes tend to interact with each other).
> - Empirically, in both homophilic and non-homophilic hypergraphs, our method (HypeBoy) outperforms other SSL methods.
>
> ---
>
> ### Response
>
> - **Clarification Q1.1 [Theory]**:
>     - We are sorry for the confusion. The additional descriptions for $\mathscr{P}$ are provided in Clarification for Homophily Ratio.
>     - We again clarify that our theorem is valid at any $\mathscr{P} \in [0,1]$. In other words, Theorem 1 and 2 hold for both homophilic and non-homophilic hypergraphs.
> - **Clarification Q1.2 [General-sense Homophily Measure]**:
>     - We have used one of the widely used hypergraph homophily analysis methods, proposed by Veldt et al. 2023 [3].
>     - The homophily ratio the reviewer has mentioned can be directly extended to hypergraphs as the proportion of hyperedges that consist of the same-class nodes to all hyperedges.
>     - However, this has limitations in measuring the extent of homophily within each hyperedge.
>         - For instance, consider two hyperedges $e_{i}$ and $e_{j}$ whose constituent node labels are $\{{0,0,0,0,0,1}\}$ and $\{{0,0,0,1,1,1}\}$, respectively.
>         - Although their label-mixed-extents significantly differ, both hyperedges are counted as non-same-class-node hyperedges.
>         - In this manner, this extension may fail to measure the higher-order homophily in a hypergraph.
>     - Notably, the homophily analysis method (Veldt et al. 2023) generalizes the homophily metric the reviewer has mentioned (i.e., the proportion of intra-class edges to all edges).
> - **Experiment Q1.3 [Homophily Levels in Hypergraphs]**:
>     - Since the homophily analysis is based on visualization analysis, we could not report a single value that represents the homophily of a dataset. Instead, analysis should be conducted by each hyperedge size (each hyperedge size has a separate plot). For each size plot, the line color represents the node class.
>         - An ***upward trend line*** indicates the corresponding class nodes exhibit higher-order homophilic characteristics in the corresponding size hyperedges.
>         - A ***downward trend line*** indicates the corresponding class nodes exhibit higher-order non-homophilic characteristics in the corresponding size hyperedges.
>     - We analyze three benchmark datasets: Cora, IMDB, and AMiner. The results are in Figure 6 of the revised manuscript (Appendix). Notably, these three datasets exhibit different homophilic characteristics. Details regarding the homophily analysis are as follows.
>         - **Cora** (*Homophilic*): Most classes show an upward trend.
>         - **IMDB** (*Non-Homophilic*): Most classes show a downward trend.
>         - **AMiner** (*Mixed*): Some classes show an upward trend and some classes show a downward trend.
> - **Experiment Q1.4 [Performance on Heterophilic Hypergraph]**:
>     - In short, HypeBoy outperforms other baseline methods in these three datasets: homophilic, non-homophilic, and partially homophilic hypergraphs (Table 1 of the main paper).
>     - Thus, we have verified that the empirical efficacy of HypeBoy is not limited to homophilic hypergraphs.
>     - This analysis is added to the revised manuscript.

---

> ### Author Response · Authors · 2023-11-17
> **Response to Question 2 [Experimental Setup]**
>
> ### Comment
>
> `I noticed that in section 5.1, the article uses a setting of 1% of the training set, which is different from most articles. Most other papers use fixed division or 5/10 nodes per category. I'm curious about what the considerations are for such an experimental setup?`
>
> ---
>
> ### Response
>
> - **Experiment Q2.1**: In the reviewer’s suggested setting, HypeBoy outperforms two SOTA hypergraph SSL methods (TriCL and HyperGCL).
>     - Since a fixed division split that is widely used for hypergraph node classification is unknown, we have employed the second setting: 5/10 training nodes per category.
>     - The results are shown below, where each method employs UniGCNII [4] as a backbone encoder.
>     - We have added this result to the revised manuscript as Table 11.
> - **Clarification Q2.2**: Moreover, we would like to clarify that our used setting is equivalent to the setting of Wei et al., 2022. [2].
>
> ---
>
> ### Additional Data Splits Experimental Results
>
> **5 training nodes**
>
> |  | Citeseer | Cora | Pubmed | Cora-CA | DBLP |
> | --- | --- | --- | --- | --- | --- |
> | No Pretraining | 0.537 (0.068) | 0.637 (0.038) | 0.716 (0.046) | 0.607 (0.049) | 0.778 (0.047) |
> | TriCL | 0.601 (0.052) | 0.710 (0.031) | 0.719 (0.038) | 0.676 (0.029) | 0.805 (0.025) |
> | HyperGCL | 0.570 (0.056) | 0.703 (0.029) | 0.722 (0.044) | 0.615 (0.044) | 0.795 (0.037) |
> | HypeBoy (Ours) | **0.631 (0.046)** | **0.720 (0.024)** | **0.729 (0.032)** | **0.678 (0.024)** | **0.815 (0.024)** |
>
> **10 training nodes**
>
> |  | Citeseer | Cora | Pubmed | Cora-CA | DBLP |
> | --- | --- | --- | --- | --- | --- |
> | No Pretraining | 0.627 (0.038) | 0.728 (0.022) | 0.737 (0.030) | 0.677 (0.030) | 0.824 (0.019) |
> | TriCL | 0.663 (0.026) | 0.757 (0.023) | 0.745 (0.031) | 0.697 (0.022) | 0.836 (0.019) |
> | HyperGCL | 0.644 (0.039) | 0.749 (0.021) | 0.743 (0.031) | 0.690 (0.034) | 0.836 (0.016) |
> | HypeBoy (Ours) | **0.677 (0.026)** | **0.758 (0.027)** | **0.749 (0.027)** | **0.709 (0.017)** | **0.846 (0.015)** |

---

> ### Author Response · Authors · 2023-11-17
> **References**
>
> `[1]: Dongjin Lee and Kijung Shin. I’m me, we’re us, and i’m us: Tri-directional contrastive learning on hypergraphs. In AAAI, 2023.`
>
> `[2]: Tianxin Wei, Yuning You, Tianlong Chen, Yang Shen, Jingrui He, and Zhangyang Wang. Augmentations in hypergraph contrastive learning: Fabricated and generative. In NeurIPS, 2022.`
>
> `[3]: Nate Veldt, Austin R Benson, and Jon Kleinberg. Combinatorial characterizations and impossibilities for higher-order homophily. Science Advances, 9(1):eabq3200, 2023.`
>
> `[4]: Jing Huang and Jie Yang. Unignn: a unified framework for graph and hypergraph neural networks. In IJCAI, 2021.`

---

### Official Review · Reviewer_4MNM · 2023-10-29

**Soundness:** 4 excellent
**Presentation:** 3 good
**Contribution:** 3 good
**Rating:** 8
**Confidence:** 4

**Summary:**

The paper makes an in-time contribution to studying the generative pretraining strategy for hypergraph neural networks. Specifically,  a novel generative SSL task on hypergraphs, hyperedge filling, is proposed, with the sound analysis demonstrating its effectiveness for node classification tasks. Extensive experiments are performed to support the claim.

The main intuition behind the magic of hyperedge filling is that the hyperedge, when it satisfies the homophily assumption, is indicative of the node membership and eventually helps node classification. This analysis supports the intuition. Although there might be some inconsistency between theory and practice, I appreciate the analysis part a lot.

**Strengths:**

- Nice and reasonable intuition to motivate the algorithm: The forming of hyperedge indicates the nodes share the same (or similar) membership. This might be motivated by the stochastic block models. It is a great intuition and I think it fits many applications such as social networks.
- Sound theoretical analysis to support the intuition. This is done by analyzing how the representations change to optimize hypergraph filling loss can improve node classification results.
- Extensive empirical results are provided. More appreciatively, the numerical characteristics, such as proximity, are examined to reach certain conclusions.

**Weaknesses:**

I do not have major criticisms for this paper. I only have some questions regarding the analysis part, which might result from missing some points during reading.

-  In the hyperedge filling analysis, it seems to not relate to the neural network architecture. How do authors think the choice of the hypergraph neural networks would affect the performance?
- In the hyperedge filling process (F2), the representation is updated with a gradient w.r.t. the hyperedge filling loss and a step size $\gamma$. Per my reading of the proof, the value of $\gamma$ (only need to be greater than 0) seems to not affect the result, while in practice this might not be true. Do I miss some points here?
- Is the edge filling optimal to perform as it is in hypergraphs? Considering I am gonna perform clique expansion to get a graph and perform edge filling. How would the result be different?

**Questions:**

Please see Weaknesses

---

> ### Author Response · Authors · 2023-11-17
> **Dear Reviewer 4MNM**
>
> Dear Reviewer 4MNM,
>
> We appreciate your generous review. We provide our responses to each of your comments below.

---

> > ### Author Response · Authors · 2023-11-17
> > **Response to Weakness 3 [Edge Filling on Clique-Expanded Graphs]**
> >
> > ### Comment
> >
> > `Is the edge filling optimal to perform as it is in hypergraphs? Considering I am gonna perform clique expansion to get a graph and perform edge filling. How would the result be different?`
> >
> > ---
> >
> > ### Response
> >
> > - Thank you for your excellent suggestion of an additional baseline.
> > - **Argument W3.1**: Since clique-expansion may cause loss of higher-order interaction information, we hypothesize that (hyper)edge filling on clique-expanded hypergraph would degrade model performance.
> > - **Experiment W3.2**: As hypothesized, both graph edge filling and hyperedge filling upon clique-expanded graph show suboptimal performance.
> >     - We have incorporated the suggested baseline methods from the reviewer, and they show suboptimal performance. The results are shown in the below table.
> >     - Specifically, the baseline method is designed as follows:
> >         - Graph edge filling method
> >             - We first transform a hypergraph into a graph via clique expansion.
> >             - Then, the method fills each edge following the procedure HypeBoy employs for a size-2 hyperedge.
> >             - GCN [6] has been used as a backbone encoder of the method.
> >             - We have added this result to the revised manuscript as Table 9.
> >         - Hyperedge filling upon clique-expanded graph
> >             - We first transform a hypergraph into a graph via clique expansion.
> >             - Here, this method performs hyperedge filling with obtained node embeddings.
> >             - GCN [6] has been used as a backbone encoder of the method.
> >             - We have added this result to the revised manuscript as Table 9.
> >
> > ---
> >
> > ### Graph Filling Baseline Method Result
> >
> > |  | Citeseer | Cora | Pubmed | Cora-CA | DBLP |
> > | --- | --- | --- | --- | --- | --- |
> > | Graph + Edge filling | 0.441 (0.113) | 0.557 (0.082) | 0.736 (0.038) | 0.555 (0.082) | 0.869 (0.005) |
> > | Graph + Hyperedge Filling | 0.428 (0.092) | 0564 (0.060) | 0.736 (0.040) | 0.550 (0.086) | 0.867 (0.005) |
> > | HypeBoy (Ours) | **0.567 (0.098)** | **0.623 (0.077)** | **0.770 (0.034)** | **0.663 (0.046)** | **0.882 (0.004)** |

---

> ### Author Response · Authors · 2023-11-17
> **Response to Weakness 1 [Choice of Hypergraph Neural Network]**
>
> ### Comment
>
> `In the hyperedge filling analysis, it seems to not relate to the neural network architecture. How do authors think the choice of the hypergraph neural networks would affect the performance?`
>
> ---
>
> ### Response
>
> - **Argument W1.1**: We have verified that a neural network that preserves the higher-order information of each hyperedge aligns well with HypeBoy.
> - **Empirical Support W1.2**: We have employed four different encoders as the backbone encoders of HypeBoy. Among the four, the encoder that uses higher-order information the least, HGNN [1], shows the worst performance.
>     - Specifically, the encoders are HGNN [1], MeanPoolingConv [2], SetGNN [3], and UniGCNII [4].
>     - A noticeable difference between HGNN and the other encoders is their use of input data structure: HGNN transforms a hypergraph into a graph via clique-expansion and performs graph convolution, while the others conduct message passing on the hypergraph structure itself (details are described in Section 2.1 of the main paper).
>     - As discussed by Dong et al., [5], clique expansion may cause significant information loss in higher-order interactions, potentially causing suboptimal performance of a machine learning model.
>     - To effectively solve the hyperedge filling task, an encoder should be capable of learning higher-order interactions, and such a requirement may cause a suboptimal performance of HGNN.
>
> ---
>
> ### Encoder Experiments
>
> |  | Average ranking | Citeseer | Cora | Cora-CA | Pubmed | DBLP-P |
> | --- | --- | --- | --- | --- | --- | --- |
> | HGNN + HypeBoy | 3.4 | 0.521 (0.091) | 0.611 (0.098) | 0.600 (0.060) | 0.768 (0.043) | 0.874 (0.005) |
> | MeanPoolingConv + HypeBoy | 2.2 | 0.545 (0.083) | 0.612 (0.078) | 0.630 (0.044) | 0.773 (0.034) | 0.868 (0.005) |
> | SetGNN + HypeBoy | 3.0 | 0.538 (0.091) | 0.623 (0.064) | 0.610 (0.039) | 0.734 (0.034) | 0.842 (0.007) |
> | UniGCNII + HypeBoy | **1.4**|  **0.567 (0.098)** | **0.623 (0.077)** | **0.663 (0.046)** | **0.770 (0.034)** | **0.882 (0.004)** |

---

> ### Author Response · Authors · 2023-11-17
> **Response to Weakness 2 [Step Size in Practice]**
>
> ### Comment
>
> `In the hyperedge filling process (F2), the representation is updated with a gradient w.r.t. the hyperedge filling loss and a step size. Per my reading of the proof, the value of (only need to be greater than 0) seems to not affect the result, while in practice this might not be true. Do I miss some points here?`
>
> ---
>
> ### Response
>
> - **Clarification W2.1**: Theorem 1 only states that the updated embedding is more beneficial and does not discuss the extent of benefit, which is expected to be influenced by the step size, as the reviewer pointed out.
> - **Clarification W2.2**: In practice, as the reviewer has mentioned, the efficacy of the hyperedge filling task is influenced by the step size. On the other hand, with a fixed step size of 0.001, HypeBoy outperforms existing hypergraph SSL methods in most of the datasets (in all our experiments, we have fixed the step size of 0.001, while that for other methods are tuned. Details are in Appendix).
> - **Argument W2.3**: Thus, we conclude that HypeBoy is relatively robust to the step size.
> - **Clarification W2.4:** Apart from the theoretical results, we delineate some possible reasons why step size may affect the node classification performance
>     - Real-world datasets may not (perfectly) align with our data assumptions that are used in theoretical analysis.
>     - We use a hypergraph neural network to obtain representations of nodes in practice, while we have assumed each representation as a free variable that is directly being updated.

---

> ### Author Response · Authors · 2023-11-17
> **References**
>
> `[1]: Yifan Feng, Haoxuan You, Zizhao Zhang, Rongrong Ji, and Yue Gao. Hypergraph neural networks. In AAAI, 2019.`
>
> `[2]: Dongjin Lee and Kijung Shin. I’m me, we’re us, and i’m us: Tri-directional contrastive learning on hypergraphs. In AAAI, 2023.`
>
> `[3]: Eli Chien, Chao Pan, Jianhao Peng, and Olgica Milenkovic. You are allset: A multiset function framework for hypergraph neural networks. In ICLR, 2022.`
>
> `[4]: Jing Huang and Jie Yang. Unignn: a unified framework for graph and hypergraph neural networks. In IJCAI, 2021.`
>
> `[5]: Yihe Dong, Will Sawin, and Yoshua Bengio. Hnhn: Hypergraph networks with hyperedge neurons. In ICML Workshop on Graph Representation Learning and Beyond (GRL+), 2020.`
>
> `[6]: Thomas N Kipf and Max Welling. Semi-supervised classification with graph convolutional networks. In ICLR, 2017.`

---

> ### Comment · Reviewer_4MNM · 2023-11-21
> **Thank You for the Response**
>
> I acknowledge the response and would like to increase my positive rate 6 --> 8.

---

> > ### Author Response · Authors · 2023-11-21
> > **To Reviewer 4MNM**
> >
> > Dear Reviewer 4MNM,
> >
> > We greatly appreciate your comments. Thank you for the time and effort you dedicated to reviewing our paper!
> >
> > Warm regards, The Authors.

---

### Official Review · Reviewer_ctQb · 2023-11-01

**Soundness:** 3 good
**Presentation:** 3 good
**Contribution:** 2 fair
**Rating:** 5
**Confidence:** 3

**Summary:**

The paper introduces a generative self-supervised learning task called "hypergraph filling"  and explores generative self-supervised learning on hypergraphs. The author focuses on addressing issues related to overemphasized proximity, dimensional collapse, and non-uniformity/alignment problems in learned representations. To tackle these issues, the author proposes the "HYPEBOY" strategy for hypergraphs, both in theory and through empirical experiments. Furthermore, the author demonstrates the effectiveness of this approach in tasks such as node classification and link prediction.

**Strengths:**

The author effectively identifies the issue of overemphasized proximity and demonstrates the beneficial impact of augmentation. Additionally, to address the problem of dimensional collapse, the author introduces a two-stage training scheme, which helps reduce the reliance on projection heads.

**Weaknesses:**

1. The rationale behind employing generative SSL for hypergraph representation is not convincingly established, and it confronts several challenges, including dimensional collapse.
2. The author devises a SSL strategy for hypergraphs by utilizing existing encoders and decoders such as UniGCNII, HNN, and MLP, without introducing any novel model designs. The concept of the projection head for hypergraph encoding is inspired by Deep Sets[1].
3.The author utilizes UniGCNII[2] as an encoder for HYPERBOY, and primarily focuses on homogeneous hypergraphs. However, it's important to note that heterogeneous hypergraphs are also prevalent, and the embedding method for hyperedges is not discussed.

[1] Manzil Zaheer, Satwik Kottur, Siamak Ravanbakhsh, Barnabas Poczos, Russ R Salakhutdinov, and Alexander J Smola. Deep sets. NeurIPS 2017: 3391-3401.
[2] Jing Huang, Jie Yang. UniGNN: a Unified Framework for Graph and Hypergraph Neural Networks.IJCAI 2021: 2563-2569.

**Questions:**

1. It's not straightforward to extend edge reconstruction methods to hyperedges in SSL. Considering this challenge, why did the author choose to employ SSL for hypergraphs without addressing the embeddings of hyperedges explicitly.
2. Could you elaborate on the real-world applications of the hypergraph filling task? How is this task practically relevant?
3. The author uses Gaussian distribution, Bernoulli sampling, and binomial distribution in the method. Could you explain the reasoning behind these choices and how they compare to more conventional methods like attention strategies and neural network approaches?

---

> ### Author Response · Authors · 2023-11-17
> **Dear Reviewer ctQb,**
>
> Dear Reviewer ctQb,
>
> We express our gratitude for your considerate review. We summarize our response to each of your comments below. In doing so, please let us first make an overarching clarification.

---

> ### Author Response · Authors · 2023-11-17
> **Overall Clarification**
>
> - We clarify that one of the primary goals of Self-Supervised Learning (SSL) is to empower a machine learning model to better solve some ***downstream** **tasks***.
> - To this end, the majority of SSL methods solve ***pretext** **tasks***, which are distinct from the target ***downstream tasks***.
> - Our primary contribution is devising a ***pretext task*** for hypergraph representation learning, called the hyperedge filling task.
>     - **The goal of our pretext task**: By solving the hyperedge filling task, we mainly aim to better solve the node classification task, which is one of the most popular downstream tasks in the hypergraph representation learning field.
>     - **Theoretical connection**: While our hyperedge filling task is seemingly not related to node classification, we have theoretically analyzed that solving the hyperedge filling task is beneficial for node classification, as described in Section 3.2 of the main paper.

---

> ### Author Response · Authors · 2023-11-17
> **Response to Weakness 1 [Rationale of Hyperedge Filling Task]**
>
> ### Comment
> `The rationale behind employing generative SSL for hypergraph representation is not convincingly established, and it confronts several challenges, including dimensional collapse.`
>
> ---
>
> ### Response
> - As described in our **Overall Clarification**, we aimed to better solve the node classification task by empowering a hypergraph neural network with the proposed hyperedge filling task.
> - **Hyperedge filling helps node classification (Sec. 3)**: Theoretically, solving the proposed SSL pretext task, hyperedge filling, can improve node classification performance. Moreover, we have experimentally demonstrated that our method HypeBoy outperforms other baseline methods.
> - **Findings in other domains (Sec. 1)**: The effectiveness of generative SSL has been demonstrated in many downstream tasks for a range of domains [1, 2].
> - **Utility in label-scarce scenarios (Sec. 1)**: Efficacy of (semi-)supervised learning (e.g., by directly aiming at node classification) is limited when available labels are scarce. Generative SSL tackles such a label-scarcity issue by providing abundant labels from the hypergraph structure itself.
>     - We experimentally validate this with our experiments in Table 1. See higher average ranks of SSL methods over the (Semi-)Supervised ones.

---

> ### Author Response · Authors · 2023-11-17
> **Response to Weakness 2 [Contribution on Neural Network]**
>
> ### Comment
> `The author devises a SSL strategy for hypergraphs by utilizing existing encoders and decoders such as UniGCNII, HNN, and MLP, without introducing any novel model designs. The concept of the projection head for hypergraph encoding is inspired by Deep Sets[1].`
>
> ---
>
> ### Response
>
> - We make the following clarifications:
>     - **Clarification W2.1**: We emphasize that our main contribution lies in ***devising a pretext task*** for hypergraph SSL, as described in Section 2.1 of the main paper, not proposing a novel neural network architecture for hypergraphs.
>     - **Clarification W2.2**: Moreover, as evidenced in the below table, the proposed hyperedge filling task can empower various hypergraph neural networks, not only UniGCNII, in the node classification task.
>         - We have additionally employed three hypergraph neural networks, which were used as backbone encoders of SOTA hypergraph SSL methods (TriCL [3] and HyperGCL [4]).
>         - As shown below, HypeBoy outperforms other SSL methods in most of the settings.
> - As such, we argue the following:
>     - **Argument W2.3**: Devising a pretext task, hyperedge filling, that can empower a wide range of hypergraph neural networks is our significant contribution.
>         - Furthermore, given HypeBoy’s potential applicability to a broad spectrum of hypergraph neural networks, our contribution is on par with introducing a novel neural network architecture.
>     - **Argument W2.4**: Devising an SSL method (consisting of augmentation, encoding, and loss function) that satisfies the desired properties (Property 1 - 3 in Section 4 of the main paper) is our non-trivial contribution.
>     - **Argument W2.5**: Thus, our contributions are valid even if we employ existing neural networks as the hypergraph encoders.
>
> ---
>
> ### Additional encoder experimental results
>
> **Encoder: HGNN [5], used as the encoder for TriCL.**
>
> |  | Citeseer | Cora | Pubmed | Cora-CA | DBLP-P |
> | --- | --- | --- | --- | --- | --- |
> | W/O Pretraining | 0.419 (0.078) | 0.500 (0.072) | 0.729 (0.050) | 0.502 (0.057) | 0.853 (0.008) |
> | TriCL | 0.493 (0.103) | 0.569 (0.010) | 0.743 (0.041) | 0.572 (0.061) | **0.874 (0.005)** |
> | HyperGCL | 0.472 (0.093) | 0.595 (0.076) | 0.763 (0.043) | 0.533 (0.069) | 0.858 (0.004) |
> | HypeBoy (Ours) | **0.521 (0.091)** | **0.611 (0.098)** | **0.768 (0.043)** | **0.600 (0.060)** | **0.874 (0.005)** |
>
> **Encoder: MeanPoolingConv [3], used as the encoder for TriCL.**
>
> |  | Citeseer | Cora | Pubmed | Cora-CA | DBLP-P |
> | --- | --- | --- | --- | --- | --- |
> | W/O Pretraining | 0.399 (0.100) | 0.485 (0.089) | 0.724 (0.040) | 0.506 (0.075) | 0.857 (0.007) |
> | TriCL | 0.466 (0.083) | 0.598 (0.089) | 0.740 (0.038) | 0.573 (0.048) | **0.870 (0.005)** |
> | HyperGCL | 0.432 (0.094) | 0.596 (0.076) | 0.742 (0.043) | 0.550 (0.067) | 0.857 (0.009) |
> | HypeBoy (Ours) | **0.545 (0.083)** | **0.612 (0.078)** | **0.773 (0.034)** | **0.630 (0.044)** | 0.868 (0.005) |
>
> **Encoder: SetGNN [6], used as the encoder for HyperGCL.**
>
> |  | Citeseer | Cora | Pubmed | Cora-CA | DBLP-P |
> | --- | --- | --- | --- | --- | --- |
> | W/O Pretraining | 0.436 (0.063) | 0.489 (0.092) | 0.698 (0.045) | 0.507 (0.067) | 0.828 (0.010) |
> | TriCL | 0.489 (0.076) | 0.577 (0.084) | 0.720 (0.045) | 0.578 (0.049) | **0.851 (0.005)** |
> | HyperGCL | 0.468 (0.085) | 0.546 (0.075) | 0.731 (0.039) | 0.569 (0.053) | 0.836 (0.007) |
> | HypeBoy (Ours) | **0.538 (0.091)** | **0.623 (0.064)** | **0.733 (0.039)** | **0.610 (0.039)** | 0.842 (0.007) |

---

> ### Author Response · Authors · 2023-11-17
> **Response to Weakness 3 [Heterogeneous Hypergraphs and Hyperedge Embedding]**
>
> ### Comment
> `The author utilizes UniGCNII[2] as an encoder for HYPERBOY, and primarily focuses on homogeneous hypergraphs. However, it's important to note that heterogeneous hypergraphs are also prevalent, and the embedding method for hyperedges is not discussed.`
>
> ---
>
> ### Response
> - To address your concerns, we conducted further experiments about heterogeneous hypergraphs. We summarize the results below:
>     - **Experiment W3.1:** In a heterogeneous hypergraph dataset, we have verified that HypeBoy outperforms two SOTA hypergraph SSL methods.
>         - **Dataset**: We found one heterogeneous hypergraph dataset (the ACM dataset [7]) that meets the following criteria: (1) exhibiting heterogeneity, (2) having node attributes and labels, and (3) being publicly accessible.
>         - **Results**: HypeBoy achieves the best performance among the SOTA SSL baselines. The results are provided in the below table. We have added this result to the revised manuscript as Table 12.
> - Further, we make the following clarification w.r.t. hyperedge embeddings:
>     - **Clarification W3.2:** Our primary goal is to obtain node representations for node classification. In this process, we do not explicitly obtain or utilize hyperedge embeddings.
>     - **Clarification W3.3:** Potentially, however, we can obtain good hyperedge embedding by appropriately employing node representations obtained from a hypergraph neural network.
>
> ---
>
> ### Results on Heterogeneous Hypergraph (ACM)
>
> |  | UniGCNII | TriCL | HyperGCL | HypeBoy (Ours) |
> | --- | --- | --- | --- | --- |
> | ACM  | 0.400 (0.031) | 0.403 (0.024) | 0.408 (0.034) | **0.417 (0.026)** |

---

> ### Author Response · Authors · 2023-11-17
> **Response to Question 1 [Extending Edge Reconstruction Method and Hyperedge Embedding]**
>
> ### Comment
> `It's not straightforward to extend edge reconstruction methods to hyperedges in SSL. Considering this challenge, why did the author choose to employ SSL for hypergraphs without addressing the embeddings of hyperedges explicitly.`
>
> ---
>
> ### Response
>
> - We make the following clarifications:
>     - **Clarification Q1.1:** Explicit hyperedge embeddings are not required for both hyperedge filling and node classification, which are the focus of this paper.
>         - Hyperedge filling task
>             - This task aims to fill a correct node to a given subset of a hyperedge. To this end, we employ representations of a node and a subset, whose steps are described in Section 4.2.
>             - Thus, we do not need hyperedge embeddings to perform hyperedge filling.
>         - Node classification task
>             - This task aims to correctly classify a node. To this end, we employ a node representation.
>             - Thus, we do not need hyperedge embeddings to perform node classification.
>     - **Experiment Q1.2**: Our proposed *hyperedge filling* method outperforms the *hyperedge reconstruction* method, which is a direct extension of graph edge reconstruction.
>         - *Hypergraph reconstruction* method details are as follows:
>             - For each SSL training epoch, we create negative-sample hyperedges by using the size-negative-sampling strategy, which is described in Appendix D.3.
>             - Then, we encourage an encoder to maximize the probability of the real hyperedges, while minimizing that of the negative-sample hyperedges.
>             - To obtain the probability of a hyperedge, we (1) obtain a hyperedge embedding via sum-pooling of embeddings of nodes that are included in the corresponding hyperedge and (2) feed the hyperedge embedding to the MLP classifier.
>         - We have added this result to the revised manuscript as Table 9.
> - Under such context, we make the following argument:
>     - **Argument Q1.3:** The superiority of the *hyperedge filling* task, which we have theoretically shown the effectiveness on the node classification, over the *hyperedge reconstruction* method is evidenced by the results in (**Experiment Q1.2**).
>
> ---
>
> ### Hyperedge Reconstruction SSL Method
>
> |  | Citeseer | Cora | Pubmed | Cora-CA | DBLP-P |
> | --- | --- | --- | --- | --- | --- |
> | Hyperedge Reconstruction | 0.439 (0.076) | 0.569 (0.097) | 0.730 (0.049) | 0.591 (0.065) | 0.858 (0.006) |
> | HypeBoy (Ours) | **0.567 (0.098)** | **0.623 (0.077)** | **0.770 (0.034)** | **0.663 (0.046)** | **0.882 (0.004)** |

---

> ### Author Response · Authors · 2023-11-17
> **Response to Question 2 [Application of Hyperedge Filling Task]**
>
> ### Comment
>
> `Could you elaborate on the real-world applications of the hypergraph filling task? How is this task practically relevant?`
>
> ---
>
> ### Response
>
> - As described in the **Overall Clarification**, our primary objective is to empower a hypergraph neural network to better solve the node classification task. That is, we employed the proposed *hyperedge filling* task as a means to achieve this goal, rather than considering it as the final objective.
> - While *hyperedge filling* is not our ultimate goal, *hyperedge filling* itself has potential applications described below. We have added these potential applications in our revised manuscript.
>     - **Email recipient recommendation**
>         - Given recipients of an email, recommend a user likely to be added as a recipient of the email.
>         - Nodes in a hypergraph indicate a user and each hyperedge indicates an email, consisting of a sender, recipients, and CCs.
>     - **Item recommendation**
>         - Given a shopping cart list, recommend an item to be co-purchased with listed items.
>         - Nodes in a hypergraph indicate an item and each hyperedge contains a set of co-purchased items.

---

> ### Author Response · Authors · 2023-11-17
> **Response to Question 3 [Assumptions]**
>
> ### Comment
>
> `The author uses Gaussian distribution, Bernoulli sampling, and binomial distribution in the method. Could you explain the reasoning behind these choices and how they compare to more conventional methods like attention strategies and neural network approaches?`
>
> ---
>
> ### Response
>
> - We clarify that the mentioned conditions are only assumed for the simplicity of our theoretical analysis (Section 3.2), and they are not utilized in our design of HypeBoy, which is our proposed SSL method.
> - We have clarified this in our revised manuscript.

---

> ### Author Response · Authors · 2023-11-17
> **References**
>
> `[1]: Kaiming He, Xinlei Chen, Saining Xie, Yanghao Li, Piotr Dollar, and Ross Girshick. Masked autoencoders are scalable vision learners. In CVPR, 2022.`
>
> `[2]: Zhenyu Hou, Xiao Liu, Yukuo Cen, Yuxiao Dong, Hongxia Yang, Chunjie Wang, and Jie Tang. Graphmae: Self-supervised masked graph autoencoders. In KDD, 2022.`
>
> `[3]: Dongjin Lee and Kijung Shin. I’m me, we’re us, and i’m us: Tri-directional contrastive learning on hypergraphs. In AAAI, 2023.`
>
> `[4]: Tianxin Wei, Yuning You, Tianlong Chen, Yang Shen, Jingrui He, and Zhangyang Wang. Augmentations in hypergraph contrastive learning: Fabricated and generative. In NeurIPS, 2022.`
>
> `[5]: Yifan Feng, Haoxuan You, Zizhao Zhang, Rongrong Ji, and Yue Gao. Hypergraph neural networks. In AAAI, 2019.`
>
> `[6]: Eli Chien, Chao Pan, Jianhao Peng, and Olgica Milenkovic. You are allset: A multiset function framework for hypergraph neural networks. In ICLR, 2022.`
>
> `[7]: Xiao Wang, Houye Ji, Chuan Shi, Bai Wang, Yanfang Ye, Peng Cui, and Philip S Yu. Heterogeneous graph attention network. In WWW, 2019.`

---

> > ### Author Response · Authors · 2023-11-22
> >
> > Dear Reviewer ctQb,
> >
> > Thank you for your thorough review of our paper. Since the discussion phase is close to the end, we would like to inquire if our responses have addressed your concerns, and if so, whether you could consider changing your rating. We remain fully committed to address any questions you may have by the end of the discussion phase.
> >
> > We sincerely appreciate your time and effort in reviewing our paper. We eagerly await your response!
> >
> > Best, \
> > The Authors

---

> > > ### Comment · Reviewer_ctQb · 2023-11-23
> > >
> > > Thank you for your responses, they have enhanced my understanding of the paper's intent. While I grasp the concept that hyperedge filling contributes to node classification, I believe merely relying on experiments and results to demonstrate the motivation behind generative Semi-Supervised Learning (SSL) may be insufficient. The issue of dimensional collapse introduced by generative SSL lacks a reasoned explanation.
> > > The authors advocate for concentrating on the tasks of hyperedge filling and node classification. Complementary experiments have indeed confirmed the superior performance of the hyperedge filling method compared to the hyperedge reconstruction method, but this claim is based solely on homogeneous hyperedge experiments. How does this method fare in the context of heterogeneous hyperedges?

---

> > > > ### Author Response · Authors · 2023-11-23
> > > > **To Reviewer ctQb**
> > > >
> > > > Dear reviewer ctQb,
> > > >
> > > > We genuinely appreciate the reviewer’s reply. We provide our responses here to address the expressed concerns. We gently request the reviewer to consider the following in their final decision.
> > > >
> > > > ## Response to Weakness 1 [Motivation]
> > > >
> > > > ### Weakness
> > > >
> > > > `While I grasp the concept that hyperedge filling contributes to node classification, I believe merely relying on experiments and results to demonstrate the motivation behind generative Semi-Supervised Learning (SSL) may be insufficient.`
> > > >
> > > > ### Response
> > > >
> > > > - Contrary to the reviewer’s claim, we did NOT ‘merely rely on experiments and results to demonstrate the motivation.’
> > > > - Our further motivations for using generative SSL are as follows:
> > > >     - **Theoretical Motivation (Theorem 2, Sec. 3.2)**: We theoretically showed how solving the proposed generative SSL task, *hyperedge filling*, can improve node classification performance.
> > > >     - **Motivation from Other Domains (Sec. 1)**: Please note that many SOTA methods in various domains, even including GPTs, utilize generative self-supervision [1, 2, 3]. Such findings suggest the strong potential of generative SSL in representation learning.
> > > >
> > > > ---
> > > >
> > > > ## Response to Weakness 2 [Dimensional collapse]
> > > >
> > > > ### Weakness
> > > >
> > > > `The issue of dimensional collapse introduced by generative SSL lacks a reasoned explanation.`
> > > >
> > > > ### Response
> > > >
> > > > - Contrary to the reviewer’s claim, we DID provide ‘a reasoned explanation’ on how the proposed method tackles dimensional collapse, both ***theoretically and empirically***.
> > > > - We summarize here how we tackle dimensional collapse:
> > > >     - **Experiment (Figure 2(a), Sec. 4.2)**: The proposed method effectively mitigates dimensional collapse *by using projection heads*.
> > > >     - **Theory (App. B.2)**: *Without a projection head*, the method may suffer from dimensional collapse issues.
> > > >
> > > > ---
> > > >
> > > > ## Response to Weakness 3 [Heterogeneous hypergraph]
> > > >
> > > > ### Weakness
> > > >
> > > > `The authors advocate for concentrating on the tasks of hyperedge filling and node classification. Complementary experiments have indeed confirmed the superior performance of the hyperedge filling method compared to the hyperedge reconstruction method, but this claim is based solely on homogeneous hyperedge experiments. How does this method fare in the context of heterogeneous hyperedges?`
> > > >
> > > > ### Response
> > > >
> > > > - To further address your concerns, we have conducted an additional experiment. (See the Table below).
> > > > - **Experimental Outcome**: In hyperedge prediction, our method outperforms the hyperedge reconstruction method in a heterogeneous hypergraph dataset.
> > > >     - **Baseline**: Hyperedge reconstruction-based method, as the reviewer suggested.
> > > >     - **Hyperparameters**: Due to the limited available time, we only tuned self-supervised learning epochs, without extensive hyperparameter search.
> > > >         - We will ready the fully-tuned results in the final version.
> > > >     - **Dataset**: ACM [4] (the public heterogeneous hypergraph with node attributes).
> > > >         - We kindly remind the reviewer that we found only one heterogeneous hypergraph dataset (ACM) that meets the following criteria: (1) exhibits heterogeneity; (2) has node attributes and labels; (3) being publicly accessible.
> > > > - If the reviewer can suggest us additional heterogeneous hypergraph datasets with node attributes, we will be happy to conduct additional experiments in our camera-ready version.
> > > >
> > > > |  | Hyperedge reconstruction | Hyperedge filling (Ours) |
> > > > | --- | --- | --- |
> > > > | ACM dataset | 0.396 (0.031) | 0.417 (0.026) |
> > > >
> > > > ---
> > > >
> > > > ---
> > > >
> > > > In summary, we are gratified that we were able to address all the reviewer’s expressed concerns. Please consider our responses in the final decision.
> > > >
> > > > Best regards,
> > > >
> > > > Authors.
> > > >
> > > > ---
> > > >
> > > > ---
> > > >
> > > > ## Reference
> > > >
> > > > `[1]: Kaiming He, Xinlei Chen, Saining Xie, Yanghao Li, Piotr Dollar, and Ross Girshick. Masked autoencoders are scalable vision learners. In CVPR, 2022.`
> > > >
> > > > `[2]: Zhenyu Hou, Xiao Liu, Yukuo Cen, Yuxiao Dong, Hongxia Yang, Chunjie Wang, and Jie Tang. Graphmae: Self-supervised masked graph autoencoders. In KDD, 2022.`
> > > >
> > > > `[3]: OpenAI. Gpt-4 technical report. 2023`
> > > >
> > > > `[4]: Xiao Wang, Houye Ji, Chuan Shi, Bai Wang, Yanfang Ye, Peng Cui, and Philip S Yu. Heterogeneous graph attention network. In WWW, 2019.`

---

### Official Review · Reviewer_3LGX · 2023-11-01

**Soundness:** 3 good
**Presentation:** 3 good
**Contribution:** 2 fair
**Rating:** 6
**Confidence:** 3

**Summary:**

This paper introduces the first generative SSL methods, especially designed for hypergraph learning. The proposed self-supervised task is to fill in the missing node from an incomplete hyperedge. The authors show theoretically and empirically that, unde some conditions, this self-supervised task is well aligned with hypernode classification, achieving better results than other contrastive SSL methods from the hypergraph literature.

**Strengths:**

- The paper identify and fill in a gap existent in the hypergraph literature: designing generative SSL tasks for hypergraph representation learning. The proposed hyperedge filling task is intuitive and well-suited for hypergraph representation learning. The theoretical finding enhances confidence in the approach.
- The experimental section is comprehensive, demonstrating the individual contribution of each component.
- The paper is generally well written.

**Weaknesses:**

- All the SSL-based results presented in the paper uses UniGCNII as a backbone. The authors motivate this decision by saying that this combinations achieves best overall performances. While I consider the comparison fair (since all the reported baselines uses UniGCNII as well), it is essential to conduct experiments that demonstrate the method's advantages when applied with various backbones. This additional experiments would clearly demonstrate the advantages of the proposed method.
- While the authors did a good job in explaining the intuition behind this, it is somewhat discouraging to observe that the suggested approach performs noticeably worse when the feature reconstruction warmup is omitted (Table 3)
- Minor: It would be useful to include the UNIGCNII baseline (without any of the 3 component) as a line in Table 3
- Given the big improvement brought by the augmentation (masking) scheme (Table 6 in appendix), I am curious to know if the other baseline methods benefit from a similar augmentation step.

**Questions:**

Please see the Weaknesses section

---

> ### Author Response · Authors · 2023-11-17
> **Dear Reviewer 3LGX**
>
> Dear Reviewer 3LGX,
>
> We deeply appreciate your constructive review. Below, we provide our response to each of your comments. Our empirical outcomes have become more solid, thanks to your comments.

---

> ### Author Response · Authors · 2023-11-17
> **Response to Weakness 1 [Various Backbone Encoders]**
>
> ### Comment
>
> `All the SSL-based results presented in the paper uses UniGCNII as a backbone. The authors motivate this decision by saying that this combinations achieves best overall performances. While I consider the comparison fair (since all the reported baselines uses UniGCNII as well), it is essential to conduct experiments that demonstrate the method's advantages when applied with various backbones. This additional experiments would clearly demonstrate the advantages of the proposed method.`
>
> ---
>
> ### Response
>
> - We additionally have employed three hypergraph encoders used in SOTA hypergraph SSL methods (TriCL [1] and HyperGCL [2]).
>     - **Additional encoders:** HGNN [3], MeanPoolingConv [1], and SetGNN [4]
> - Our method (HypeBoy) outperformed all SOTA hypergraph SSL methods in all the used encoders. The results are presented in the below tables.
>     - **Details**: Compared to the 2nd-best SSL method (TriCL), HypeBoy achieves up to 4.2% better accuracy in all the encoder settings.
> - We have added these results in Table 7 of our revised manuscript.
>
> ---
>
> ### Encoder Experiments
>
> **Encoder: HGNN [3], used in TriCL [1].**
>
> |  | Citeseer | Cora | Pubmed | Cora-CA | DBLP-P |
> | --- | --- | --- | --- | --- | --- |
> | W/O Pretraining | 0.419 (0.078) | 0.500 (0.072) | 0.729 (0.050) | 0.502 (0.057) | 0.853 (0.008) |
> | TriCL | 0.493 (0.103) | 0.569 (0.010) | 0.743 (0.041) | 0.572 (0.061) | **0.874 (0.005)** |
> | HyperGCL | 0.472 (0.093) | 0.595 (0.076) | 0.763 (0.043) | 0.533 (0.069) | 0.858 (0.004) |
> | HypeBoy (Ours) | **0.521 (0.091)** | **0.611 (0.098)** | **0.768 (0.043)** | **0.600 (0.060)** | **0.874 (0.005)** |
>
> **Encoder: MeanPoolingConv [1], used in TriCL [1].**
>
> |  | Citeseer | Cora | Pubmed | Cora-CA | DBLP-P |
> | --- | --- | --- | --- | --- | --- |
> | W/O Pretraining | 0.399 (0.100) | 0.485 (0.089) | 0.724 (0.040) | 0.506 (0.075) | 0.857 (0.007) |
> | TriCL | 0.466 (0.083) | 0.598 (0.089) | 0.740 (0.038) | 0.573 (0.048) | **0.870 (0.005)**|
> | HyperGCL | 0.432 (0.094) | 0.596 (0.076) | 0.742 (0.043) | 0.550 (0.067) | 0.857 (0.009) |
> | HypeBoy (Ours) | **0.545 (0.083)**| **0.612 (0.078)** | **0.773 (0.034)** | **0.630 (0.044)** | 0.868 (0.005) |
>
>
> **Encoder: SetGNN [4], used in HyperGCL [2].**
>
> |  | Citeseer | Cora | Pubmed | Cora-CA | DBLP-P |
> | --- | --- | --- | --- | --- | --- |
> | W/O Pretraining | 0.436 (0.063) | 0.489 (0.092) | 0.698 (0.045) | 0.507 (0.067) | 0.828 (0.010) |
> | TriCL | 0.489 (0.076) | 0.577 (0.084) | 0.720 (0.045) | 0.578 (0.049) | **0.851 (0.005)**|
> | HyperGCL | 0.468 (0.085) | 0.546 (0.075) | 0.731 (0.039) | 0.569 (0.053) | 0.836 (0.007) |
> | HypeBoy (Ours) | **0.538 (0.091)** | **0.623 (0.064)** | **0.734 (0.034)** | **0.610 (0.039)** | 0.842 (0.007) |

---

> ### Author Response · Authors · 2023-11-17
> **Response to Weakness 2 [Limited Performance of Only Hyperedge Filling Method]**
>
> ### Comment
> `While the authors did a good job in explaining the intuition behind this, it is somewhat discouraging to observe that the suggested approach performs noticeably worse when the feature reconstruction warmup is omitted (Table 3).`
>
> ---
>
> ### Response
>
> - We make the following clarifications:
> - **Clarification W2.1**: Our method outperforms two SOTA hypergraph SSL methods, even without feature reconstruction warmup.
>     - In terms of the average ranks, *HypeBoy variant V2* (no feature reconstruction) outperforms *TriCL* and *HyperGCL* (the SOTA hypergraph SSL methods).
>     - This can be derived by comparing Tables 1 and 3 of the main paper.
> - **Clarification W2.2**: Moreover, the *hyperedge filling* task is more effective than *feature reconstruction*.
>     - In terms of the average ranks, *V2* (hyperedge filling with projection heads) outperforms *V3* (only feature reconstruction).
> - **Clarification W2.3**: Lastly, feature reconstruction on hypergraphs is also our contribution, since we are the first to employ the feature reconstruction scheme on hypergraphs, to our best knowledge.

---

> ### Author Response · Authors · 2023-11-17
> **Response to Weakness 3 [Adding UNIGCNII also in ablation study]**
>
> ### Comment
> `Minor: It would be useful to include the UNIGCNII baseline (without any of the 3 component) as a line in Table 3`
>
> ---
>
> ### Response
> - We have revised Table 3 based on your comment: UniGCNII [5] is added as an additional baseline in Table 3.
> - As we originally concluded, we find that each component of HypeBoy contributes to its performance gain.
>
> ---
>
> ### Updated Table 3 (The first line “NA” is added).
> The ablation study with four variants of HypeBoy on node classification under the fine-tuning protocol.
> F.R., H.F., and P.H. denote Feature Reconstruction, Hyperedge Filling, and Projection Heads, respectively. A.R. denotes the average ranking among all methods.
> ***NA denotes no pretraining***. HypeBoy outperforms others in most datasets, justifying each of its components.
>
> |  | F.R. | H.F. | P.H. | Citeseer | Cora | Pubmed | Cora-CA | DBLP-P | DBLP-A | AMiner | IMDB | MN-40 | 20 News | House | A.R. |
> | --- | --- | --- | --- | --- | --- | --- | --- | --- | --- | --- | --- | --- | --- | --- | --- |
> | NA | ✗ | ✗ | ✗ | 0.442 (0.090) | 0.485 (0.074) | 0.741 (0.039) | 0.548 (0.070) | 0.874 (0.006) | 0.658 (0.039) | 0.325 (0.017) | 0.425 (0.039) | 0.908 (0.011) | 0.709 (0.010) | 0.508 (0.043) | 5.5 |
> | V1 | ✗ | ✓ | ✗ | 0.516 (0.112) | 0.607 (0.082) | 0.762 (0.036) | 0.635 (0.060) | 0.881 (0.005) | 0.785 (0.029) | 0.335 (0.028) | 0.468 (0.031) | 0.900 (0.011) | 0.774 (0.009) | 0.685 (0.045) | 4.3 |
> | V2 | ✗ | ✓ | ✓ | 0.527 (0.096) | 0.597 (0.092) | 0.767 (0.032) | 0.635 (0.060) | 0.882 (0.005) | 0.791 (0.039) | 0.338 (0.022) | 0.469 (0.033) | 0.906 (0.010) | 0.770 (0.009) | 0.696 (0.049) | 3.3 |
> | V3 | ✓ | ✗ | ✗ | 0.520 (0.093) | 0.589 (0.082) | 0.741 (0.039) | 0.612 (0.066) | 0.878 (0.004) | 0.799 (0.023) | 0.339 (0.021) | 0.463 (0.027) | 0.914 (0.009) | 0.775 (0.009) | 0.701 (0.048) | 3.6 |
> | V4 | ✓ | ✓ | ✗ | 0.560 (0.099) | 0.618 (0.085) | 0.765 (0.031) | 0.653 (0.043) | 0.880 (0.004) | 0.803 (0.024) | 0.340 (0.020) | 0.475 (0.023) | 0.908 (0.010) | 0.774 (0.010) | 0.693 (0.050) | 2.5 |
> | Ours | ✓ | ✓ | ✓ | 0.567 (0.098) | 0.623 (0.077) | 0.770 (0.034) | 0.663 (0.046) | 0.882 (0.004) | 0.806 (0.023) | 0.341 (0.022) | 0.476 (0.025) | 0.904 (0.009) | 0.776 (0.009) | 0.704 (0.048) | 1.4 |

---

> ### Author Response · Authors · 2023-11-17
> **Response to Weakness 4 [Augmentation of Baseline Methods]**
>
> ### Comment
> `Given the big improvement brought by the augmentation (masking) scheme (Table 6 in appendix), I am curious to know if the other baseline methods benefit from a similar augmentation step.`
>
> ---
>
> ### Response
>
> - **Clarification W4.1:** HyperGCL and TriCL (the SOTA hypergraph SSL baseline methods) also employ augmentation strategies, and their results were obtained with the augmentations.
>     - Specifically, both HyperGCL and TriCL are contrastive learning methods that create hypergraph views by augmenting input hypergraphs.
>     - To this end, they utilize the below methods:
>         - HyperGCL: This method generates augmented views of a hypergraph by using a hypergraph VAE model. The hypergraph VAE model is being trained with contrastive loss.
>         - TriCL: This method masks certain columns of the input node feature matrix and drops a certain number of nodes from each hyperedge.
> - **Argument W4.2:** Thus, we emphasize the *fairness* of our experiments and the *superiority* of our proposed model and self-supervision loss.

---

> ### Author Response · Authors · 2023-11-17
> **References**
>
> `[1]: Dongjin Lee and Kijung Shin. I’m me, we’re us, and i’m us: Tri-directional contrastive learning on hypergraphs. In AAAI, 2023.`
>
> `[2]: Tianxin Wei, Yuning You, Tianlong Chen, Yang Shen, Jingrui He, and Zhangyang Wang. Augmentations in hypergraph contrastive learning: Fabricated and generative. In NeurIPS, 2022.`
>
> `[3]: Yifan Feng, Haoxuan You, Zizhao Zhang, Rongrong Ji, and Yue Gao. Hypergraph neural networks. In AAAI, 2019`
>
> `[4]: Eli Chien, Chao Pan, Jianhao Peng, and Olgica Milenkovic. You are allset: A multiset function framework for hypergraph neural networks. In ICLR, 2022.`
>
> `[5]: Jing Huang and Jie Yang. Unignn: a unified framework for graph and hypergraph neural networks. In IJCAI, 2021.`

---

> ### Comment · Reviewer_3LGX · 2023-11-21
> **Rebuttal reply**
>
> Dear Authors,
>
> Thank you for carefully addressing the review.
>
> Before the rebuttal, my main concern regards the absence of experiments involving multiple backbones. The additional experiments included in the response demonstrate that the proposed method yields significant improvements irrespective of the hypergraph processing method. I believe that this is an important empirical contribution, which strengthen and validates the proposed approach.
>
> In light of this, I decided to increases my score to 6: marginally above the acceptance threshold.
>
> Best,
> Reviewer 3LGX

---

> > ### Author Response · Authors · 2023-11-21
> > **To Reviewer 3LGX**
> >
> > Dear Reviewer 3LGX,
> >
> > Thank you for your valuable feedback! We are gratified to know that our response has met your principal concerns. Your time and effort in reviewing our paper are deeply appreciated.
> >
> > Warm regards, The Authors.

---

### Author Response · Authors · 2023-11-17
**For All the Reviewers**

Dear Esteemed Reviewers

We would like to express our sincere gratitude to all the reviewers for their insightful feedback. Your comments have been invaluable in strengthening the substance and clarity of our work.

We have diligently addressed your concerns and have thoroughly revised our manuscript accordingly. The parts influenced by your suggestions have been highlighted in blue for ease of reference.

We thank all the reviewers again and remain open to further dialogue regarding any additional questions you may have.

Warm regards,
The Authors.

---

### Author Response · Authors · 2023-11-20
**Gratitude for Your Reviews and Open for Further Discussions**

Dear Reviewers,

We express our gratitude for your valuable time in reviewing our work. We are available to address any additional questions you may have regarding our work or rebuttal. Kindly inform us if there are any such concerns.

Sincerely,

The Authors.

---

### Meta-Review · Area_Chair_G4Ct · 2023-12-06

**Metareview:**

This paper presents a novel generative self-supervised representation learning method for hypergraphs. It considers the task of hyperedge  filling for which the authors shows a theoretical connection with node classification. The proposed method, called HypeBoy, is composed of 3 steps: (i) hypergraph augmentation, (ii) hypergraph encoding, and (iii) hypergraph filling. The method is evaluated on multiple benchmarks.


On the positive side, the reviews have underlined that designing Semi-Supervised Learning (SSL) tasks for hypergraph augmentation appears to be novel and interesting, the approach follows a reasonable intuition, the proposed 2-stage method is interesting, the theoretical analysis is sound, the experimental evaluation is extensive and clear and finally the paper is well written.
On the negative side, experiments could consider other backbone and more generally other methods, additionally other baselines could be considered, dimensional collapse could be more discussed, the discussion/presentation can be improved, the assumption of considering the same category for each hyper-edge and the interest of hyper-edge filling alone can be more justified.

The authors did a strong effort to provide a thorough rebuttal.
They provide notably additional experiments provided in Appendix E of the revision, without being exhaustive, these new experiments include considering different backbones and new methods, the extension of the ablation study, the use of other encoders, heterogeneous hypergraphs,  hyperedge reconstruction, edge filling on clique-expanded graphs, node swapping for assessing the impact of homophily,  effectiveness of hyper edge alone, ...
They also add  clarifications on the method, precisions on the assumptions, mode collapse.

Most of the issues raised by the reviewers were addressed. The submission was significantly improved with respect to the remarks raised by reviewers. The contribution appears of good quality.
I propose then acceptance.

**Justification For Why Not Higher Score:**

Many weaknesses have been identified in the original submission and even if the authors did a strong job to address reviewers issues, I think that the paper does not worth a spotlight.

**Justification For Why Not Lower Score:**

The setting is novel and interesting with good results. The rebuttal helped to improve the submission.

---

### Decision · Program_Chairs · 2024-01-16

Accept (poster)